# Can Temporal-Difference and Q-Learning Learn Representation? A Mean-Field Analysis

**Yufeng Zhang**
Northwestern University
Evanston, IL 60208
yufengzhang2023@u.northwestern.edu

**Qi Cai**
Northwestern University
Evanston, IL 60208
qicai2022@u.northwestern.edu

**Zhuoran Yang**
Princeton University
Princeton, NJ 08544
zy6@princeton.edu

**Yongxin Chen**
Georgia Institute of Technology
Atlanta, GA 30332
yongchen@gatech.edu

**Zhaoran Wang**
Northwestern University
Evanston, IL 60208
zhaoranwang@gmail.com

## Abstract

Temporal-difference and Q-learning play a key role in deep reinforcement learning, where they are empowered by expressive nonlinear function approximators such as neural networks. At the core of their empirical successes is the learned feature representation, which embeds rich observations, e.g., images and texts, into the latent space that encodes semantic structures. Meanwhile, the evolution of such a feature representation is crucial to the convergence of temporal-difference and Q-learning.

In particular, temporal-difference learning converges when the function approximator is linear in a feature representation, which is fixed throughout learning, and possibly diverges otherwise. We aim to answer the following questions:

*When the function approximator is a neural network, how does the associated feature representation evolve? If it converges, does it converge to the optimal one?*

We prove that, utilizing an overparameterized two-layer neural network, temporal-difference and Q-learning globally minimize the mean-squared projected Bellman error at a sublinear rate. Moreover, the associated feature representation converges to the optimal one, generalizing the previous analysis of [21] in the neural tangent kernel regime, where the associated feature representation stabilizes at the initial one. The key to our analysis is a mean-field perspective, which connects the evolution of a finite-dimensional parameter to its limiting counterpart over an infinite-dimensional Wasserstein space. Our analysis generalizes to soft Q-learning, which is further connected to policy gradient.

## 1 Introduction

Deep reinforcement learning achieves phenomenal empirical successes, especially in challenging applications where an agent acts upon rich observations, e.g., images and texts. Examples include video gaming [56], visuomotor manipulation [51], and language generation [39]. Such empirical successes are empowered by expressive nonlinear function approximators such as neural networks, which are used to parameterize both policies (actors) and value functions (critics) [46]. In particular, the neural network learned from interacting with the environment induces a data-dependent feature representation, which embeds rich observations into a latent space encoding semantic structures

[12, 40, 49, 75]. In contrast, classical reinforcement learning mostly relies on a handcrafted feature representation that is fixed throughout learning [65].

In this paper, we study temporal-difference (TD) [64] and Q-learning [71], two of the most prominent algorithms in deep reinforcement learning, which are further connected to policy gradient [73] through its equivalence to soft Q-learning [37, 57, 58, 61]. In particular, we aim to characterize how an overparameterized two-layer neural network and its induced feature representation evolve in TD and Q-learning, especially their rate of convergence and global optimality. A fundamental obstacle, however, is that such an evolving feature representation possibly leads to the divergence of TD and Q-learning. For example, TD converges when the value function approximator is linear in a feature representation, which is fixed throughout learning, and possibly diverges otherwise [10, 18, 67].

To address such an issue of divergence, nonlinear gradient TD [15] explicitly linearizes the value function approximator locally at each iteration, that is, using its gradient with respect to the parameter as an evolving feature representation. Although nonlinear gradient TD converges, it is unclear whether the attained solution is globally optimal. On the other hand, when the value function approximator in TD is an overparameterized multi-layer neural network, which is required to be properly scaled, such a feature representation stabilizes at the initial one [21], making the explicit local linearization in nonlinear gradient TD unnecessary. Moreover, the implicit local linearization enabled by overparameterization allows TD (and Q-learning) to converge to the globally optimal solution. However, such a required scaling, also known as the neural tangent kernel (NTK) regime [43], effectively constrains the evolution of the induced feature presentation to an infinitesimal neighborhood of the initial one, which is not data-dependent.

**Contribution.** Going beyond the NTK regime, we prove that, when the value function approximator is an overparameterized two-layer neural network, TD and Q-learning globally minimize the mean-squared projected Bellman error (MSPBE) at a sublinear rate. Moreover, in contrast to the NTK regime, the induced feature representation is able to deviate from the initial one and subsequently evolve into the globally optimal one, which corresponds to the global minimizer of the MSPBE. We further extend our analysis to soft Q-learning, which is connected to policy gradient.

The key to our analysis is a mean-field perspective, which allows us to associate the evolution of a finite-dimensional parameter with its limiting counterpart over an infinite-dimensional Wasserstein space [4, 5, 68, 69]. Specifically, by exploiting the permutation invariance of the parameter, we associate the neural network and its induced feature representation with an empirical distribution, which, at the infinite-width limit, further corresponds to a population distribution. The evolution of such a population distribution is characterized by a partial differential equation (PDE) known as the continuity equation. In particular, we develop a generalized notion of one-point monotonicity [38], which is tailored to the Wasserstein space, especially the first variation formula therein [5], to characterize the evolution of such a PDE solution, which, by a discretization argument, further quantifies the evolution of the induced feature representation.

**Related Work.** When the value function approximator is linear, the convergence of TD is extensively studied in both continuous-time [16, 17, 42, 47, 67] and discrete-time [14, 29, 48, 63] settings. See [31] for a detailed survey. Also, when the value function approximator is linear, [25, 55, 78] study the convergence of Q-learning. When the value function approximator is nonlinear, TD possibly diverges [10, 18, 67]. [15] propose nonlinear gradient TD, which converges but only to a locally optimal solution. See [13, 36] for a detailed survey. When the value function approximator is an overparameterized multi-layer neural network, [21] prove that TD converges to the globally optimal solution in the NTK regime. See also the independent work of [1, 19, 20, 62], where the state space is required to be finite. In contrast to the previous analysis in the NTK regime, our analysis allows TD to attain a data-dependent feature representation that is globally optimal.

Meanwhile, our analysis is related to the recent breakthrough in the mean-field analysis of stochastic gradient descent (SGD) for the supervised learning of an overparameterized two-layer neural network [23, 27, 34, 35, 44, 53, 54, 72]. See also the previous analysis in the NTK regime [2, 3, 7–9, 22, 24, 26, 30, 32, 33, 43, 45, 50, 52, 76, 77]. Specifically, the previous mean-field analysis casts SGD as the Wasserstein gradient flow of an energy functional, which corresponds to the objective function in supervised learning. In contrast, TD follows the stochastic semigradient of the MSPBE [65], which is biased. As a result, there does not exist an energy functional for casting TD as its Wasserstein

gradient flow. Instead, our analysis combines a generalized notion of one-point monotonicity [38] and the first variation formula in the Wasserstein space [5], which is of independent interest.

**Notations.** We denote by $\mathscr{B}(\mathcal{X})$ the Borel $\sigma$-algebra over the space $\mathcal{X}$. Let $\mathscr{P}(\mathcal{X})$ be the set of Borel probability measures over the measurable space $(\mathcal{X}, \mathscr{B}(\mathcal{X}))$. We denote by $[N] = \{1, 2, \ldots, N\}$ for any $N \in \mathbb{N}_+$. Also, we denote by $\mathcal{B}^n(x; r) = \{y \in \mathbb{R}^n \,|\, \|y - x\| \leq r\}$ the closed ball in $\mathbb{R}^n$. Given a curve $\rho : \mathbb{R} \to \mathcal{X}$, we denote by $\rho'_s = \partial_t \rho_t \,|_{t=s}$ its derivative with respect to the time. For a function $f : \mathcal{X} \to \mathbb{R}$, we denote by $\mathrm{Lip}(f) = \sup_{x,y \in \mathcal{X}, x \neq y} |f(x) - f(y)| / \|x - y\|$ its Lipschitz constant. For an operator $F : \mathcal{X} \to \mathcal{X}$ and a measure $\mu \in \mathscr{P}(\mathcal{X})$, we denote by $F_\sharp \mu = \mu \circ F^{-1}$ the push forward of $\mu$ through $F$. We denote by $D_{\mathrm{KL}}$ and $D_{\chi^2}$ the Kullback-Leibler (KL) divergence and the $\chi^2$ divergence, respectively.

## 2 Background

### 2.1 Policy Evaluation

We consider a Markov decision process $(\mathcal{S}, \mathcal{A}, P, R, \gamma, \mathcal{D}_0)$, where $\mathcal{S} \subseteq \mathbb{R}^{d_1}$ is the state space, $\mathcal{A} \subseteq \mathbb{R}^{d_2}$ is the action space, $P : \mathcal{S} \times \mathcal{A} \to \mathscr{P}(\mathcal{S})$ is the transition kernel, $R : \mathcal{S} \times \mathcal{A} \to \mathscr{P}(\mathbb{R})$ is the reward distribution, $\gamma \in (0, 1)$ is the discount factor, and $\mathcal{D}_0 \in \mathscr{P}(\mathcal{S})$ is the initial state distribution. An agent following a policy $\pi : \mathcal{S} \to \mathscr{P}(\mathcal{A})$ interacts with the environment in the following manner. At a state $s_t$, the agent takes an action $a_t$ according to $\pi(\cdot \,|\, s_t)$ and receives from the environment a random reward $r_t$ following $R(\cdot \,|\, s_t, a_t)$. Then, the environment transits into the next state $s_{t+1}$ according to $P(\cdot \,|\, s_t, a_t)$. We measure the performance of a policy $\pi$ via the expected cumulative reward $J(\pi)$, which is defined as follows,

$$J(\pi) = \mathbb{E}\Big[\sum_{t=0}^{\infty} \gamma^t \cdot r_t \,\Big|\, s_0 \sim \mathcal{D}_0, a_t \sim \pi(\cdot \,|\, s_t), r_t \sim R(\cdot \,|\, s_t, a_t), s_{t+1} \sim P(\cdot \,|\, s_t, a_t)\Big]. \quad (2.1)$$

In policy evaluation, we are interested in the state-action value function (Q-function) $Q^\pi : \mathcal{S} \times \mathcal{A} \to \mathbb{R}$, which is defined as follows,

$$Q^\pi(s, a) = \mathbb{E}\Big[\sum_{t=0}^{\infty} \gamma^t \cdot r_t \,\Big|\, s_0 = s, a_0 = a, a_t \sim \pi(\cdot \,|\, s_t), r_t \sim R(\cdot \,|\, s_t, a_t), s_{t+1} \sim P(\cdot \,|\, s_t, a_t)\Big].$$

We learn the Q-function by minimizing the mean-squared Bellman error (MSBE), which is defined as follows,

$$\mathrm{MSBE}(Q) = \frac{1}{2} \cdot \mathbb{E}_{(s,a) \sim \mathcal{D}}\Big[\big(Q(s, a) - \mathcal{T}^\pi Q(s, a)\big)^2\Big].$$

Here $\mathcal{D} \in \mathscr{P}(\mathcal{S} \times \mathcal{A})$ is the stationary distribution induced by the policy $\pi$ of interest and $\mathcal{T}^\pi$ is the corresponding Bellman operator, which is defined as follows,

$$\mathcal{T}^\pi Q(s, a) = \mathbb{E}\big[r + \gamma \cdot Q(s', a') \,\big|\, r \sim R(\cdot \,|\, s, a), s' \sim P(\cdot \,|\, s, a), a' \sim \pi(\cdot \,|\, s')\big].$$

However, $\mathcal{T}^\pi Q$ may be not representable by a given function class $\mathcal{F}$. Hence, we turn to minimizing a surrogate of the MSBE over $Q \in \mathcal{F}$, namely the mean-squared projected Bellman error (MSPBE), which is defined as follows,

$$\mathrm{MSPBE}(Q) = \frac{1}{2} \cdot \mathbb{E}_{(s,a) \sim \mathcal{D}}\Big[\big(Q(s, a) - \Pi_\mathcal{F} \mathcal{T}^\pi Q(s, a)\big)^2\Big], \quad (2.2)$$

where $\Pi_\mathcal{F}$ is the projection onto $\mathcal{F}$ with respect to the $\mathcal{L}_2(\mathcal{D})$-norm. The global minimizer of the MSPBE is the fixed point solution to the projected Bellman equation $Q = \Pi_\mathcal{F} \mathcal{T}^\pi Q$.

In temporal-difference (TD) learning, corresponding to the MSPBE defined in (2.2), we parameterize the Q-function with $\widehat{Q}(\cdot; \theta)$ and update the parameter $\theta$ via stochastic semigradient descent [65],

$$\theta' = \theta - \epsilon \cdot \big(\widehat{Q}(s, a; \theta) - r - \gamma \cdot \widehat{Q}(s', a'; \theta)\big) \cdot \nabla_\theta \widehat{Q}(s, a; \theta), \quad (2.3)$$

where $\epsilon > 0$ is the stepsize and $(s, a, r, s', a') \sim \widetilde{\mathcal{D}}$. Here we denote by $\widetilde{\mathcal{D}} \in \mathscr{P}(\mathcal{S} \times \mathcal{A} \times \mathbb{R} \times \mathcal{S} \times \mathcal{A})$ the distribution of $(s, a, r, s', a')$, where $(s, a) \sim \mathcal{D}$, $r \sim R(\cdot \,|\, s, a)$, $s' \sim P(\cdot \,|\, s, a)$, and $a' \sim \pi(\cdot \,|\, s')$.

## 2.2 Wasserstein Space

Let $\Theta \subseteq \mathbb{R}^D$ be a Polish space. We denote by $\mathscr{P}_2(\Theta) \subseteq \mathscr{P}(\Theta)$ the set of probability measures with finite second moments. Then, the Wasserstein-2 distance between $\mu, \nu \in \mathscr{P}_2(\Theta)$ is defined as follows,

$$\mathcal{W}_2(\mu, \nu) = \inf\Big\{ \mathbb{E}\big[\|X - Y\|^2\big]^{1/2} \,\Big|\, \mathrm{law}(X) = \mu, \mathrm{law}(Y) = \nu \Big\}, \tag{2.4}$$

where the infimum is taken over the random variables $X$ and $Y$ on $\Theta$. Here we denote by $\mathrm{law}(X)$ the distribution of a random variable $X$. We call $\mathcal{M} = (\mathscr{P}_2(\Theta), \mathcal{W}_2)$ the Wasserstein space, which is an infinite-dimensional manifold [69]. In particular, such a structure allows us to write any tangent vector at $\mu \in \mathcal{M}$ as $\rho_0'$ for a corresponding curve $\rho : [0, 1] \to \mathscr{P}_2(\Theta)$ that satisfies $\rho_0 = \mu$. Here $\rho_0'$ denotes $\partial_t \rho_t \,|_{t=0}$. Specifically, under certain regularity conditions, for any curve $\rho : [0, 1] \to \mathscr{P}_2(\Theta)$, the continuity equation $\partial_t \rho_t = -\mathrm{div}(\rho_t v_t)$ corresponds to a vector field $v : [0, 1] \times \Theta \to \mathbb{R}^D$, which endows the infinite-dimensional manifold $\mathscr{P}_2(\Theta)$ with a weak Riemannian structure in the following sense [69]. Given any tangent vectors $u$ and $\widetilde{u}$ at $\mu \in \mathcal{M}$ and the corresponding vector fields $v, \widetilde{v}$, which satisfy $u + \mathrm{div}(\mu v) = 0$ and $\widetilde{u} + \mathrm{div}(\mu\widetilde{v}) = 0$, respectively, we define the inner product of $u$ and $\widetilde{u}$ as follows,

$$\langle u, \widetilde{u} \rangle_\mu = \int \langle v, \widetilde{v} \rangle \, \mathrm{d}\mu, \tag{2.5}$$

which yields a Riemannian metric. Here $\langle v, \widetilde{v} \rangle$ is the inner product on $\mathbb{R}^D$. Such a Riemannian metric further induces a norm $\|u\|_\mu = \langle u, u \rangle_\mu^{1/2}$ for any tangent vector $u \in T_\mu \mathcal{M}$ at any $\mu \in \mathcal{M}$, which allows us to write the Wasserstein-2 distance defined in (2.4) as follows,

$$\mathcal{W}_2(\mu, \nu) = \inf\left\{ \left( \int_0^1 \|\rho_t'\|_{\rho_t}^2 \, \mathrm{d}t \right)^{1/2} \,\middle|\, \rho : [0, 1] \to \mathcal{M}, \rho_0 = \mu, \rho_1 = \nu \right\}. \tag{2.6}$$

Here $\rho_s'$ denotes $\partial_t \rho_t \,|_{t=s}$ for any $s \in [0, 1]$. In particular, the infimum in (2.6) is attained by the geodesic $\widetilde{\rho} : [0, 1] \to \mathscr{P}_2(\Theta)$ connecting $\mu, \nu \in \mathcal{M}$. Moreover, the geodesics on $\mathcal{M}$ are constant-speed, that is,

$$\|\widetilde{\rho}_t'\|_{\widetilde{\rho}_t} = \mathcal{W}_2(\mu, \nu), \quad \forall t \in [0, 1]. \tag{2.7}$$

# 3 Temporal-Difference Learning

For notational simplicity, we write $\mathbb{R}^d = \mathbb{R}^{d_1} \times \mathbb{R}^{d_2}$, $\mathcal{X} = \mathcal{S} \times \mathcal{A} \subseteq \mathbb{R}^d$, and $x = (s, a) \in \mathcal{X}$ for any $s \in \mathcal{S}$ and $a \in \mathcal{A}$.

**Parameterization of Q-Function.** We consider the parameter space $\mathbb{R}^D$ and parameterize the Q-function with the following two-layer neural network,

$$\widehat{Q}(x; \theta^{(m)}) = \frac{\alpha}{m} \sum_{i=1}^m \sigma(x; \theta_i), \tag{3.1}$$

where $\theta^{(m)} = (\theta_1, \ldots, \theta_m) \in \mathbb{R}^{D \times m}$ is the parameter, $m \in \mathbb{N}_+$ is the width, $\alpha > 0$ is the scaling parameter, and $\sigma : \mathbb{R}^d \times \mathbb{R}^D \to \mathbb{R}$ is the activation function. Assuming the activation function in (3.1) takes the form of $\sigma(x; \theta) = b \cdot \widetilde{\sigma}(x; w)$ for $\theta = (w, b)$, we recover the standard form of two-layer neural networks, where $\widetilde{\sigma}$ is the rectified linear unit or the sigmoid function. Such a parameterization is also used in [23, 26, 53]. For $\{\theta_i\}_{i=1}^m$ independently sampled from a distribution $\rho \in \mathscr{P}(\mathbb{R}^D)$, we have the following infinite-width limit of (3.1),

$$Q(x; \rho) = \alpha \cdot \int \sigma(x; \theta) \, \mathrm{d}\rho(\theta). \tag{3.2}$$

For the empirical distribution $\widehat{\rho}^{(m)} = m^{-1} \cdot \sum_{i=1}^m \delta_{\theta_i}$ corresponding to $\{\theta_i\}_{i=1}^m$, we have that $Q(x; \widehat{\rho}^{(m)}) = \widehat{Q}(x; \theta^{(m)})$.

**TD Dynamics.** In what follows, we consider the TD dynamics,

$$\theta_i(k+1)$$
$$= \theta_i(k) - \eta\epsilon \cdot \alpha \cdot \Big(\widehat{Q}\big(x_k; \theta^{(m)}(k)\big) - r_k - \gamma \cdot \widehat{Q}\big(x'_k; \theta^{(m)}(k)\big)\Big) \cdot \nabla_\theta \sigma\big(x_k; \theta_i(k)\big), \quad (3.3)$$

where $i \in [m]$, $(x_k, r_k, x'_k) \sim \widetilde{\mathcal{D}}$, and $\epsilon > 0$ is the stepsize with the scaling parameter $\eta > 0$. Without loss of generality, we assume that $(x_k, r_k, x'_k)$ is independently sampled from $\widetilde{\mathcal{D}}$, while our analysis straightforwardly generalizes to the setting of Markov sampling [14, 74, 78]. For an initial distribution $\rho_0 \in \mathscr{P}(\mathbb{R}^D)$, we initialize $\{\theta_i\}_{i=1}^m$ as $\theta_i \overset{\text{i.i.d.}}{\sim} \rho_0$ $(i \in [m])$. See Algorithm 1 in §A for a detailed description.

**Mean-Field Limit.** Corresponding to $\epsilon \to 0^+$ and $m \to \infty$, the continuous-time and infinite-width limit of the TD dynamics in (3.3) is characterized by the following partial differential equation (PDE) with $\rho_0$ as the initial distribution,

$$\partial_t \rho_t = -\eta \cdot \text{div}\big(\rho_t \cdot g(\cdot; \rho_t)\big). \quad (3.4)$$

Here $g(\cdot; \rho_t) : \mathbb{R}^D \to \mathbb{R}^D$ is a vector field, which is defined as follows,

$$g(\theta; \rho) = -\alpha \cdot \mathbb{E}_{(x,r,x') \sim \widetilde{\mathcal{D}}} \Big[\big(Q(x; \rho) - r - \gamma \cdot Q(x'; \rho)\big) \cdot \nabla_\theta \sigma(x; \theta)\Big]. \quad (3.5)$$

Note that (3.4) holds in the sense of distributions [5]. See [6, 53, 54] for the existence, uniqueness, and regularity of the PDE solution $\rho_t$ in (3.4). In the sequel, we refer to the continuous-time and infinite-width limit with $\epsilon \to 0^+$ and $m \to \infty$ as the mean-field limit. Let $\widehat{\rho}_k^{(m)} = m^{-1} \cdot \sum_{i=1}^m \delta_{\theta_i(k)}$ be the empirical distribution corresponding to $\{\theta_i(k)\}_{i=1}^m$ in (3.3). The following proposition proves that the PDE solution $\rho_t$ in (3.4) well approximates the TD dynamics $\theta^{(m)}(k)$ in (3.3).

**Proposition 3.1** (Informal Version of Proposition D.1). Let the initial distribution $\rho_0$ be the standard Gaussian distribution $N(0, I_D)$. Under certain regularity conditions, $\widehat{\rho}_{\lfloor t/\epsilon \rfloor}^{(m)}$ weakly converges to $\rho_t$ as $\epsilon \to 0^+$ and $m \to \infty$.

The proof of Proposition 3.1 is based on the propagation of chaos [53, 54, 66]. In contrast to [53, 54], the PDE in (3.4) can not be cast as a gradient flow, since there does not exist a corresponding energy functional. Thus, their analysis is not directly applicable to our setting. We defer the detailed discussion on the approximation analysis to §D. Proposition 3.1 allows us to convert the TD dynamics over the finite-dimensional parameter space to its counterpart over the infinite-dimensional Wasserstein space, where the infinitely wide neural network $Q(\cdot; \rho)$ in (3.2) is linear in the distribution $\rho$.

**Feature Representation.** We are interested in the evolution of the feature representation

$$\Big(\nabla_\theta \sigma\big(x; \theta_1(k)\big)^\top, \dots, \nabla_\theta \sigma\big(x; \theta_m(k)\big)^\top\Big)^\top \in \mathbb{R}^{Dm} \quad (3.6)$$

corresponding to $\theta^{(m)}(k) = (\theta_1(k), \dots, \theta_m(k)) \in \mathbb{R}^{D \times m}$. Such a feature representation is used to analyze the TD dynamics $\theta^{(m)}(k)$ in (3.3) in the NTK regime [21], which corresponds to setting $\alpha = \sqrt{m}$ in (3.1). Meanwhile, the nonlinear gradient TD dynamics [15] explicitly uses such a feature representation at each iteration to locally linearize the Q-function. Moreover, up to a rescaling, such a feature representation corresponds to the kernel

$$\mathbb{K}(x, x'; \widehat{\rho}_k^{(m)}) = \int \nabla_\theta \sigma(x; \theta)^\top \nabla_\theta \sigma(x'; \theta) \, \mathrm{d}\widehat{\rho}_k^{(m)}(\theta),$$

which by Proposition 3.1 further induces the kernel

$$\mathbb{K}(x, x'; \rho_t) = \int \nabla_\theta \sigma(x; \theta)^\top \nabla_\theta \sigma(x'; \theta) \, \mathrm{d}\rho_t(\theta) \quad (3.7)$$

at the mean-field limit with $\epsilon \to 0^+$ and $m \to \infty$. Such a correspondence allows us to use the PDE solution $\rho_t$ in (3.4) as a proxy for characterizing the evolution of the feature representation in (3.6).

# 4 Main Results

We first introduce the assumptions for our analysis.

**Assumption 4.1.** We assume that the state-action pair $x = (s, a)$ satisfies $\|x\| \leq 1$ for any $s \in \mathcal{S}$ and $a \in \mathcal{A}$.

Assumption 4.1 can be ensured by normalizing all state-action pairs. Such an assumption is commonly used in the mean-field analysis of neural networks [6, 23, 27, 34, 35, 53, 54]. We remark that our analysis straightforwardly generalizes to the setting where $\|x\| \leq C$ for an absolute constant $C > 0$.

**Assumption 4.2.** We assume that the activation function $\sigma$ in (3.1) satisfies

$$\left|\sigma(x;\theta)\right| \leq B_0, \quad \left\|\nabla_\theta \sigma(x;\theta)\right\| \leq B_1 \cdot \|x\|, \quad \left\|\nabla^2_{\theta\theta} \sigma(x;\theta)\right\|_{\mathrm{F}} \leq B_2 \cdot \|x\|^2 \qquad (4.1)$$

for any $x \in \mathcal{X}$. Also, we assume that the reward $r$ satisfies $|r| \leq B_r$.

Assumption 4.2 holds for a broad range of neural networks. For example, let $\theta = (w, b) \in \mathbb{R}^{D-1} \times \mathbb{R}$. The activation function

$$\sigma^\dagger(x;\theta) = B_0 \cdot \tanh(b) \cdot \mathrm{sigmoid}(w^\top x) \qquad (4.2)$$

satisfies (4.1) in Assumption 4.2. Moreover, the infinitely wide neural network in (3.2) with the activation function $\sigma^\dagger$ in (4.2) induces the following function class,

$$\mathcal{F}^\dagger = \left\{ \int \beta \cdot \mathrm{sigmoid}(w^\top x)\, \mathrm{d}\mu(w, \beta) \,\middle|\, \mu \in \mathscr{P}\big(\mathbb{R}^{D-1} \times [-B_0, B_0]\big) \right\},$$

where $\beta = B_0 \cdot \tanh(b) \in [-B_0, B_0]$. By the universal approximation theorem [11, 60], $\mathcal{F}^\dagger$ captures a rich class of functions.

Throughout the rest of this paper, we consider the following function class,

$$\mathcal{F} = \left\{ \int \sigma_0(b) \cdot \sigma_1(x; w)\, \mathrm{d}\rho(w, b) \,\middle|\, \rho \in \mathscr{P}_2(\mathbb{R}^{D-1} \times \mathbb{R}) \right\}, \qquad (4.3)$$

which is induced by the infinitely wide neural network in (3.2) with $\theta = (w, b) \in \mathbb{R}^{D-1} \times \mathbb{R}$ and the following activation function,

$$\sigma(x;\theta) = \sigma_0(b) \cdot \sigma_1(x; w).$$

We assume that $\sigma_0$ is an odd function, that is, $\sigma_0(b) = -\sigma_0(-b)$, which implies $\int \sigma(x;\theta)\, \mathrm{d}\rho_0(\theta) = 0$. Note that the set of infinitely wide neural networks taking the forms of (3.2) is $\alpha \cdot \mathcal{F}$, which is larger than $\mathcal{F}$ in (4.3) by the scaling parameter $\alpha > 0$. Thus, $\alpha$ can be viewed as the degree of "overrepresentation". Without loss of generality, we assume that $\mathcal{F}$ is complete. The following theorem characterizes the global optimality and convergence of the PDE solution $\rho_t$ in (3.4).

**Theorem 4.3.** There exists a unique fixed point solution to the projected Bellman equation $Q = \Pi_\mathcal{F} \mathcal{T}^\pi Q$, which takes the form of $Q^*(x) = \int \sigma(x;\theta)\, \mathrm{d}\bar\rho(\theta)$. Also, $Q^*$ is the global minimizer of the MSPBE defined in (2.2). We assume that $D_{\chi^2}(\bar\rho \,\|\, \rho_0) < \infty$ and $\bar\rho(\theta) > 0$ for any $\theta \in \mathbb{R}^D$. Under Assumptions 4.1 and 4.2, it holds for $\eta = \alpha^{-2}$ in (3.4) that

$$\inf_{t \in [0,T]} \mathbb{E}_{x \sim \mathcal{D}}\Big[\big(Q(x;\rho_t) - Q^*(x)\big)^2\Big] \leq \frac{D_{\chi^2}(\bar\rho \,\|\, \rho_0)}{2(1-\gamma) \cdot T} + \frac{C_*}{(1-\gamma) \cdot \alpha}, \qquad (4.4)$$

where $C_* > 0$ is a constant that depends on $D_{\chi^2}(\bar\rho \,\|\, \rho_0)$, $B_1$, $B_2$, and $B_r$.

*Proof.* See §5 for a detailed proof. □

Theorem 4.3 proves that the optimality gap $\mathbb{E}_{x \sim \mathcal{D}}[(Q(x;\rho_t) - Q^*(x))^2]$ decays to zero at a sublinear rate up to the error of $O(\alpha^{-1})$, where $\alpha > 0$ is the scaling parameter in (3.1). Varying $\alpha$ leads to a tradeoff between such an error of $O(\alpha^{-1})$ and the deviation of $\rho_t$ from $\rho_0$. Specifically, in §5 we prove that $\rho_t$ deviates from $\rho_0$ by the divergence $D_{\chi^2}(\rho_t \,\|\, \rho_0) \leq O(\alpha^{-2})$. Hence, a smaller $\alpha$ allows $\rho_t$ to move further away from $\rho_0$, inducing a feature representation that is more different from the initial one [34, 35]. See (3.6)-(3.7) for the correspondence of $\rho_t$ with the feature representation and

the kernel that it induces. On the other hand, a smaller $\alpha$ yields a larger error of $O(\alpha^{-1})$ in (4.4) of Theorem 4.3. In contrast, the NTK regime [21], which corresponds to setting $\alpha = \sqrt{m}$ in (3.1), only allows $\rho_t$ to deviate from $\rho_0$ by the divergence $D_{\chi^2}(\rho_t \| \rho_0) \leq O(m^{-1}) = o(1)$. In other words, the NTK regime fails to induce a feature representation that is significantly different from the initial one. In summary, our analysis goes beyond the NTK regime, which allows us to characterize the evolution of the feature representation towards the (near-)optimal one. Moreover, based on Proposition 3.1 and Theorem 4.3, we establish the following corollary, which characterizes the global optimality and convergence of the TD dynamics $\theta^{(m)}(k)$ in (3.3).

**Corollary 4.4.** Under the same conditions of Theorem 4.3, it holds with probability at least $1 - \delta$ that

$$\min_{\substack{k \leq T/\epsilon \\ (k \in \mathbb{N})}} \mathbb{E}_{x \sim \mathcal{D}} \left[ \left( \widehat{Q}(x; \theta^{(m)}(k)) - Q^*(x) \right)^2 \right] \leq \frac{D_{\chi^2}(\bar{\rho} \| \rho_0)}{2(1 - \gamma) \cdot T} + \frac{C_*}{(1 - \gamma) \cdot \alpha} + \Delta(\epsilon, m, \delta, T), \quad (4.5)$$

where $C_* > 0$ is the constant of (4.4) in Theorem 4.3 and $\Delta(\epsilon, m, \delta, T) > 0$ is an error term such that

$$\lim_{m \to \infty} \lim_{\epsilon \to 0^+} \Delta(\epsilon, m, \delta, T) = 0.$$

*Proof.* See §D.2 for a detailed proof. □

In (4.5) of Corollary 4.4, the error term $\Delta(\epsilon, m, \delta, T)$ characterizes the error of approximating the TD dynamics $\theta^{(m)}(k)$ in (3.3) using the PDE solution $\rho_t$ in (3.4). In particular, such an error vanishes at the mean-field limit.

## 5 Proof of Main Results

We first introduce two technical lemmas. Recall that $\mathcal{F}$ is defined in (4.3), $Q(x; \rho)$ is defined in (3.2), and $g(\theta; \rho)$ is defined in (3.5).

**Lemma 5.1.** There exists a unique fixed point solution to the projected Bellman equation $Q = \Pi_{\mathcal{F}} \mathcal{T}^\pi Q$, which takes the form of $Q^*(x) = \int \sigma(x; \theta) \, d\bar{\rho}(\theta)$. Also, there exists $\rho^* \in \mathscr{P}_2(\mathbb{R}^D)$ that satisfies the following properties,

(i) $Q(x; \rho^*) = Q^*(x)$ for any $x \in \mathcal{X}$,

(ii) $g(\cdot; \rho^*) = 0$ for $\bar{\rho}$-a.e., and

(iii) $\mathcal{W}_2(\rho^*, \rho_0) \leq \alpha^{-1} \cdot \bar{D}$, where $\bar{D} = D_{\chi^2}(\bar{\rho} \| \rho_0)^{1/2}$.

*Proof.* See §C.1 for a detailed proof. The proof of (iii) is adopted from [23], which focuses on supervised learning. □

Lemma 5.1 establishes the existence of the fixed point solution $Q^*$ to the projected Bellman equation $Q = \Pi_{\mathcal{F}} \mathcal{T}^\pi Q$. Furthermore, such a fixed point solution $Q^*$ can be parameterized with the infinitely wide neural network $Q(\cdot; \rho^*)$ in (3.2). Meanwhile, the Wasserstein-2 distance between $\rho^*$ and the initial distribution $\rho_0$ is upper bounded by $O(\alpha^{-1})$. Based on the existence of $Q^*$ and the property of $\rho^*$ in Lemma 5.1, we establish the following lemma that characterizes the evolution of $\mathcal{W}_2(\rho_t, \rho^*)$, where $\rho_t$ is the PDE solution in (3.4).

**Lemma 5.2.** We assume that $\mathcal{W}_2(\rho_t, \rho^*) \leq 2\mathcal{W}_2(\rho_0, \rho^*)$, $D_{\chi^2}(\bar{\rho} \| \rho_0) < \infty$, and $\bar{\rho}(\theta) > 0$ for any $\theta \in \mathbb{R}^D$. Under Assumptions 4.1 and 4.2, it holds that

$$\frac{\mathrm{d}}{\mathrm{d}t} \frac{\mathcal{W}_2(\rho_t, \rho^*)^2}{2} \leq -(1 - \gamma) \cdot \eta \cdot \mathbb{E}_{x \sim \mathcal{D}} \left[ \left( Q(x; \rho_t) - Q^*(x) \right)^2 \right] + C_* \cdot \alpha^{-1} \cdot \eta, \quad (5.1)$$

where $C_* > 0$ is a constant depending on $D_{\chi^2}(\bar{\rho} \| \rho_0)$, $B_1$, $B_2$, and $B_r$.

*Proof.* See §C.2 for a detailed proof. □

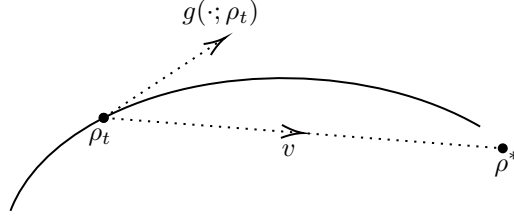

Figure 1: We illustrate the first variation formula $\frac{\mathrm{d}\mathcal{W}_2(\rho_t,\rho^*)^2}{2} = -\langle g(\cdot;\rho_t), v\rangle_{\rho_t}$, where $v$ is the vector field corresponding to the geodesic that connects $\rho_t$ and $\rho^*$. See Lemma E.2 for details.

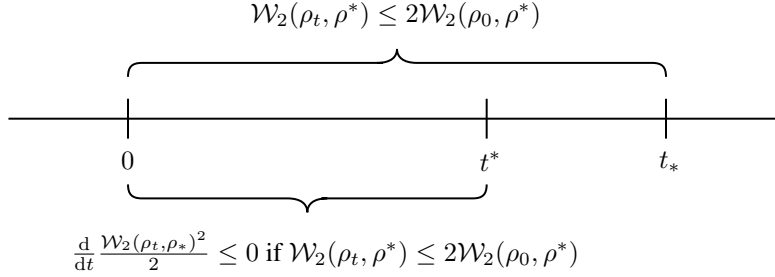

Figure 2: For any $0 \le t \le \min\{t^*, t_*\}$, (5.1) of Lemma 5.2 holds and $\frac{\mathrm{d}}{\mathrm{d}t}\frac{\mathcal{W}_2(\rho_t,\rho_*)^2}{2} \le 0$.

The proof of Lemma 5.2 is based on the first variation formula of the Wasserstein-2 distance (Lemma E.2), which is illustrated in Figure 1, and the one-point monotonicity of $g(\cdot;\beta_t)$ along a curve $\beta$ on the Wasserstein space (Lemma C.1). When the right-hand side of (5.1) is nonpositive, Lemma 5.2 characterizes the decay of $\mathcal{W}_2(\rho_t,\rho^*)$. We are now ready to present the proof of Theorem 4.3.

*Proof.* We use a continuous counterpart of the induction argument. We define

$$t^* = \inf\left\{\tau \in \mathbb{R}_+ \,\Big|\, \mathbb{E}_{x\sim\mathcal{D}}\Big[(1-\gamma)\cdot\big(Q(x;\rho_\tau) - Q^*(x)\big)^2\Big] < C_* \cdot \alpha^{-1}\right\}. \tag{5.2}$$

In other words, the right-hand side of (5.1) in Lemma 5.2 is nonpositive for any $t \le t^*$, that is,

$$-(1-\gamma)\cdot\mathbb{E}_{x\sim\mathcal{D}}\Big[\big(Q(x;\rho_t) - Q^*(x)\big)^2\Big] + C_* \cdot \alpha^{-1} \le 0. \tag{5.3}$$

Also, we define

$$t_* = \inf\big\{\tau \in \mathbb{R}_+ \,\big|\, \mathcal{W}_2(\rho_\tau,\rho^*) > 2\mathcal{W}_2(\rho_0,\rho^*)\big\}. \tag{5.4}$$

In other words, (5.1) of Lemma 5.2 holds for any $t \le t_*$. Thus, for any $0 \le t \le \min\{t^*, t_*\}$, it holds that $\frac{\mathrm{d}}{\mathrm{d}t}\frac{\mathcal{W}_2(\rho_t,\rho_*)^2}{2} \le 0$. Figure 2 illustrates the definition of $t^*$ and $t_*$ in (5.2) and (5.4), respectively.

We now prove that $t_* \ge t^*$ by contradiction. By the continuity of $\mathcal{W}_2(\rho_t,\rho^*)^2$ with respect to $t$ [5], it holds that $t_* > 0$, since $\mathcal{W}_2(\rho_0,\rho^*) < 2\mathcal{W}_2(\rho_0,\rho^*)$. For the sake of contradiction, we assume that $t_* < t^*$, by (5.1) of Lemma 5.2 and (5.3), it holds for any $0 \le t \le t_*$ that

$$\frac{\mathrm{d}}{\mathrm{d}t}\frac{\mathcal{W}_2(\rho_t,\rho^*)^2}{2} \le 0,$$

which implies that $\mathcal{W}_2(\rho_t,\rho^*) \le \mathcal{W}_2(\rho_0,\rho^*)$ for any $0 \le t \le t_*$. This contradicts the definition of $t_*$ in (5.4). Thus, it holds that $t_* \ge t^*$, which implies that (5.1) of Lemma 5.2 holds for any $0 \le t \le t^*$.

If $t^* \le T$, (5.3) implies Theorem 4.3. If $t^* > T$, by (5.1) of Lemma 5.2, it holds for any $0 \le t \le T$ that

$$\frac{\mathrm{d}}{\mathrm{d}t}\frac{\mathcal{W}_2(\rho_t,\rho^*)^2}{2} \le -(1-\gamma)\cdot\eta\cdot\mathbb{E}_{x\sim\mathcal{D}}\Big[\big(Q(x;\rho_t) - Q^*(x)\big)^2\Big] + C_* \cdot \alpha^{-1}\cdot\eta \le 0,$$

which further implies that

$$\mathbb{E}_{x \sim \mathcal{D}}\Big[\big(Q(x; \rho_t) - Q^*(x)\big)^2\Big] \leq -(1 - \gamma)^{-1} \cdot \eta^{-1} \cdot \frac{\mathrm{d}}{\mathrm{d}t} \frac{\mathcal{W}_2(\rho_t, \rho^*)^2}{2} + \frac{C_*}{(1 - \gamma) \cdot \alpha}. \qquad (5.5)$$

Upon telescoping (5.5) and setting $\eta = \alpha^{-2}$, we obtain that

$$\begin{aligned}
\inf_{t \in [0,T]} \mathbb{E}_{\mathcal{D}}&\Big[\big(Q(x; \rho_t) - Q^*(x)\big)^2\Big] \\
&\leq T^{-1} \cdot \int_0^T \mathbb{E}_{x \sim \mathcal{D}}\Big[\big(Q(x; \rho_t) - Q^*(x)\big)^2\Big] \mathrm{d}t \\
&\leq 1/2 \cdot (1 - \gamma)^{-1} \cdot \eta^{-1} \cdot T^{-1} \cdot \mathcal{W}_2(\rho_0, \rho^*)^2 + C_* \cdot (1 - \gamma)^{-1} \cdot \alpha^{-1} \\
&\leq 1/2 \cdot (1 - \gamma)^{-1} \cdot \bar{D}^2 \cdot T^{-1} + C_* \cdot (1 - \gamma)^{-1} \cdot \alpha^{-1},
\end{aligned}$$

where the last inequality follows from the fact that $\eta = \alpha^{-2}$ and (iii) of Lemma 5.1. Thus, we complete the proof of Theorem 4.3. $\qquad\square$

## 6 Extension to Q-Learning and Policy Improvement

In §B, we extend our analysis of TD to Q-learning and soft Q-learning for policy improvement. In §B.1, we introduce Q-learning and its mean-field limit. In §B.2, we establish the global optimality and convergence of Q-learning. In §B.3, we further extend our analysis to soft Q-learning, which is equivalent to a variant of policy gradient [37, 57, 58, 61].

## Broader Impact

The popularity of RL creates a responsibility for researchers to design algorithms with guaranteed safety and robustness, which rely on their stability and convergence. In this paper, we provide a theoretical understanding of the global optimality and convergence of the TD and Q-learning with neural network parameterization. We believe that our work is an important step forward in the algorithm design of RL in emerging high-stakes applications, such as autonomous driving, personalized medicine, power systems, and robotics.

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
