[Supplementary Material 1]

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

# A   Pseudocode of TD Learning

In this section, we present the pseudocode of TD learning in Algorithm 1, which is introduced in §3.

---

**Algorithm 1** Temporal-Difference Learning with Two-Layer Neural Network for Policy Evaluation

---

    **Initialization:** $\theta_i(0) \overset{\text{i.i.d.}}{\sim} \rho_0$ $(i \in [m])$, number of iterations $K = \lfloor T/\epsilon \rfloor$, and policy $\pi$ of interest.
    **for** $k = 0, \ldots, K - 1$ **do**
        Sample the state-action pair $(s, a)$ from the stationary distribution $\mathcal{D}$ of $\pi$, receive the reward $r$, and obtain the subsequent state-action pair $(s', a')$.
        Calculate the Bellman residual $\delta = \widehat{Q}(x; \theta^{(m)}(k)) - r - \gamma \cdot \widehat{Q}(x'; \theta^{(m)}(k))$, where $x = (s, a)$ and $x' = (s', a')$.
        Perform the TD update $\theta_i(k + 1) \leftarrow \theta_i(k) - \eta\epsilon \cdot \alpha \cdot \delta \cdot \nabla_\theta \sigma(x; \theta_i(k))$ $(i \in [m])$.
    **end for**
  **Output:** $\{\theta^{(m)}(k)\}_{k=0}^{K-1}$

---

# B   Q-Learning and Policy Improvement

In this section, we extend our analysis of TD to Q-learning and soft Q-learning for policy improvement. In §B.1, we introduce Q-learning and its mean-field limit. In §B.2, we establish the global optimality and convergence of Q-learning. In §B.3, we further extend our analysis to soft Q-learning, which is equivalent to policy gradient.

## B.1   Q-Learning

Q-learning aims to solve the following projected Bellman optimality equation,

$$Q = \Pi_{\mathcal{F}} \mathcal{T}^* Q. \tag{B.1}$$

Here $\mathcal{T}^*$ is the Bellman optimality operator, which is defined as follows,

$$\mathcal{T}^* Q(s, a) = \mathbb{E}\big[r + \gamma \cdot \max_{\underline{a} \in \mathcal{A}} Q(s', \underline{a}) \,\big|\, r \sim R(\cdot \,|\, s, a), s' \sim P(\cdot \,|\, s, a)\big].$$

When $\Pi_{\mathcal{F}}$ is the identity mapping, the fixed point solution to (B.1) is the Q-function $Q^{\pi^*}$ of the optimal policy $\pi^*$, which maximizes the expected total reward $J(\pi)$ defined in (2.1) [65]. We consider the parameterization of the Q-function in (3.1) and update the parameter $\theta^{(m)}$ as follows,

$$\theta_i(k + 1) \tag{B.2}$$
$$= \theta_i(k) - \eta\epsilon \cdot \alpha \cdot \Big(\widehat{Q}\big(s_k, a_k; \theta^{(m)}(k)\big) - r_k - \gamma \cdot \max_{\underline{a} \in \mathcal{A}} \widehat{Q}\big(s'_k, \underline{a}; \theta^{(m)}(k)\big)\Big) \cdot \nabla_\theta \sigma\big(s_k, a_k; \theta_i(k)\big),$$

where $i \in [m]$, $(s_k, a_k)$ is sampled from the stationary distribution $\mathcal{D}_{\mathrm{E}} \in \mathscr{P}(\mathcal{S} \times \mathcal{A})$ of an exploration policy $\pi_{\mathrm{E}}$, $r_k \sim R(\cdot \,|\, s_k, a_k)$ is the reward, and $s'_k \sim P(\cdot \,|\, s_k, a_k)$ is the subsequent state. For notational simplicity, we denote by $\widetilde{\mathcal{D}}_{\mathrm{E}} \in \mathscr{P}(\mathcal{S} \times \mathcal{A} \times \mathbb{R} \times \mathcal{S})$ the distribution of $(s_k, a_k, r_k, s'_k)$. For an initial distribution $\nu_0 \in \mathscr{P}(\mathbb{R}^D)$, we initialize $\{\theta_i\}_{i=1}^m$ as $\theta_i \overset{\text{i.i.d.}}{\sim} \rho_0$ $(i \in [m])$. See Algorithm 2 for a detailed description.

**Mean-Field Limit.** Corresponding to $\epsilon \to 0^+$ and $m \to \infty$, the mean-field limit of the Q-learning dynamics in (B.2) is characterized by the following PDE with $\nu_0$ as the initial distribution,

$$\partial_t \nu_t = -\eta \cdot \mathrm{div}\big(\nu_t \cdot h(\cdot; \nu_t)\big). \tag{B.3}$$

Here $h(\cdot; \nu_t) : \mathbb{R}^D \to \mathbb{R}^D$ is a vector field, which is defined as follows,

$$h(\theta; \nu) = -\alpha \cdot \mathbb{E}_{(s,a,r,s') \sim \widetilde{\mathcal{D}}_{\mathrm{E}}}\Big[\big(Q(s, a; \nu) - r - \gamma \cdot \max_{\underline{a} \in \mathcal{A}} Q(s', \underline{a}; \nu)\big) \cdot \nabla_\theta \sigma(s, a; \theta)\Big]. \tag{B.4}$$

In parallel to Proposition 3.1, the empirical distribution $\widehat{\nu}_k^{(m)} = m^{-1} \cdot \sum_{i=1}^m \delta_{\theta_i(k)}$ weakly converges to $\nu_{k\epsilon}$ as $\epsilon \to 0^+$ and $m \to \infty$.

---

**Algorithm 2** Q-Learning with Two-Layer Neural Network for Policy Improvement

---

**Initialization.** $\theta_i(0) \overset{\text{i.i.d.}}{\sim} \nu_0$ ($i \in [m]$), number of iterations $K = \lfloor T/\epsilon \rfloor$, and exploration policy $\pi_{\mathrm{E}}$.

**for** $k = 0, \ldots, K-1$ **do**

    Sample the state-action pair $(s, a)$ from the stationary distribution $\mathcal{D}_{\mathrm{E}}$ of $\pi_{\mathrm{E}}$, receive the reward $r$, and obtain the subsequent state $s'$.

    Calculate the Bellman residual $\delta = \widehat{Q}(x; \theta^{(m)}(k)) - r - \gamma \cdot \widehat{Q}(x'; \theta^{(m)}(k))$, where $x = (s, a)$ and $x' = (s', \mathrm{argmax}_{\underline{a} \in \mathcal{A}} \widehat{Q}(s', \underline{a}; \theta^{(m)}(k)))$.

    Perform the Q-learning update $\theta_i(k+1) \leftarrow \theta_i(k) - \eta \epsilon \cdot \alpha \cdot \delta \cdot \nabla_\theta \sigma(x; \theta_i(k))$ ($i \in [m]$).

**end for**

**Output:** $\{\theta^{(m)}(k)\}_{k=0}^{K-1}$

---

## B.2 Global Optimality and Convergence of Q-Learning

The $\max$ operator in the Bellman optimality operator $\mathcal{T}^*$ makes the analysis of Q-learning more challenging than that of TD. Correspondingly, we lay out an extra regularity condition on the exploration policy $\pi_{\mathrm{E}}$. Recall that the function class $\mathcal{F}$ is defined in (4.3).

**Assumption B.1.** We assume for an absolute constant $\kappa > 0$ and any $Q^1, Q^2 \in \mathcal{F}$ that

$$\mathbb{E}_{(s,a) \sim \mathcal{D}_{\mathrm{E}}}\left[\left(Q^1(s,a) - Q^2(s,a)\right)^2\right] \geq (\gamma + \kappa)^2 \cdot \mathbb{E}_{(s,a) \sim \mathcal{D}_{\mathrm{E}}}\left[\left(\max_{\underline{a} \in \mathcal{A}} Q^1(s, \underline{a}) - \max_{\underline{a} \in \mathcal{A}} Q^2(s, \underline{a})\right)^2\right].$$

Although Assumption B.1 is strong, we are not aware of any weaker regularity condition in the literature, even in the linear setting [25, 55, 78] and the NTK regime [21]. Let the initial distribution $\nu_0$ be the standard Gaussian distribution $N(0, I_D)$. In parallel to Theorem 4.3, we establish the following theorem, which characterizes the global optimality and convergence of Q-learning. Recall that we write $\mathcal{X} = \mathcal{S} \times \mathcal{A}$ and $x = (s, a) \in \mathcal{X}$. Also, $\nu_t$ is the PDE solution in (B.3), while $\theta^{(m)}(k)$ is the Q-learning dynamics in (B.2).

**Theorem B.2.** There exists a unique fixed point solution to the projected Bellman optimality equation $Q = \Pi_{\mathcal{F}} \mathcal{T}^* Q$, which takes the form of $Q^\dagger(x) = \int \sigma(x; \theta) \, \mathrm{d}\bar{\nu}(\theta)$. We assume that $D_{\chi^2}(\bar{\nu} \,\|\, \nu_0) < \infty$ and $\bar{\nu}(\theta) > 0$ for any $\theta \in \mathbb{R}^D$. Under Assumptions 4.1, 4.2, and B.1, it holds for $\eta = \alpha^{-2}$ that

$$\inf_{t \in [0,T]} \mathbb{E}_{x \sim \mathcal{D}_{\mathrm{E}}}\left[\left(Q(x; \nu_t) - Q^\dagger(x)\right)^2\right] \leq \frac{(\kappa + \gamma) \cdot D_{\chi^2}(\bar{\nu} \,\|\, \nu_0)}{2\kappa \cdot T} + \frac{(\kappa + \gamma) \cdot C_*}{\kappa \cdot \alpha}, \tag{B.5}$$

where $C_* > 0$ is a constant depending on $D_{\chi^2}(\bar{\nu} \,\|\, \nu_0)$, $B_1$, $B_2$, and $B_r$. Moreover, it holds with probability at least $1 - \delta$ that

$$\min_{\substack{k \leq T/\epsilon \\ (k \in \mathbb{N})}} \mathbb{E}_{x \sim \mathcal{D}_{\mathrm{E}}}\left[\left(\widehat{Q}(x; \theta^{(m)}(k)) - Q^\dagger(x)\right)^2\right]$$

$$\leq \frac{(\kappa + \gamma) \cdot D_{\chi^2}(\bar{\nu} \,\|\, \nu_0)}{2\kappa \cdot T} + \frac{(\kappa + \gamma) \cdot C_*}{\kappa \cdot \alpha} + \Delta(\epsilon, m, \delta, T), \tag{B.6}$$

where $\Delta(\epsilon, m, \delta, T) > 0$ is an error term such that

$$\lim_{m \to \infty} \lim_{\epsilon \to 0^+} \Delta(\epsilon, m, \delta, T) = 0.$$

*Proof.* See §B.4 for a detailed proof. $\qquad\square$

Theorem B.2 proves that the optimality gap $\mathbb{E}_{x \sim \mathcal{D}_{\mathrm{E}}}[(Q(x; \nu_t) - Q^\dagger(x))^2]$ decays to zero at a sublinear rate up to the error of $O(\alpha^{-1})$, where $\alpha > 0$ is the scaling parameter in (3.1). In parallel to Theorem 4.3, varying $\alpha$ leads to a tradeoff between such an error of $O(\alpha^{-1})$ and the deviation of $\nu_t$ from $\nu_0$. Moreover, based on the counterparts of Proposition 3.1 and Lemma D.6, Theorem B.2 gives the global optimality and convergence of the Q-learning dynamics $\theta^{(m)}(k)$ in (B.2), which is in parallel to Corollary 4.4.

## B.3  Soft Q-Learning

In this section, we generalize Theorem B.2 to soft Q-learning. To introduce soft Q-learning, we first define the soft Bellman optimality operator as follows,

$$\mathcal{T}_\beta Q(s,a) = \mathbb{E}\big[r + \gamma \cdot \mathop{\mathrm{softmax}}_{\underline{a}\in\mathcal{A}}{}^\beta Q(s',\underline{a}) \,\big|\, r \sim R(\cdot\,|\,s,a), s' \sim P(\cdot\,|\,s,a)\big],$$

where the softmax operator is defined as follows,

$$\mathop{\mathrm{softmax}}_{\underline{a}\in\mathcal{A}}{}^\beta Q(s,\underline{a}) = \beta \cdot \log \mathbb{E}_{\underline{a}\sim\bar\pi(\cdot\,|\,s)}\Big[\exp\big(\beta^{-1}\cdot Q(s,\underline{a})\big)\Big].$$

Here $\bar\pi(\cdot\,|\,s)$ is the uniform policy. Soft Q-learning aims to find the fixed point solution to the projected soft Bellman optimality equation $Q = \Pi_\mathcal{F}\mathcal{T}_\beta Q$. In parallel to the Q-learning dynamics in (B.2), we consider the following soft Q-learning dynamics,

$$\theta_i(k+1) \tag{B.7}$$
$$= \theta_i(k) - \eta\epsilon \cdot \alpha \cdot \Big(\widehat{Q}\big(s_k, a_k; \theta^{(m)}(k)\big) - r_k - \gamma \cdot \mathop{\mathrm{softmax}}_{\underline{a}\in\mathcal{A}}{}^\beta \widehat{Q}\big(s'_k, \underline{a}; \theta^{(m)}(k)\big)\Big) \cdot \nabla_\theta \sigma\big(s_k, a_k; \theta_i(k)\big),$$

whose mean-field limit is characterized by the following PDE,

$$\partial_t \nu_t = -\eta \cdot \mathrm{div}\big(\nu_t \cdot h(\cdot; \nu_t)\big). \tag{B.8}$$

In parallel to (B.4), $h(\cdot; \nu_t) : \mathbb{R}^D \to \mathbb{R}^D$ is a vector field, which is defined as follows,

$$h(\theta; \nu) = -\alpha \cdot \mathbb{E}_{(s,a,r,s')\sim\widetilde{\mathcal{D}}_\mathrm{E}}\Big[\big(Q(s,a;\nu) - r - \gamma \cdot \mathop{\mathrm{softmax}}_{\underline{a}\in\mathcal{A}}{}^\beta Q(s',\underline{a};\nu)\big) \cdot \nabla_\theta \sigma(s,a;\theta)\Big].$$

In parallel to Assumption B.1, we lay out the following regularity condition.

**Assumption B.3.** We assume for an absolute constant $\kappa > 0$ and any $\nu^1, \nu^2 \in \mathscr{P}(\mathbb{R}^D)$ that

$$\mathbb{E}_{(s,a)\sim\mathcal{D}_\mathrm{E}}\Big[\big(Q(s,a;\nu^1) - Q(s,a;\nu^2)\big)^2\Big]$$
$$\geq (\gamma+\kappa)^2 \cdot \mathbb{E}_{(s,a)\sim\mathcal{D}_\mathrm{E}}\Big[\big(\mathop{\mathrm{softmax}}_{\underline{a}\in\mathcal{A}}{}^\beta Q(s,\underline{a};\nu^1) - \mathop{\mathrm{softmax}}_{\underline{a}\in\mathcal{A}}{}^\beta Q(s,\underline{a};\nu^2)\big)^2\Big].$$

The following proposition parallels Theorem B.2, which characterizes the global optimality and convergence of soft Q-learning. Recall that $\nu_t$ is the PDE solution in (B.8) and $\theta^{(m)}(k)$ is the soft Q-learning dynamics in (B.7).

**Proposition B.4.** There exists a unique fixed point solution to the projected soft Bellman optimality equation $Q = \Pi_\mathcal{F}\mathcal{T}_\beta Q$, which takes the form of $Q^\ddagger(x) = \int \sigma(x;\theta)\,\mathrm{d}\underline{\nu}(\theta)$. We assume that $D_{\chi^2}(\underline{\nu}\,\|\,\nu_0) < \infty$ and $\underline{\nu}(\theta) > 0$ for any $\theta \in \mathbb{R}^D$. Under Assumptions 4.1, 4.2, and B.3, it holds for $\eta = \alpha^{-2}$ that

$$\inf_{t\in[0,T]} \mathbb{E}_{x\sim\mathcal{D}_\mathrm{E}}\Big[\big(Q(x;\nu_t) - Q^\ddagger(x)\big)^2\Big] \leq \frac{(\kappa+\gamma)\cdot D_{\chi^2}(\underline{\nu}\,\|\,\nu_0)}{2\kappa\cdot T} + \frac{(\kappa+\gamma)\cdot C_*}{\kappa\cdot\alpha},$$

where $C_* > 0$ is a constant depending on $D_{\chi^2}(\underline{\nu}\,\|\,\nu_0)$, $B_1$, $B_2$, and $B_r$. Moreover, it holds with probability at least $1 - \delta$ that

$$\min_{\substack{k\leq T/\epsilon \\ (k\in\mathbb{N})}} \mathbb{E}_{x\sim\mathcal{D}_\mathrm{E}}\Big[\big(\widehat{Q}\big(x;\theta^{(m)}(k)\big) - Q^\ddagger(x)\big)^2\Big] \leq \frac{(\kappa+\gamma)\cdot D_{\chi^2}(\underline{\nu}\,\|\,\nu_0)}{2\kappa\cdot T} + \frac{(\kappa+\gamma)\cdot C_*}{\kappa\cdot\alpha} + \Delta(\epsilon, m, \delta, T),$$

where $\Delta(\epsilon, m, \delta, T) > 0$ is an error term such that

$$\lim_{m\to\infty}\lim_{\epsilon\to 0^+} \Delta(\epsilon, m, \delta, T) = 0.$$

*Proof.* Replacing the max operator by the softmax operator in the proof of Theorem B.2 in §B.4 implies Proposition B.4. $\qquad\square$

Moreover, soft Q-learning is equivalent to a variant of policy gradient [37, 57, 58, 61]. Hence, Proposition B.4 also characterizes the global optimality and convergence of such a variant of policy gradient.

## B.4 Proof of Theorem B.2

For notational simplicity, we denote by $\mathbb{E}_{\mathcal{D}_{\mathrm{E}}}$ the expectation with respect to $x \sim \mathcal{D}_{\mathrm{E}}$ and $\mathbb{E}_{\widetilde{\mathcal{D}}_{\mathrm{E}}}$ the expectation with respect to $(x, r, x') \sim \widetilde{\mathcal{D}}_{\mathrm{E}}$.

*Proof.* In parallel to the proof of Lemma 5.1 in §C.1, to establish the existence and uniqueness of the fixed point solution to the projected Bellman optimality equation $Q = \Pi_{\mathcal{F}} \mathcal{T}^* Q$, it suffices to show that $\Pi_{\mathcal{F}} \mathcal{T}^* : \mathcal{F} \to \mathcal{F}$ is a contraction mapping. In particular, it holds for any $Q^1, Q^2 \in \mathcal{F}$ that

$$\|\Pi_{\mathcal{F}} \mathcal{T}^* Q^1 - \Pi_{\mathcal{F}} \mathcal{T}^* Q^2\|^2_{\mathcal{L}_2(\mathcal{D}_{\mathrm{E}})} \leq \gamma^2 \cdot \mathbb{E}_{\widetilde{\mathcal{D}}_{\mathrm{E}}}\left[\left(\max_{\underline{a} \in \mathcal{A}} Q^1(s', \underline{a}) - \max_{\underline{a} \in \mathcal{A}} Q^2(s', \underline{a})\right)^2\right]$$

$$= \gamma^2 \cdot \mathbb{E}_{\mathcal{D}_{\mathrm{E}}}\left[\left(\max_{\underline{a} \in \mathcal{A}} Q^1(s, \underline{a}) - \max_{\underline{a} \in \mathcal{A}} Q^2(s, \underline{a})\right)^2\right]$$

$$\leq \frac{\gamma^2}{(\gamma + \kappa)^2} \cdot \mathbb{E}_{\mathcal{D}_{\mathrm{E}}}\left[\left(Q^1(s, a) - Q^2(s, a)\right)^2\right],$$

where the equality follows from the fact that $\mathcal{D}_{\mathrm{E}}$ is the stationary distribution and the last inequality follows from Assumption B.1. Thus, $\Pi_{\mathcal{F}} \mathcal{T}^* : \mathcal{F} \to \mathcal{F}$ is a contraction mapping. Following from the Banach fixed point theorem [28], there exists a unique fixed point solution $Q^\dagger \in \mathcal{F}$ to the projected Bellman optimality equation $Q = \Pi_{\mathcal{F}} \mathcal{T}^* Q$. Moreover, in parallel to the proof of Lemma 5.1 in §C.1, there exists $\nu^\dagger \in \mathscr{P}_2(\mathbb{R}^D)$ such that $Q(x; \nu^\dagger) = Q^\dagger(x)$, $h(x; \nu^\dagger) = 0$, and $\mathcal{W}_2(\nu^\dagger, \nu_0) \leq \alpha^{-1} \cdot \bar{D}$, where $\bar{D} = D_{\chi^2}(\bar{\nu} \| \nu_0)^{1/2}$.

For notational simplicity, we define $Q^{\mathcal{A}}(x) = \max_{\underline{a} \in \mathcal{A}} Q(s, \underline{a})$. In parallel to (C.13) in the proof of Lemma 5.2 in §C.2, we have that

$$\frac{\mathrm{d}}{\mathrm{d}t} \frac{\mathcal{W}_2(\nu_t, \nu^\dagger)^2}{2} = \eta \cdot \underbrace{\int_0^1 \langle \partial_s h(\cdot; \beta_s), v_s \rangle_{\beta_s} \mathrm{d}s}_{\text{(i)}} + \eta \cdot \underbrace{\int_0^1 \int \langle h(\theta; \beta_s), \partial_s(v_s \cdot \beta_s)(\theta) \rangle \mathrm{d}\theta\, \mathrm{d}s}_{\text{(ii)}}, \quad \text{(B.9)}$$

where $\beta : [0, 1] \to \mathscr{P}_2(\mathbb{R}^D)$ is the geodesic connecting $\nu_t$ and $\nu^\dagger$ with $\partial_s \beta_s = -\operatorname{div}(\beta_s \cdot v_s)$.

**Upper bounding term (i) of** (B.9). In parallel to (C.5) and (C.6) in the proof of Lemma C.1, we have that

$$\langle \partial_s h(\cdot; \beta_s), v_s \rangle_{\beta_s} = -\mathbb{E}_{\widetilde{\mathcal{D}}_{\mathrm{E}}}\left[\partial_s\big(Q(x; \beta_s) - \gamma \cdot Q^{\mathcal{A}}(x'; \beta_s)\big) \cdot \partial_s Q(x; \beta_s)\right] \quad \text{(B.10)}$$

$$\leq -\mathbb{E}_{\mathcal{D}_{\mathrm{E}}}\left[\big(\partial_s Q(x; \beta_s)\big)^2\right] + \gamma \cdot \mathbb{E}_{\mathcal{D}_{\mathrm{E}}}\left[\big(\partial_s Q(x; \beta_s)\big)^2\right]^{1/2} \cdot \mathbb{E}_{\mathcal{D}_{\mathrm{E}}}\left[\big(\partial_s Q^{\mathcal{A}}(x; \beta_s)\big)^2\right]^{1/2}.$$

For the second term on the right-hand side of (B.10), we have that

$$\mathbb{E}_{\mathcal{D}_{\mathrm{E}}}\left[\big(\partial_s Q^{\mathcal{A}}(x; \beta_s)\big)^2\right] = \lim_{u \to 0} \mathbb{E}_{\mathcal{D}_{\mathrm{E}}}\left[\left(u^{-1} \cdot \big(Q^{\mathcal{A}}(x; \beta_{s+u}) - Q^{\mathcal{A}}(x; \beta_s)\big)\right)^2\right]$$

$$\leq (\gamma + \kappa)^{-2} \cdot \lim_{u \to 0} u^{-2} \cdot \mathbb{E}_{\mathcal{D}_{\mathrm{E}}}\left[\big(Q(x; \beta_{s+u}) - Q(x; \beta_s)\big)^2\right]$$

$$= (\gamma + \kappa)^{-2} \cdot \mathbb{E}_{\mathcal{D}_{\mathrm{E}}}\left[\big(\partial_s Q(x; \beta_s)\big)^2\right], \quad \text{(B.11)}$$

where the inequality follows from Assumption B.1 and the fact that $Q(\cdot; \nu) \in \alpha \cdot \mathcal{F}$. Plugging (B.11) into (B.10), we have that

$$\langle \partial_s h(\cdot; \beta_s), v_s \rangle_{\beta_s} \leq -\frac{\kappa}{\gamma + \kappa} \cdot \mathbb{E}_{\mathcal{D}_{\mathrm{E}}}\left[\big(\partial_s Q(x; \beta_s)\big)^2\right],$$

which further implies that

$$\int_0^1 \langle \partial_s h(\cdot; \beta_s), v_s \rangle_{\beta_s} \mathrm{d}s \leq -\frac{\kappa}{\gamma + \kappa} \cdot \int_0^1 \mathbb{E}_{\mathcal{D}_{\mathrm{E}}}\left[\big(\partial_s Q(x; \beta_s)\big)^2\right] \mathrm{d}s$$

$$\leq -\frac{\kappa}{\gamma + \kappa} \cdot \mathbb{E}_{\mathcal{D}_{\mathrm{E}}}\left[\left(\int_0^1 \partial_s Q(x; \beta_s)\, \mathrm{d}s\right)^2\right]$$

$$= -\frac{\kappa}{\gamma + \kappa} \cdot \mathbb{E}_{\mathcal{D}_{\mathrm{E}}}\left[\big(Q(x; \nu_t) - Q(x; \nu^\dagger)\big)^2\right]. \quad \text{(B.12)}$$

**Upper bounding term (ii) of** (B.9). In parallel to the proof of Lemma C.2 in §C.2, noting that $|Q^{\mathcal{A}}(x;\nu)| \leq \sup_{x \in \mathcal{X}} |Q(x;\nu)|$ for any $\nu \in \mathscr{P}_2(\mathbb{R}^D)$, we have that

$$\big\|\nabla_\theta h(\theta;\nu_t)\big\|_{\mathrm{F}} \leq \alpha \cdot B_2 \cdot \big(2\alpha \cdot B_1 \cdot \mathcal{W}_2(\nu_t,\nu_0) + B_r\big).$$

In parallel to (C.15) and (C.16), we have that

$$\int_0^1 \int \Big| \big\langle h(\theta;\beta_s), \partial_s(v_s \cdot \beta_s)(\theta) \big\rangle \Big| \,\mathrm{d}\theta \,\mathrm{d}s \leq C_* \cdot \alpha^{-1}, \tag{B.13}$$

where $C_* > 0$ is a constant that depends on $\bar{D}$, $B_1$, $B_2$, and $B_r$.

Plugging (B.12) and (B.13) into (B.9), we have that

$$\frac{\mathrm{d}}{\mathrm{d}t} \frac{\mathcal{W}_2(\nu_t,\nu^\dagger)^2}{2} \leq -\frac{\eta \cdot \kappa}{\gamma + \kappa} \cdot \mathbb{E}_{\mathcal{D}_{\mathrm{E}}}\Big[ \big(Q(x;\nu_t) - Q(x;\nu^\dagger)\big)^2 \Big] + C_* \cdot \eta \cdot \alpha^{-1}.$$

Thus, in parallel to the proof of Theorem 4.3 in §5, we have that

$$\inf_{t \in [0,T]} \mathbb{E}_{\mathcal{D}}\Big[ \big(Q(x;\nu_t) - Q^\dagger(x)\big)^2 \Big] \leq \frac{(\kappa + \gamma) \cdot D_{\chi^2}(\bar{\nu} \,\|\, \nu_0)}{2\kappa \cdot T} + C_* \cdot \alpha^{-1} \cdot \frac{\kappa + \gamma}{\kappa},$$

which completes the proof of (B.5) in Theorem B.2. Meanwhile, in parallel to the proof of Lemma D.6 in §D.2, we upper bound the error of approximating $\widehat{\nu}_k$ by $\nu_{k\epsilon}$, which further implies (B.6) of Theorem B.2. $\qquad\square$

# C  Proofs of Supporting Lemmas

For notational simplicity, we denote by $\mathbb{E}_{\mathcal{D}}$ the expectation with respect to $x \sim \mathcal{D}$ and $\mathbb{E}_{\widetilde{\mathcal{D}}}$ the expectation with respect to $(x,r,x') \sim \widetilde{\mathcal{D}}$. Also, with a slight abuse of notations, we write $\theta^{(m)} = \{\theta_i\}_{i=1}^m$.

## C.1  Proof of Lemma 5.1

*Proof.* **Existence and uniqueness of** $Q^*$. To establish the existence of the fixed point solution $Q^*$ to the projected Bellman equation $Q = \Pi_{\mathcal{F}}\mathcal{T}^\pi Q$, it suffices to show that $\Pi_{\mathcal{F}}\mathcal{T}^\pi : \mathcal{F} \to \mathcal{F}$ is a contraction mapping. It holds for any $Q^1, Q^2 \in \mathcal{F}$ that

$$\|\Pi_{\mathcal{F}}\mathcal{T}^\pi Q^1 - \Pi_{\mathcal{F}}\mathcal{T}^\pi Q^2\|_{\mathcal{L}_2(\mathcal{D})}^2 \leq \gamma^2 \cdot \mathbb{E}_{\widetilde{\mathcal{D}}}\Big[ \big(Q^1(x') - Q^2(x')\big)^2 \Big]$$
$$= \gamma^2 \cdot \big\|Q^1 - Q^2\big\|_{\mathcal{L}_2(\mathcal{D})}^2,$$

where the last equality follows from the fact that $\mathcal{D}$ is the stationary distribution. Thus, $\Pi_{\mathcal{F}}\mathcal{T}^\pi : \mathcal{F} \to \mathcal{F}$ is a contraction mapping. Note that $\mathcal{F}$ is complete. Following from the Banach fixed point theorem [28], there exists a unique $Q^* \in \mathcal{F}$ that solves the projected Bellman equation $Q = \Pi_{\mathcal{F}}\mathcal{T}^\pi Q$. Moreover, by the definition of $\mathcal{F}$ in (4.3), there exists $\bar{\rho} \in \mathscr{P}_2(\mathbb{R}^D)$ such that

$$Q^*(x) = \int \sigma(x;\theta) \,\mathrm{d}\bar{\rho}(\theta).$$

**Proof of (i) in Lemma 5.1.** We define

$$\rho^* = \rho_0 + \alpha^{-1} \cdot (\bar{\rho} - \rho_0). \tag{C.1}$$

By the definition of $Q(\cdot;\rho)$ in (3.2) and the fact that $Q(x;\rho_0) = 0$, we have that $Q(x;\rho^*) = Q^*(x)$, which completes the proof of (i) in Lemma 5.1.

**Proof of (ii) in Lemma 5.1.** For (ii) of Lemma 5.1, note that $Q(\cdot;\rho^*) = \Pi_{\mathcal{F}}\mathcal{T}^\pi Q(\cdot;\rho^*)$. Thus, we have that

$$\big\langle Q(\cdot;\rho^*) - \mathcal{T}^\pi Q(\cdot;\rho^*), f(\cdot) - Q(\cdot;\rho^*) \big\rangle_{\mathcal{D}} \geq 0, \quad \forall f \in \mathcal{F},$$

which further implies that

$$\mathbb{E}_{\widetilde{\mathcal{D}}}\Big[\big(Q(x;\rho^*) - r - \gamma \cdot Q(x';\rho^*)\big) \cdot \int \sigma(x;\theta)\,\mathrm{d}(\rho - \bar{\rho})(\theta)\Big] \geq 0, \quad \forall \rho \in \mathscr{P}_2(\mathbb{R}^D). \tag{C.2}$$

Let $\rho = (\mathrm{id} + h \cdot v)_{\sharp} \bar{\rho}$ for a sufficiently small scaling parameter $h \in \mathbb{R}_+$ and any Lipschitz-continuous mapping $v : \mathbb{R}^D \to \mathbb{R}^D$. Then, following from (C.2), we have that

$$\int \mathbb{E}_{\widetilde{\mathcal{D}}}\Big[\big(Q(x;\rho^*) - r - \gamma \cdot Q(x';\rho^*)\big) \cdot \big(\sigma\big(x;\theta + h \cdot v(\theta)\big) - \sigma(x;\theta)\big)\Big]\,\mathrm{d}\bar{\rho}(\theta) \geq 0 \tag{C.3}$$

for any $v : \mathbb{R}^D \to \mathbb{R}^D$. Dividing the both sides of (C.3) by $h$ and letting $h \to 0^+$, we have for any $v : \mathbb{R}^D \to \mathbb{R}^D$ that

$$0 \leq \int \mathbb{E}_{\widetilde{\mathcal{D}}}\Big[\big(Q(x;\rho^*) - r - \gamma \cdot Q(x';\rho^*)\big) \cdot \big\langle \nabla_\theta \sigma(x;\theta), v(\theta)\big\rangle\Big]\,\mathrm{d}\bar{\rho}(\theta)$$

$$= -\alpha^{-1} \cdot \int \big\langle g(\theta;\rho^*), v(\theta)\big\rangle\,\mathrm{d}\bar{\rho}(\theta),$$

where the equality follows from the definition of $g$ in (3.5). Thus, we have that $g(\theta;\rho^*) = 0$ for $\bar{\rho}$-a.e., which completes the proof of (ii) in Lemma 5.1.

**Proof of (iii) in Lemma 5.1.** Following from the definition of $\rho^*$ in (C.1), we have that

$$D_{\chi^2}(\rho^* \,\|\, \rho_0)$$
$$= \int \bigg(\frac{\rho^*(\theta)}{\rho_0(\theta)} - 1\bigg)^2 \,\mathrm{d}\rho_0(\theta) = \int \bigg(\frac{(1 - \alpha^{-1}) \cdot \rho_0(\theta) + \alpha^{-1} \cdot \bar{\rho}(\theta)}{\rho_0(\theta)} - 1\bigg)^2 \,\mathrm{d}\rho_0(\theta) = \alpha^{-2} \cdot \bar{D}^2,$$

where $\bar{D} = D_{\chi^2}(\bar{\rho} \,\|\, \rho_0)^{1/2}$. By Lemma E.3, we have that

$$\mathcal{W}_2(\rho^*, \rho_0) \leq D_{\mathrm{KL}}(\rho^* \,\|\, \rho_0)^{1/2} \leq D_{\chi^2}(\rho^* \,\|\, \rho_0)^{1/2} \leq \alpha^{-1} \cdot \bar{D},$$

which completes the proof of (iii) in Lemma 5.1. $\qquad\square$

## C.2   Proof of Lemma 5.2

We first introduce the following lemmas. The first lemma establishes the one-point monotonicity of $g(\cdot;\beta_t)$ along a curve $\beta : [0,1] \to \mathscr{P}_2(\mathbb{R}^D)$ on the Wasserstein space.

**Lemma C.1.** Let $\beta : [0,1] \to \mathscr{P}_2(\mathbb{R}^D)$ be a curve such that $\partial_t \beta_t = -\operatorname{div}(\beta_t \cdot v_t)$ for a vector field $v$. We have that

$$\big\langle \partial_t g(\cdot;\beta_t), v_t\big\rangle_{\beta_t} \leq -(1 - \gamma) \cdot \mathbb{E}_{\mathcal{D}}\Big[\big(\partial_t Q(x;\beta_t)\big)^2\Big].$$

Furthermore, we have that

$$\int_0^1 \big\langle \partial_s g(\cdot;\beta_s), v_s\big\rangle_{\beta_s}\,\mathrm{d}s \leq -(1 - \gamma) \cdot \mathbb{E}_{\mathcal{D}}\Big[\big(Q(x;\beta_0) - Q(x;\beta_1)\big)^2\Big]. \tag{C.4}$$

*Proof.* Following from the definition of $g$ in (3.5), we have that

$$\partial_t g(\theta;\beta_t) = -\alpha \cdot \mathbb{E}_{\widetilde{\mathcal{D}}}\Big[\partial_t\big(Q(x;\beta_t) - \gamma \cdot Q(x';\beta_t)\big) \cdot \nabla_\theta \sigma(x;\theta)\Big].$$

Thus, following from integration by parts and the continuity equation $\partial_t \beta_t = -\operatorname{div}(\beta_t \cdot v_t)$, we have that

$$\big\langle \partial_t g(\cdot;\beta_t), v_t\big\rangle_{\beta_t} = -\int \bigg\langle \alpha \cdot \mathbb{E}_{\widetilde{\mathcal{D}}}\Big[\partial_t\big(Q(x;\beta_t) - \gamma \cdot Q(x';\beta_t)\big) \cdot \nabla_\theta \sigma(x;\theta)\Big], v_t(\theta) \cdot \beta_t(\theta)\bigg\rangle\,\mathrm{d}\theta$$

$$= -\int \alpha \cdot \mathbb{E}_{\widetilde{\mathcal{D}}}\Big[\partial_t\big(Q(x;\beta_t) - \gamma \cdot Q(x';\beta_t)\big) \cdot \sigma(x;\theta)\Big] \cdot \partial_t \beta_t(\theta)\,\mathrm{d}\theta$$

$$= -\mathbb{E}_{\widetilde{\mathcal{D}}}\Big[\partial_t\big(Q(x;\beta_t) - \gamma \cdot Q(x';\beta_t)\big) \cdot \partial_t Q(x;\beta_t)\Big], \tag{C.5}$$

where the last equality follows from the definition of $Q$ in (3.2). Applying the Cauchy-Schwartz inequality to (C.5), we have that

$$
\begin{aligned}
\big\langle \partial_t g(\cdot; \beta_t), v_t \big\rangle_{\beta_t} &= -\mathbb{E}_{\widetilde{\mathcal{D}}}\Big[\big(\partial_t Q(x; \beta_t)\big)^2\Big] + \gamma \cdot \mathbb{E}_{\widetilde{\mathcal{D}}}\big[\partial_t Q(x'; \beta_t) \cdot \partial_t Q(x; \beta_t)\big] \\
&\leq -\mathbb{E}_{\widetilde{\mathcal{D}}}\Big[\big(\partial_t Q(x; \beta_t)\big)^2\Big] + \gamma \cdot \mathbb{E}_{\widetilde{\mathcal{D}}}\Big[\big(\partial_t Q(x; \beta_t)\big)^2\Big]^{1/2} \cdot \mathbb{E}_{\widetilde{\mathcal{D}}}\Big[\big(\partial_t Q(x'; \beta_t)\big)^2\Big]^{1/2} \\
&= -(1-\gamma) \cdot \mathbb{E}_{\mathcal{D}}\Big[\big(\partial_t Q(x; \beta_t)\big)^2\Big],
\end{aligned}
\tag{C.6}
$$

where the last equality follows from the fact that the marginal distributions of $\widetilde{\mathcal{D}}$ with respect to $x$ and $x'$ are $\mathcal{D}$, since $\mathcal{D}$ is the stationary distribution. Furthermore, we have that

$$
\begin{aligned}
\int_0^1 \big\langle \partial_s g(\cdot; \beta_s), v_s \big\rangle_{\beta_s} \,\mathrm{d}s &\leq -(1-\gamma) \cdot \int_0^1 \mathbb{E}_{\mathcal{D}}\Big[\big(\partial_s Q(x; \beta_s)\big)^2\Big]\,\mathrm{d}s \\
&\leq -(1-\gamma) \cdot \mathbb{E}_{\mathcal{D}}\bigg[\Big(\int_0^1 \partial_s Q(x; \beta_s)\,\mathrm{d}s\Big)^2\bigg] \\
&= -(1-\gamma) \cdot \mathbb{E}_{\mathcal{D}}\Big[\big(Q(x; \beta_1) - Q(x; \beta_0)\big)^2\Big],
\end{aligned}
$$

which completes the proof of Lemma C.1. $\qquad\square$

The following lemma upper bounds the norms of $Q$ and $\nabla_\theta g$.

**Lemma C.2.** Under Assumptions 4.1 and 4.2, it holds for any $\rho \in \mathscr{P}_2(\mathbb{R}^D)$ that

$$
\sup_{x \in \mathcal{X}} \big|Q(x; \rho)\big| \leq \alpha \cdot \min\big\{B_1 \cdot \mathcal{W}_2(\rho, \rho_0),\, B_0\big\},
\tag{C.7}
$$

$$
\sup_{\theta \in \mathbb{R}^D} \big\|\nabla_\theta g(\theta; \rho)\big\|_{\mathrm{F}} \leq \alpha \cdot B_2 \cdot \min\big\{2\alpha \cdot B_1 \cdot \mathcal{W}_2(\rho, \rho_0) + B_r,\, 2\alpha \cdot B_0 + B_r\big\}.
\tag{C.8}
$$

*Proof.* We introduce the Wasserstein-1 distance, which is defined as

$$
\mathcal{W}_1(\mu^1, \mu^2) = \inf\Big\{\mathbb{E}\big[\|X - Y\|\big] \,\Big|\, \mathrm{law}(X) = \mu^1, \mathrm{law}(Y) = \mu^2\Big\}
$$

for any $\mu^1, \mu^2 \in \mathscr{P}(\mathbb{R}^D)$ with finite first moments. Thus, we have that $\mathcal{W}_1(\mu^1, \mu^2) \leq \mathcal{W}_2(\mu^1, \mu^2)$. The Wasserstein-1 distance has the following dual representation [5],

$$
\mathcal{W}_1(\mu^1, \mu^2) = \sup\bigg\{\int f(x)\,\mathrm{d}(\mu^1 - \mu^2)(x) \,\bigg|\, \text{continuous } f: \mathbb{R}^D \to \mathbb{R}, \mathrm{Lip}(f) \leq 1\bigg\}.
\tag{C.9}
$$

Following from Assumptions 4.1 and 4.2, we have that $\|\nabla_\theta \sigma(x; \theta)\| \leq B_1$ for any $x \in \mathcal{X}$ and $\theta \in \mathbb{R}^D$, which implies that $\mathrm{Lip}(\sigma(x; \cdot)/B_1) \leq 1$ for any $x \in \mathcal{X}$. Note that $Q(x; \rho_0) = 0$ for any $x \in \mathcal{X}$. Thus, by (C.9) we have for any $\rho \in \mathscr{P}_2(\mathbb{R}^D)$ and $x \in \mathcal{X}$ that

$$
\big|Q(x; \rho)\big| = \alpha \cdot \bigg|\int \sigma(x; \theta) \cdot \mathrm{d}(\rho - \rho_0)(\theta)\bigg| \leq \alpha \cdot B_1 \cdot \mathcal{W}_1(\rho, \rho_0) \leq \alpha \cdot B_1 \cdot \mathcal{W}_2(\rho, \rho_0).
\tag{C.10}
$$

Meanwhile, following from Assumptions 4.1 and 4.2, we have for any $x \in \mathcal{X}$ and $\rho \in \mathscr{P}_2(\mathbb{R}^D)$ that

$$
\big|Q(x; \rho)\big| = \alpha \cdot \bigg|\int \sigma(x; \theta)\,\mathrm{d}\rho(\theta)\bigg| \leq \alpha \cdot B_0.
\tag{C.11}
$$

Combining (C.10) and (C.11), we have for any $\rho \in \mathscr{P}_2(\mathbb{R}^D)$ that

$$
\sup_{x \in \mathcal{X}} \big|Q(x; \rho)\big| \leq \alpha \cdot \min\big\{B_1 \cdot \mathcal{W}_2(\rho, \rho_0),\, B_0\big\},
\tag{C.12}
$$

which completes the proof of (C.7) in Lemma C.2. Following from the definition of $g$ in (3.5), we have for any $x \in \mathcal{X}$ and $\rho \in \mathscr{P}_2(\mathbb{R}^D)$ that

$$
\begin{aligned}
\big\|\nabla_\theta g(\theta; \rho)\big\|_{\mathrm{F}} &\leq \alpha \cdot \mathbb{E}_{\widetilde{\mathcal{D}}}\Big[\big|Q(x; \rho) - r - \gamma \cdot Q(x'; \rho)\big| \cdot \big\|\nabla_{\theta\theta}^2 \sigma(x; \theta)\big\|_{\mathrm{F}}\Big] \\
&\leq \alpha \cdot \min\big\{2\alpha \cdot B_1 \cdot \mathcal{W}_2(\rho, \rho_0) + B_r,\, 2\alpha \cdot B_0 + B_r\big\} \cdot B_2.
\end{aligned}
$$

Here the last inequality follows from (C.12) and the fact that $\|\nabla_{\theta\theta}^2 \sigma(x; \theta)\|_{\mathrm{F}} \leq B_2$ for any $x \in \mathcal{X}$ and $\rho \in \mathscr{P}_2(\mathbb{R}^D)$, which follows from Assumptions 4.1 and 4.2. Thus, we complete the proof of Lemma C.2. $\qquad\square$

We are now ready to present the proof of Lemma 5.2.

*Proof.* Recall that $\rho_t$ is the PDE solution in (3.4), that is,

$$\partial_t \rho_t = -\eta \cdot \mathrm{div}\big(\rho_t \cdot g(\cdot; \rho_t)\big),$$

where

$$g(\theta; \rho) = -\alpha \cdot \mathbb{E}_{\widetilde{\mathcal{D}}}\Big[\big(Q(x; \rho) - r - \gamma \cdot Q(x'; \rho)\big) \cdot \nabla_\theta \sigma(x; \theta)\Big].$$

We fix a $t \in [0, T]$. We denote by $\beta : [0, 1] \to \mathscr{P}_2(\mathbb{R}^D)$ the geodesic connecting $\rho_t$ and $\rho^*$. Specifically, $\beta$ satisfies that $\beta'_s = -\mathrm{div}(\beta_s \cdot v_s)$ for a vector field $v$. Following from Lemma E.2, we have that

$$
\begin{aligned}
\frac{\mathrm{d}}{\mathrm{d}t} \frac{\mathcal{W}_2(\rho_t, \rho^*)^2}{2} &= -\eta \cdot \big\langle g(\cdot; \rho_t), v_0 \big\rangle_{\rho_t} \\
&= \eta \cdot \int_0^1 \partial_s \big\langle g(\cdot; \beta_s), v_s \big\rangle_{\beta_s} \, \mathrm{d}s - \eta \cdot \big\langle g(\cdot; \rho^*), v_1 \big\rangle_{\rho^*} \\
&= \eta \cdot \underbrace{\int_0^1 \big\langle \partial_s g(\cdot; \beta_s), v_s \big\rangle_{\beta_s} \, \mathrm{d}s}_{(i)} + \eta \cdot \underbrace{\int_0^1 \int \big\langle g(\theta; \beta_s), \partial_s (v_s \cdot \beta_s)(\theta) \big\rangle \, \mathrm{d}\theta \, \mathrm{d}s}_{(ii)},
\end{aligned}
$$

(C.13)

where the last equality follows from (ii) of Lemma 5.1.

For term (i) of (C.13), following from (C.4) of Lemma C.1, we have that

$$
\int_0^1 \big\langle \partial_s g(\cdot; \beta_s), v_s \big\rangle_{\beta_s} \, \mathrm{d}s \leq -(1 - \gamma) \cdot \mathbb{E}_{\mathcal{D}}\Big[\big(Q(x; \beta_0) - Q(x; \beta_1)\big)^2\Big]
$$

$$
= -(1 - \gamma) \cdot \mathbb{E}_{\mathcal{D}}\Big[\big(Q(x; \rho_t) - Q^*(x)\big)^2\Big]. \tag{C.14}
$$

For term (ii) of (C.14), we have that

$$
\int \Big| \big\langle g(\theta; \beta_s), \partial_s (v_s \cdot \beta_s)(\theta) \big\rangle \Big| \, \mathrm{d}\theta = \int \Big| \big\langle \nabla_\theta g(\theta; \beta_s), \beta_s(\theta) \cdot v_s(\theta) \otimes v_s(\theta) \big\rangle \Big| \, \mathrm{d}\theta
$$

$$
\leq \sup_{\theta \in \mathbb{R}^D} \big\| \nabla_\theta g(\theta; \beta_s) \big\|_{\mathrm{F}} \cdot \| v_s \|_{\beta_s}^2,
$$

where the equality follows from integration by parts and Lemma E.4. Since $\beta$ is the geodesic connecting $\rho_t$ and $\rho^*$, (2.7) implies that $\| v_s \|_{\beta_s}^2 = \mathcal{W}_2(\beta_0, \beta_1)^2 = \mathcal{W}_2(\rho_t, \rho^*)^2$ for any $s \in [0, 1]$. Applying (C.8) of Lemma C.2, we have that

$$
\int \Big| \big\langle g(\theta; \beta_s), \partial_s (v_s \cdot \beta_s)(\theta) \big\rangle \Big| \, \mathrm{d}\theta \leq \alpha \cdot B_2 \cdot \big(2\alpha \cdot B_1 \cdot \mathcal{W}_2(\rho_t, \rho_0) + B_r\big) \cdot \mathcal{W}_2(\rho_t, \rho^*)^2
$$

$$
\leq 4\alpha \cdot B_2 \cdot \big(6\alpha \cdot B_1 \cdot \mathcal{W}_2(\rho_0, \rho^*) + B_r\big) \cdot \mathcal{W}_2(\rho_0, \rho^*)^2, \tag{C.15}
$$

where the last inequality follows from the condition of Lemma 5.2 that $\mathcal{W}_2(\rho_t, \rho^*) \leq 2\mathcal{W}_2(\rho_0, \rho^*)$ and the fact that $\mathcal{W}_2(\rho_t, \rho_0) \leq \mathcal{W}_2(\rho_t, \rho^*) + \mathcal{W}_2(\rho_0, \rho^*)$. Then, applying (iii) of Lemma 5.1 to (C.15), we have that

$$
\int_0^1 \int \Big| \big\langle g(\theta; \beta_s), \partial_s (v_s \cdot \beta_s)(\theta) \big\rangle \Big| \, \mathrm{d}\theta \, \mathrm{d}s \leq 4\alpha^{-1} \cdot B_2 \cdot \bar{D}^2 \cdot (6B_1 \cdot \bar{D} + B_r)
$$

$$
= C_* \cdot \alpha^{-1}, \tag{C.16}
$$

where $C_* > 0$ is a constant depending on $\bar{D}$, $B_1$, $B_2$, and $B_r$.

Finally, plugging (C.14) and (C.16) into (C.13), we have that

$$
\frac{\mathrm{d}}{\mathrm{d}t} \frac{\mathcal{W}_2(\rho_t, \rho^*)^2}{2} \leq -(1 - \gamma) \cdot \eta \cdot \mathbb{E}_{\mathcal{D}}\Big[\big(Q(x; \rho_t) - Q^*(x)\big)^2\Big] + C_* \cdot \alpha^{-1} \cdot \eta,
$$

which completes the proof of Lemma 5.2. $\qquad\square$

# D    Mean-Field Limit of Neural Networks

In this section, we prove Proposition 3.1, whose formal version is presented as follows. Recall that $\rho_t$ is the PDE solution in (3.4) and $\widehat{\rho}_k = m^{-1} \cdot \sum_{i=1}^m \theta_i(k)$ is the empirical distribution of $\theta^{(m)}(k) = \{\theta_i(k)\}_{i=1}^m$. Note that we omit the dependence of $\widehat{\rho}_k$ on $m$ and $\epsilon$ for notational simplicity.

**Proposition D.1** (Formal Version of Proposition 3.1). Let $f : \mathbb{R}^D \to \mathbb{R}$ be any continuous function such that $\|f\|_\infty \le 1$ and $\mathrm{Lip}(f) \le 1$. Under Assumptions 4.1 and 4.2, it holds that

$$\sup_{\substack{k \le T/\epsilon \\ (k \in \mathbb{N})}} \left| \int f(\theta) \, \mathrm{d}\rho_{k\epsilon}(\theta) - \int f(\theta) \, \mathrm{d}\widehat{\rho}_k(\theta) \right|$$

$$\le B \cdot e^{BT} \cdot \left( \sqrt{\log(m/\delta)/m} + \sqrt{\epsilon \cdot \big( D + \log(m/\delta) \big)} \right)$$

with probability at least $1 - \delta$. Here $B$ is a constant that depends on $\alpha, \eta, \gamma, B_r$, and $B_j$ ($j \in \{0, 1, 2\}$).

The proof of Proposition D.1 is based on [6, 53, 54], which utilizes the propagation of chaos [66]. Recall that $g(\cdot; \rho)$ is a vector field defined as follows,

$$g(\theta; \rho) = -\alpha \cdot \mathbb{E}_{\widetilde{\mathcal{D}}} \Big[ \big( Q(x; \rho) - r - \gamma \cdot Q(x'; \rho) \big) \cdot \nabla_\theta \sigma(x; \theta) \Big].$$

Correspondingly, we define the finite-width and stochastic counterparts of $g(\theta; \rho)$ as follows,

$$\widehat{g}(\theta; \theta^{(m)}) = -\alpha \cdot \mathbb{E}_{\widetilde{\mathcal{D}}} \Big[ \big( \widehat{Q}(x; \theta^{(m)}) - r - \gamma \cdot \widehat{Q}(x'; \theta^{(m)}) \big) \cdot \nabla_\theta \sigma(x; \theta) \Big], \tag{D.1}$$

$$\widehat{G}_k(\theta; \theta^{(m)}) = -\alpha \cdot \big( \widehat{Q}(x_k; \theta^{(m)}) - r_k - \gamma \cdot \widehat{Q}(x'_k; \theta^{(m)}) \big) \cdot \nabla_\theta \sigma(x_k; \theta), \tag{D.2}$$

where $(x_k, r_k, x'_k) \sim \widetilde{\mathcal{D}}$. Following from [6, 53], we consider the following four dynamics.

- **Temporal-difference (TD).** We consider the following TD dynamics $\theta^{(m)}(k)$, where $k \in \mathbb{N}$, with $\theta_i(0) \overset{\text{i.i.d.}}{\sim} \rho_0$ ($i \in [m]$) as its initialization,

$$\theta_i(k+1) = \theta_i(k) - \eta\epsilon \cdot \alpha \cdot \Big( \widehat{Q}\big(x_k; \theta^{(m)}(k)\big) - r_k - \gamma \cdot \widehat{Q}\big(x'_k; \theta^{(m)}(k)\big) \Big) \cdot \nabla_\theta \sigma\big(x_k; \theta_i(k)\big)$$

$$= \theta_i(k) + \eta\epsilon \cdot \widehat{G}_k\big(\theta_i(k); \theta^{(m)}(k)\big), \tag{D.3}$$

where $(x_k, r_k, x'_k) \sim \widetilde{\mathcal{D}}$. Note that this definition is equivalent to (2.3).

- **Expected temporal-difference (ETD).** We consider the following expected TD dynamics $\breve{\theta}^{(m)}(k)$, where $k \in \mathbb{N}$, with $\breve{\theta}_i(0) = \theta_i(0)$ ($i \in [m]$) as its initialization,

$$\breve{\theta}_i(k+1) = \breve{\theta}_i(k) - \eta\epsilon \cdot \alpha \cdot \mathbb{E}_{\widetilde{\mathcal{D}}} \Big[ \Big( \widehat{Q}\big(x; \breve{\theta}^{(m)}(k)\big) - r - \gamma \cdot \widehat{Q}\big(x'; \breve{\theta}^{(m)}(k)\big) \Big) \cdot \nabla_\theta \sigma\big(x; \breve{\theta}_i(k)\big) \Big]$$

$$= \breve{\theta}_i(k) + \eta\epsilon \cdot \widehat{g}\big(\breve{\theta}_i(k); \breve{\theta}^{(m)}(k)\big). \tag{D.4}$$

- **Continuous-time temporal-difference (CTTD).** We consider the following continuous-time TD dynamics $\widetilde{\theta}^{(m)}(t)$, where $t \in \mathbb{R}_+$, with $\widetilde{\theta}_i(0) = \theta_i(0)$ ($i \in [m]$) as its initialization,

$$\frac{\mathrm{d}}{\mathrm{d}t}\widetilde{\theta}_i(t) = -\eta \cdot \alpha \cdot \mathbb{E}_{\widetilde{\mathcal{D}}} \Big[ \Big( \widehat{Q}\big(x; \widetilde{\theta}^{(m)}(t)\big) - r - \gamma \cdot \widehat{Q}\big(x'; \widetilde{\theta}^{(m)}(t)\big) \Big) \cdot \nabla_\theta \sigma\big(x; \widetilde{\theta}_i(t)\big) \Big]$$

$$= \eta \cdot \widehat{g}\big(\widetilde{\theta}_i(t); \widetilde{\theta}^{(m)}(t)\big). \tag{D.5}$$

- **Ideal particle (IP).** We consider the following ideal particle dynamics $\bar{\theta}^{(m)}(t)$, where $t \in \mathbb{R}_+$, with $\bar{\theta}_i(0) = \theta_i(0)$ ($i \in [m]$) as its initialization,

$$\frac{\mathrm{d}}{\mathrm{d}t}\bar{\theta}_i(t) = -\eta \cdot \alpha \cdot \mathbb{E}_{\widetilde{\mathcal{D}}} \Big[ \big( Q(x; \rho_t) - r - \gamma \cdot Q(x'; \rho_t) \big) \cdot \nabla_\theta \sigma\big(x; \bar{\theta}_i(t)\big) \Big]$$

$$= \eta \cdot g\big(\bar{\theta}_i(t); \rho_t\big), \tag{D.6}$$

where $\rho_t$ is the PDE solution in (3.4).

We aim to prove that $\widehat{\rho}_k = m^{-1} \cdot \sum_{i=1}^m \delta_{\theta_i(k)}$ weakly converges to $\rho_{k\epsilon}$. For any continuous function $f : \mathbb{R}^D \to \mathbb{R}$ such that $\|f\|_\infty \leq 1$ and $\mathrm{Lip}(f) \leq 1$, we use the IP, CTTD, and ETD dynamics as the interpolating dynamics,

$$
\overbrace{\left| \int f(\theta)\, \mathrm{d}\rho_{k\epsilon}(\theta) - \int f(\theta)\, \mathrm{d}\widehat{\rho}_k(\theta) \right|}^{\mathrm{PDE} - \mathrm{TD}}
$$

$$
\leq \left| \int f(\theta)\, \mathrm{d}\rho_{k\epsilon}(\theta) - m^{-1} \cdot \sum_{i=1}^m f\big(\bar{\theta}_i(k\epsilon)\big) \right| + \left| m^{-1} \cdot \sum_{i=1}^m f\big(\bar{\theta}_i(k\epsilon)\big) - m^{-1} \cdot \sum_{i=1}^m f\big(\widetilde{\theta}_i(k\epsilon)\big) \right|
$$

$$
+ \left| m^{-1} \cdot \sum_{i=1}^m f\big(\widetilde{\theta}_i(k\epsilon)\big) - m^{-1} \cdot \sum_{i=1}^m f\big(\check{\theta}_i(k)\big) \right| + \left| m^{-1} \cdot \sum_{i=1}^m f\big(\check{\theta}_i(k)\big) - m^{-1} \cdot \sum_{i=1}^m f\big(\theta_i(k)\big) \right|
$$

$$
\leq \underbrace{\left| \int f(\theta)\, \mathrm{d}\rho_{k\epsilon}(\theta) - m^{-1} \cdot \sum_{i=1}^m f\big(\bar{\theta}_i(k\epsilon)\big) \right|}_{\mathrm{PDE} - \mathrm{IP}} + \underbrace{\big\|\bar{\theta}^{(m)}(k\epsilon) - \widetilde{\theta}^{(m)}(k\epsilon)\big\|_{(m)}}_{\mathrm{IP} - \mathrm{CTTD}}
$$

$$
+ \underbrace{\big\|\widetilde{\theta}^{(m)}(k\epsilon) - \check{\theta}^{(m)}(k)\big\|_{(m)}}_{\mathrm{CTTD} - \mathrm{ETD}} + \underbrace{\big\|\check{\theta}^{(m)}(k) - \theta^{(m)}(k)\big\|_{(m)}}_{\mathrm{ETD} - \mathrm{TD}}, \tag{D.7}
$$

where the last inequality follows from the the fact that $\mathrm{Lip}(f) \leq 1$. Here the norm $\|\cdot\|_{(m)}$ of $\theta^{(m)} = \{\theta_i\}_{i=1}^m$ is defined as follows,

$$
\|\theta^{(m)}\|_{(m)} = \sup_{i \in [m]} \|\theta_i\|. \tag{D.8}
$$

In what follows, we define $B > 0$ as a constant that depends on $\alpha, \eta, \gamma, B_r$, and $B_j$ $(j \in \{0, 1, 2\})$, whose value varies from line to line. We establish the following lemmas to upper bound the terms on the right-hand side of (D.8).

**Lemma D.2** (Upper Bound of PDE – IP). Let $f$ be any continuous function such that $\|f\|_\infty \leq 1$ and $\mathrm{Lip}(f) \leq 1$. Under Assumptions 4.1 and 4.2, it holds for any $f$ that

$$
\sup_{t \in [0,T]} \left| \int f(\theta)\, \mathrm{d}\rho_t(\theta) - m^{-1} \cdot \sum_{i=1}^m f\big(\bar{\theta}_i(t)\big) \right| \leq B \cdot \sqrt{\log(mT/\delta)/m}
$$

with probability at least $1 - \delta$.

*Proof.* See §D.1.1 for a detailed proof. □

**Lemma D.3** (Upper Bound of IP – CTTD). Under Assumptions 4.1 and 4.2, it holds that

$$
\sup_{t \in [0,T]} \big\|\bar{\theta}^{(m)}(t) - \widetilde{\theta}^{(m)}(t)\big\|_{(m)} \leq B \cdot e^{BT} \cdot \sqrt{\log(m/\delta)/m}
$$

with probability at least $1 - \delta$.

*Proof.* See §D.1.2 for a detailed proof. □

**Lemma D.4** (Upper Bound of CTTD – ETD). Under Assumptions 4.1 and 4.2, it holds that

$$
\sup_{\substack{k \leq T/\epsilon \\ (k \in \mathbb{N})}} \big\|\widetilde{\theta}^{(m)}(k\epsilon) - \check{\theta}^{(m)}(k)\big\|_{(m)} \leq B \cdot e^{BT} \cdot \epsilon.
$$

*Proof.* See §D.1.3 for a detailed proof. □

**Lemma D.5** (Upper Bound of ETD – TD). Under Assumptions 4.1 and 4.2, it holds that

$$
\sup_{\substack{k \leq T/\epsilon \\ (k \in \mathbb{N})}} \big\|\check{\theta}^{(m)}(k) - \theta^{(m)}(k)\big\|_{(m)} \leq B \cdot e^{BT} \cdot \sqrt{\epsilon \cdot \big(D + \log(m/\delta)\big)}
$$

with probability at least $1 - \delta$

*Proof.* See §D.1.4 for a detailed proof. □

We are now ready to present the proof of Proposition D.1.

*Proof.* Plugging Lemmas D.2-D.5 into (D.7), we have that

$$\sup_{\substack{k \leq T/\epsilon \\ (k \in \mathbb{N})}} \left| \int f(\theta) \, \mathrm{d}\rho_{k\epsilon}(\theta) - \int f(\theta) \, \mathrm{d}\widehat{\rho}_k(\theta) \right|$$

$$\leq B \cdot e^{BT} \cdot \left( \sqrt{\log(m/\delta)/m} + \sqrt{\epsilon \cdot \big(D + \log(m/\delta)\big)} \right)$$

with probability at least $1 - \delta$. Thus, we complete the proof of Proposition D.1. □

## D.1 Proofs of Lemmas D.2-D.5

In this section, we present the proofs of Lemmas D.2-D.5, which are based on [6, 53, 54]. We include the required technical lemmas in §D.3. Recall that $B > 0$ is a constant that depends on $\alpha$, $\eta$, $\gamma$, $B_r$, and $B_j$ ($j \in \{0, 1, 2\}$), whose value varies from line to line.

### D.1.1 Proof of Lemma D.2

*Proof.* For the IP dynamics in (D.6), it holds that $\bar{\theta}_i(t) \sim \rho_t$ ($i \in [m]$) (Proposition 8.1.8 in [5]). Furthermore, since the randomness of $\bar{\theta}_i(t)$ comes from $\theta_i(0)$ while $\theta_i(0)$ ($i \in [m]$) are independent, we have that $\bar{\theta}_i(t) \overset{\text{i.i.d.}}{\sim} \rho_t$ ($i \in [m]$). Thus, we have that

$$\mathbb{E}_{\rho_t} \left[ m^{-1} \cdot \sum_{i=1}^{m} f\big(\bar{\theta}_i(t)\big) \right] = \int f(\theta) \, \mathrm{d}\rho_t(\theta).$$

Let $\theta^{1,(m)} = \{\theta_1, \ldots, \theta_i^1, \ldots, \theta_m\}$ and $\theta^{2,(m)} = \{\theta_1, \ldots, \theta_i^2, \ldots, \theta_m\}$ be two sets that only differ in the $i$-th element. Then, by the condition of Lemma D.2 that $\|f\|_\infty \leq 1$, we have that

$$\left| m^{-1} \cdot \sum_{j=1}^{m} f(\theta_j^1) - m^{-1} \cdot \sum_{j=1}^{m} f(\theta_j^2) \right| = m^{-1} \cdot \left| f(\theta_i^1) - f(\theta_i^2) \right| \leq 2/m.$$

Applying McDiarmid's inequality [70], we have for a fixed $t \in [0, T]$ that

$$\mathbb{P}\left( \left| m^{-1} \cdot \sum_{i=1}^{m} f\big(\bar{\theta}_i(t)\big) - \int f(\theta) \, \mathrm{d}\rho_t(\theta) \right| \geq p \right) \leq \exp(-mp^2/4). \tag{D.9}$$

Moreover, we have for any $s, t \in [0, T]$ that

$$\left| \left| m^{-1} \cdot \sum_{i=1}^{m} f\big(\bar{\theta}_i(t)\big) - \int f(\theta) \, \mathrm{d}\rho_t(\theta) \right| - \left| m^{-1} \cdot \sum_{i=1}^{m} f\big(\bar{\theta}_i(s)\big) - \int f(\theta) \, \mathrm{d}\rho_s(\theta) \right| \right|$$

$$\leq \left| m^{-1} \cdot \sum_{i=1}^{m} f\big(\bar{\theta}_i(t)\big) - m^{-1} \cdot \sum_{i=1}^{m} f\big(\bar{\theta}_i(s)\big) \right| + \left| \int f(\theta) \, \mathrm{d}\rho_t(\theta) - \int f(\theta) \, \mathrm{d}\rho_s(\theta) \right|$$

$$\leq \left\| \bar{\theta}^{(m)}(t) - \bar{\theta}^{(m)}(s) \right\|_{(m)} + \mathcal{W}_1(\rho_t, \rho_s)$$

$$\leq \left\| \bar{\theta}^{(m)}(t) - \bar{\theta}^{(m)}(s) \right\|_{(m)} + \mathcal{W}_2(\rho_t, \rho_s),$$

where the second inequality follows from the fact that $\mathrm{Lip}(f) \leq 1$ and (C.9). Applying (D.38) and (D.40) of Lemma D.8, we have for any $s, t \in [0, T]$ that

$$\left| \left| m^{-1} \cdot \sum_{i=1}^{m} f\big(\bar{\theta}_i(t)\big) - \int f(\theta) \, \mathrm{d}\rho_t(\theta) \right| - \left| m^{-1} \cdot \sum_{i=1}^{m} f\big(\bar{\theta}_i(s)\big) - \int f(\theta) \, \mathrm{d}\rho_s(\theta) \right| \right| \leq B \cdot |t - s|.$$

Applying the union bound to (D.9) for $t \in \iota \cdot \{0, 1, \ldots, \lfloor T/\iota \rfloor\}$, we have that

$$\mathbb{P}\left( \sup_{t \in [0,T]} \left| m^{-1} \cdot \sum_{i=1}^{m} f(\bar\theta_i(t)) - \int f(\theta) \, \mathrm{d}\rho_t(\theta) \right| \geq p + B \cdot \iota \right) \leq (T/\iota + 1) \cdot \exp(-mp^2/4).$$

Setting $\iota = m^{-1/2}$ and $p = B \cdot \sqrt{\log(mT/\delta)/m}$, we have that

$$\sup_{t \in [0,T]} \left| m^{-1} \cdot \sum_{i=1}^{m} f(\bar\theta_i(t)) - \int f(\theta) \, \mathrm{d}\rho_t(\theta) \right| \leq B \cdot \sqrt{\log(mT/\delta)/m}$$

with probability at least $1 - \delta$. Thus, we complete the proof of Lemma D.2. $\qquad\square$

### D.1.2   Proof of Lemma D.3

*Proof.* Recall that $g$ and $\widehat{g}$ are defined in (3.5) and (D.1), respectively, that is,

$$g(\theta; \rho) = -\alpha \cdot \mathbb{E}_{\widetilde{\mathcal{D}}}\Big[ \big( Q(x; \rho) - r - \gamma \cdot Q(x'; \rho) \big) \cdot \nabla_\theta \sigma(x; \theta) \Big],$$

$$\widehat{g}(\theta; \theta^{(m)}) = -\alpha \cdot \mathbb{E}_{\widetilde{\mathcal{D}}}\Big[ \big( \widehat{Q}(x; \theta^{(m)}) - r - \gamma \cdot \widehat{Q}(x'; \theta^{(m)}) \big) \cdot \nabla_\theta \sigma(x; \theta) \Big].$$

Following from the definition of $\widetilde\theta_i(t)$ and $\bar\theta_i(t)$ in (D.5) and (D.6), respectively, we have for any $i \in [m]$ and $t \in [0, T]$ that

$$\begin{aligned}
&\big\| \bar\theta_i(t) - \widetilde\theta_i(t) \big\| \\
&\leq \int_0^t \left\| \frac{\mathrm{d}\widetilde\theta_i(s)}{\mathrm{d}s} - \frac{\mathrm{d}\bar\theta_i(s)}{\mathrm{d}s} \right\| \mathrm{d}s \\
&= \eta \cdot \int_0^t \left\| \widehat{g}\big(\widetilde\theta_i(s); \widetilde\theta^{(m)}(s)\big) - g\big(\bar\theta_i(s); \rho_s\big) \right\| \mathrm{d}s \\
&\leq \eta \cdot \int_0^t \left\| \widehat{g}\big(\widetilde\theta_i(s); \widetilde\theta^{(m)}(s)\big) - \widehat{g}\big(\bar\theta_i(s); \bar\theta^{(m)}(s)\big) \right\| \mathrm{d}s + \eta \cdot \int_0^t \left\| \widehat{g}\big(\bar\theta_i(s); \bar\theta^{(m)}(s)\big) - g\big(\bar\theta_i(s); \rho_s\big) \right\| \mathrm{d}s \\
&\leq B \cdot \int_0^t \big\| \widetilde\theta^{(m)}(s) - \bar\theta^{(m)}(s) \big\|_{(m)} \, \mathrm{d}s + \eta \cdot \int_0^t \left\| \widehat{g}\big(\bar\theta_i(s); \bar\theta^{(m)}(s)\big) - g\big(\bar\theta_i(s); \rho_s\big) \right\| \mathrm{d}s, \quad \text{(D.10)}
\end{aligned}$$

where the last inequality follows from (D.35) of Lemma D.7. We now upper bound the second term on the right-hand side of (D.10). Following from the definition of $\widehat{Q}$, $Q$, and $\widehat{g}$ in (3.1), (3.2), and (D.1), respectively, we have for any $s \in [0, T]$ and $i \in [m]$ that

$$\left\| \widehat{g}\big(\bar\theta_i(s); \bar\theta^{(m)}(s)\big) - g\big(\bar\theta_i(s); \rho_s\big) \right\| = \alpha^2 \cdot \left\| m^{-1} \cdot \sum_{j=1}^{m} Z_i^j(s) \right\|, \tag{D.11}$$

where

$$Z_i^j(s) = \mathbb{E}_{\widetilde{\mathcal{D}}}\left[ \left( \sigma\big(x; \bar\theta_j(s)\big) - \int \sigma(x; \theta) \, \mathrm{d}\rho_s(\theta) - \gamma \cdot \sigma\big(x'; \bar\theta_j(s)\big) + \gamma \cdot \int \sigma(x'; \theta) \, \mathrm{d}\rho_s(\theta) \right) \cdot \nabla_\theta \sigma\big(x; \bar\theta_i(s)\big) \right].$$

Following from Assumptions 4.1 and 4.2, we have that $\|Z_i^j(s)\| \leq B$. When $i \neq j$, following from the fact that $\bar\theta_i(s) \overset{\text{i.i.d.}}{\sim} \rho_s$ $(i \in [m])$, it holds that $\mathbb{E}[Z_i^j(s) \,|\, \bar\theta_i(s)] = 0$. Following from Lemma D.9, we have for fixed $s \in [0, T]$ and $i \in [m]$ that

$$\begin{aligned}
\mathbb{P}\left( \left\| m^{-1} \cdot \sum_{j \neq i} Z_i^j(s) \right\| \geq B \cdot (m^{-1/2} + p) \right) &= \mathbb{E}\left[ \mathbb{P}\left( \left\| m^{-1} \cdot \sum_{j \neq i} Z_i^j(s) \right\| \geq B \cdot (m^{-1/2} + p) \,\middle|\, \bar\theta_i(s) \right) \right] \\
&\leq \exp(-mp^2). \quad \text{(D.12)}
\end{aligned}$$

By (C.9), we have that

$$\sup_{x \in \mathcal{X}} \left| \int \sigma(x; \theta) \, \mathrm{d}\rho_s(\theta) - \int \sigma(x; \theta) \, \mathrm{d}\rho_t(\theta) \right| \leq B \cdot \mathcal{W}_1(\rho_s, \rho_t) \leq B \cdot \mathcal{W}_2(\rho_s, \rho_t) \leq B \cdot |s - t|,$$

where the last inequality follows from (D.40) of Lemma D.8. Thus, following from Assumptions 4.1 and 4.2, Lemma D.8, and the fact that $\mathrm{Lip}(fg) \leq \|f\|_\infty \cdot \mathrm{Lip}(g) + \|g\|_\infty \cdot \mathrm{Lip}(f)$ for any functions $f$ and $g$, we have for any $s, t \in [0, T]$ that

$$\left| \left\| m^{-1} \cdot \sum_{j \neq i} Z_i^j(s) \right\| - \left\| m^{-1} \cdot \sum_{j \neq i} Z_i^j(t) \right\| \right| \leq B \cdot |t - s|.$$

Applying the union bound to (D.12) for $i \in [m]$ and $t \in \iota \cdot \{0, 1, \ldots, \lfloor T/\iota \rfloor\}$, we have that

$$\mathbb{P}\left( \sup_{\substack{i \in [m], \\ s \in [0,T]}} \left\| m^{-1} \cdot \sum_{j \neq i} Z_i^j(s) \right\| \geq B \cdot (m^{-1/2} + p) + B\iota \right) \leq m \cdot (T/\iota + 1) \cdot \exp(-mp^2).$$

Setting $\iota = m^{-1/2}$ and $p = B \cdot \sqrt{\log(mT/\delta)/m}$, we have that

$$\sup_{\substack{i \in [m], \\ s \in [0,T]}} \left\| m^{-1} \cdot \sum_{j \neq i} Z_i^j(s) \right\| \leq B \cdot \sqrt{\log(mT/\delta)/m} \tag{D.13}$$

with probability at least $1 - \delta$. When $i = j$, it holds that $\|m^{-1} \cdot Z_i^i(s)\| \leq B/m$ in (D.11), which follows from Assumptions 4.1 and 4.2. Thus, plugging (D.13) into (D.11), we have that

$$\sup_{\substack{i \in [m], \\ s \in [0,T]}} \left\| \widehat{g}\big(\bar{\theta}_i(s); \bar{\theta}^{(m)}(s)\big) - g\big(\bar{\theta}_i(s); \rho_s\big) \right\| \leq \sup_{\substack{i \in [m], \\ s \in [0,T]}} \alpha^2 \cdot \left( \left\| m^{-1} \cdot Z_i^i(s) \right\| + \left\| m^{-1} \cdot \sum_{j \neq i} Z_i^j(s) \right\| \right)$$

$$\leq B \cdot \sqrt{\log(mT/\delta)/m} \tag{D.14}$$

with probability at least $1 - \delta$.

Conditioning on the event in (D.14), we obtain from (D.10) that

$$\left\| \widetilde{\theta}^{(m)}(t) - \bar{\theta}^{(m)}(t) \right\|_{(m)} \leq B \cdot \int_0^t \left\| \widetilde{\theta}^{(m)}(s) - \bar{\theta}^{(m)}(s) \right\|_{(m)} \mathrm{d}s + BT \cdot \sqrt{\log(mT/\delta)/m}$$

for any $t \in [0, T]$. Following from Gronwall's Lemma [41], we have that

$$\left\| \widetilde{\theta}^{(m)}(t) - \bar{\theta}^{(m)}(t) \right\|_{(m)} \leq B \cdot e^{Bt} \cdot BT \cdot \sqrt{\log(mT/\delta)/m}$$

$$\leq B \cdot e^{BT} \cdot \sqrt{\log(m/\delta)/m}, \qquad \forall t \in [0, T]$$

with probability at least $1 - \delta$. Here the last inequality holds since we allow the value of $B$ to vary from line to line. Thus, we complete the proof of Lemma D.3 $\qquad \square$

### D.1.3 Proof of Lemma D.4

*Proof.* By the definition of $\widehat{g}$, $\breve{\theta}_i(t)$, and $\widetilde{\theta}_i(t)$ in (D.1), (D.4), and (D.5), respectively, it holds that

$$\left\| \widetilde{\theta}_i(k\epsilon) - \breve{\theta}_i(k) \right\| \leq \eta \cdot \int_0^{k\epsilon} \left\| \widehat{g}\big(\widetilde{\theta}_i(s); \widetilde{\theta}^{(m)}(s)\big) - \widehat{g}\big(\breve{\theta}_i(\lfloor s/\epsilon \rfloor); \breve{\theta}^{(m)}(\lfloor s/\epsilon \rfloor)\big) \right\| \mathrm{d}s$$

$$\leq \eta \cdot \int_0^{k\epsilon} \left\| \widehat{g}\big(\widetilde{\theta}_i(s); \widetilde{\theta}^{(m)}(s)\big) - \widehat{g}\big(\widetilde{\theta}_i(\lfloor s/\epsilon \rfloor \cdot \epsilon); \widetilde{\theta}^{(m)}(\lfloor s/\epsilon \rfloor \cdot \epsilon)\big) \right\| \mathrm{d}s$$

$$+ \eta \cdot \sum_{\ell=0}^{k-1} \left\| \widehat{g}\big(\widetilde{\theta}_i(\ell\epsilon); \widetilde{\theta}^{(m)}(\ell\epsilon)\big) - \widehat{g}\big(\breve{\theta}_i(\ell); \breve{\theta}^{(m)}(\ell)\big) \right\|$$

$$\leq B \cdot k \cdot \epsilon^2 + B \cdot \sum_{\ell=0}^{k-1} \left\| \widetilde{\theta}^{(m)}(\ell\epsilon) - \breve{\theta}^{(m)}(\ell) \right\|_{(m)},$$

where the last inequality follows from (D.35) of Lemma D.7 and (D.39) of Lemma D.8. Following from the definition of $\|\cdot\|_{(m)}$ in (D.8), it holds for any $k \leq T/\epsilon$ ($k \in \mathbb{N}$) that

$$\left\| \widetilde{\theta}^{(m)}(k\epsilon) - \breve{\theta}^{(m)}(k) \right\|_{(m)} \leq B \cdot T \cdot \epsilon + B \cdot \sum_{\ell=0}^{k-1} \left\| \widetilde{\theta}^{(m)}(\ell\epsilon) - \breve{\theta}^{(m)}(\ell) \right\|_{(m)}.$$

Following from the discrete Gronwall's lemma [41], we have that

$$\sup_{\substack{k \leq T/\epsilon \\ (k \in \mathbb{N})}} \big\| \widetilde{\theta}^{(m)}(k\epsilon) - \breve{\theta}^{(m)}(k) \big\|_{(m)} \leq B^2 \cdot T \cdot \epsilon \cdot e^{BT} \leq B \cdot e^{BT} \cdot \epsilon,$$

where the last inequality holds since we allow the value of $B$ to vary from line to line. Thus, we complete the proof of Lemma D.4. $\qquad\square$

### D.1.4 Proof of Lemma D.5

*Proof.* Let $\mathcal{G}_k = \sigma(\theta^{(m)}(0), z_0, \ldots, z_k)$ be the $\sigma$-algebra generated by $\theta^{(m)}(0)$ and $z_\ell = (x_\ell, r_\ell, x'_\ell)$ $(\ell \leq k)$. Recall that $\widehat{g}$ and $\widehat{G}_k$ are defined in (D.1) and (D.2), respectively. We have for any $i \in [m]$ and $k \in \mathbb{N}_+$ that

$$\mathbb{E}\Big[ \widehat{G}_k\big(\theta_i(k); \theta^{(m)}(k)\big) \,\Big|\, \mathcal{G}_{k-1} \Big] = \widehat{g}\big(\theta_i(k); \theta^{(m)}(k)\big).$$

Recall that $\theta^{(m)}(k)$ and $\breve{\theta}^{(m)}(k)$ are the TD and ETD dynamics defined in (D.3) and (D.4), respectively. Thus, we have for any $i \in [m]$ and $k \in \mathbb{N}_+$ that

$$\big\| \breve{\theta}_i(k) - \theta_i(k) \big\| = \eta\epsilon \cdot \bigg\| \sum_{\ell=0}^{k-1} \widehat{G}_\ell\big(\theta_i(\ell); \theta^{(m)}(\ell)\big) - \sum_{\ell=0}^{k-1} \widehat{g}\big(\breve{\theta}_i(\ell); \breve{\theta}^{(m)}(\ell)\big) \bigg\|$$

$$\leq \eta\epsilon \cdot \bigg\| \sum_{\ell=0}^{k-1} X_i(\ell) \bigg\| + \eta\epsilon \cdot \sum_{\ell=0}^{k-1} \Big\| \widehat{g}\big(\breve{\theta}_i(\ell); \breve{\theta}^{(m)}(\ell)\big) - \widehat{g}\big(\theta_i(\ell); \theta^{(m)}(\ell)\big) \Big\|$$

$$\leq \eta\epsilon \cdot \big\| A_i(k) \big\| + B\epsilon \cdot \sum_{\ell=0}^{k-1} \big\| \breve{\theta}^{(m)}(\ell) - \theta^{(m)}(\ell) \big\|_{(m)}, \qquad \text{(D.15)}$$

where the last inequality follows from (D.35) of Lemma D.7, and $X_i(\ell)$ and $A_i(k)$ are defined as

$$X_i(0) = 0,$$

$$X_i(\ell) = \widehat{G}_\ell\big(\theta_i(\ell); \theta^{(m)}(\ell)\big) - \mathbb{E}\Big[ \widehat{G}_\ell\big(\theta_i(\ell); \theta^{(m)}(\ell)\big) \,\Big|\, \mathcal{G}_{\ell-1} \Big] \quad \forall \ell \geq 1,$$

$$A_i(k) = \sum_{\ell=0}^{k-1} X_i(\ell).$$

Following from (D.32) of Lemma D.7, we have that $\|X_i(\ell)\| \leq B$. Thus, the stochastic process $\{A_i(k)\}_{k \in \mathbb{N}_+}$ is a martingale with $\|A_i(k) - A_i(k-1)\| \leq B$. Applying Lemma D.10, we have that

$$\mathbb{P}\Big( \max_{\substack{k \leq T/\epsilon \\ (k \in \mathbb{N}_+)}} \big\| A_i(k) \big\| \geq B \cdot \sqrt{T/\epsilon} \cdot (\sqrt{D} + p) \Big) \leq \exp(-p^2). \qquad \text{(D.16)}$$

Applying the union bound to (D.16) for $i \in [m]$, we have that

$$\mathbb{P}\Big( \max_{\substack{i \in [m], \\ k \leq T/\epsilon \; (k \in \mathbb{N}_+)}} \big\| A_i(k) \big\| \geq B \cdot \sqrt{T/\epsilon} \cdot (\sqrt{D} + p) \Big) \leq m \cdot \exp(-p^2).$$

By setting $p = \sqrt{\log(m/\delta)}$, we have that

$$\big\| A_i(k) \big\| \leq B \cdot \sqrt{T/\epsilon} \cdot \big(\sqrt{D} + \sqrt{\log(m/\delta)}\big), \quad \forall i \in [m], k \leq T/\epsilon \; (k \in \mathbb{N}_+) \qquad \text{(D.17)}$$

with probability at least $1 - \delta$. By (D.15) and (D.17), we have that

$$\big\| \breve{\theta}^{(m)}(k) - \theta^{(m)}(k) \big\|_{(m)}$$

$$\leq B \cdot \sqrt{T\epsilon} \cdot \big(\sqrt{D} + \sqrt{\log(m/\delta)}\big) + B\epsilon \cdot \sum_{\ell=0}^{k-1} \big\| \breve{\theta}^{(m)}(\ell) - \theta^{(m)}(\ell) \big\|_{(m)}, \quad \forall k \leq T/\epsilon \; (k \in \mathbb{N})$$

with probability at least $1 - \delta$. Applying the discrete Gronwall's Lemma [41], we have that

$$\big\| \breve{\theta}^{(m)}(k) - \theta^{(m)}(k) \big\|_{(m)} \leq B \cdot e^{BT} \cdot B \cdot \sqrt{T\epsilon} \cdot \big(\sqrt{D} + \sqrt{\log(m/\delta)}\big)$$

$$\leq B \cdot e^{BT} \cdot \sqrt{\epsilon \cdot \big(D + \log(m/\delta)\big)}, \quad \forall k \leq T/\epsilon \; (k \in \mathbb{N})$$

with probability at least $1 - \delta$. Here the last inequality holds since we allow the value of $B$ to vary from line to line. Thus, we complete the proof of Lemma D.5. $\qquad\square$

## D.2 Proof of Corollary 4.4

The proof of Corollary 4.4 follows from Theorem 4.3 and the following lemma, which characterizes the error of approximating the TD dynamics $\theta^{(m)}(k)$ in (3.3) using the PDE solution $\rho_t$ in (3.4).

**Lemma D.6.** Let $B$ be a constant that depends on $\alpha$, $\eta$, $\gamma$, $B_0$, $B_1$, and $B_2$. Under Assumptions 4.1 and 4.2, it holds for any $k \leq T/\epsilon$ $(k \in \mathbb{N})$ that

$$
\mathbb{E}_{x \sim \mathcal{D}}\left[\left(\widehat{Q}\big(x; \theta^{(m)}(k)\big) - Q^*(x)\right)^2\right]
$$
$$
\leq \mathbb{E}_{x \sim \mathcal{D}}\left[\left(Q(x; \rho_{k\epsilon}) - Q^*(x)\right)^2\right] + B \cdot e^{BT} \cdot \left(\sqrt{m^{-1} \cdot \log(m/\delta)} + \sqrt{\epsilon \cdot \left(D + \log(m/\delta)\right)}\right)
$$

with probability at least $1 - \delta$.

*Proof.* Recall that $\widehat{Q}$ and $Q(\cdot; \rho)$ are defined in (3.1) and (3.2), respectively. For notational simplicity, we denote the optimality gaps for $\theta^{(m)} = \{\theta_i\}_{i=1}^m$ and $\rho \in \mathscr{P}_2(\mathbb{R}^D)$ by

$$
L(\theta^{(m)}) = \mathbb{E}_{\mathcal{D}}\left[\left(\widehat{Q}(x; \theta^{(m)}) - Q^*(x)\right)^2\right], \tag{D.18}
$$

$$
\bar{L}(\rho) = \mathbb{E}_{\mathcal{D}}\left[\left(Q(x; \rho) - Q^*(x)\right)^2\right]. \tag{D.19}
$$

Recall that $\theta^{(m)}(k)$, $\bar{\theta}^{(m)}(k\epsilon)$, and $\rho_t$ are the TD dynamics, the IP dynamics, and the PDE solution defined in (D.3), (D.6), and (3.4), respectively. It holds for any $k \in \mathbb{N}$ that

$$
\left|L\big(\theta^{(m)}(k)\big) - \bar{L}(\rho_{k\epsilon})\right| \leq \underbrace{\left|L\big(\theta^{(m)}(k)\big) - L\big(\bar{\theta}^{(m)}(k\epsilon)\big)\right|}_{\text{(i)}} + \underbrace{\left|L\big(\bar{\theta}^{(m)}(k\epsilon)\big) - \bar{L}(\rho_{k\epsilon})\right|}_{\text{(ii)}}. \tag{D.20}
$$

In what follows, we upper bound the two terms on the right-hand side of (D.20).

**Upper bounding term (i) of** (D.20)**.** Following from the definition of $L$ in (D.18), it holds for any $k \in \mathbb{N}$ that

$$
\left|L\big(\theta^{(m)}(k)\big) - L\big(\bar{\theta}^{(m)}(k\epsilon)\big)\right|
$$
$$
= \left|\mathbb{E}_{\mathcal{D}}\left[\left(\widehat{Q}\big(x; \theta^{(m)}(k)\big) + \widehat{Q}\big(x; \bar{\theta}_i(k\epsilon)\big) - 2Q^*(x)\right) \cdot \left(\widehat{Q}\big(x; \theta^{(m)}(k)\big) - \widehat{Q}\big(x; \bar{\theta}_i(k\epsilon)\big)\right)\right]\right|. \tag{D.21}
$$

Following from (D.30), (D.31), and (D.36) of Lemma D.7, we have for any $k \in \mathbb{N}$ that

$$
\sup_{x \in \mathcal{X}}\left|\widehat{Q}\big(x; \theta^{(m)}(k)\big) + \widehat{Q}\big(x; \bar{\theta}_i(k\epsilon)\big) - 2Q^*(x)\right| \leq B, \tag{D.22}
$$

$$
\sup_{x \in \mathcal{X}}\left|\widehat{Q}\big(x; \theta^{(m)}(k)\big) - \widehat{Q}\big(x; \bar{\theta}_i(k\epsilon)\big)\right| \leq B \cdot \left\|\theta^{(m)}(k) - \bar{\theta}^{(m)}(k\epsilon)\right\|_{(m)}. \tag{D.23}
$$

Thus, we have that

$$
\left|L\big(\theta^{(m)}(k)\big) - L\big(\bar{\theta}^{(m)}(k\epsilon)\big)\right|
$$
$$
\leq B \cdot \left\|\theta^{(m)}(k) - \bar{\theta}^{(m)}(k\epsilon)\right\|_{(m)}
$$
$$
\leq B \cdot e^{BT} \cdot \left(\sqrt{\log(m/\delta)/m} + \sqrt{\epsilon \cdot \left(D + \log(m/\delta)\right)}\right), \quad \forall k \leq T/\epsilon \ (k \in \mathbb{N}) \tag{D.24}
$$

with probability at least $1 - \delta$. Here the last inequality follows from Lemmas D.3-D.5.

**Upper bounding term (ii) of** (D.20)**.** Let $t = k\epsilon$. It holds for any $t \in [0, T]$ that

$$
\left|L\big(\bar{\theta}^{(m)}(t)\big) - \bar{L}(\rho_t)\right| \leq \left|L\big(\bar{\theta}^{(m)}(t)\big) - \mathbb{E}_{\rho_t}\left[L\big(\bar{\theta}^{(m)}(t)\big)\right]\right| + \left|\mathbb{E}_{\rho_t}\left[L\big(\bar{\theta}^{(m)}(t)\big)\right] - \bar{L}(\rho_t)\right|, \tag{D.25}
$$

where the expectation is with respect to $\bar{\theta}_i(t) \overset{\text{i.i.d.}}{\sim} \rho_t$ $(i \in [m])$. For the second term on the right-hand side of (D.25), following from the fact that $\mathbb{E}_{\rho_t}[\widehat{Q}(x; \bar{\theta}^{(m)}(t))] = Q(x; \rho_t)$ for any $x \in \mathcal{X}$, we have that

$$
\left| \mathbb{E}_{\rho_t}\left[ L(\bar{\theta}^{(m)}(t)) \right] - \bar{L}(\rho_t) \right| = \left| \int \mathbb{E}_{\rho_t}\left[ \widehat{Q}(x; \bar{\theta}^{(m)}(t))^2 - Q(x; \rho_t)^2 \right] \mathrm{d}\mathcal{D}(x) \right|
$$
$$
= \left| \int \mathrm{Var}_{\rho_t}\left[ \widehat{Q}(x; \bar{\theta}^{(m)}(t)) \right] \mathrm{d}\mathcal{D}(x) \right|
$$
$$
\leq B/m, \tag{D.26}
$$

where the inequality follows from the fact that $\|\sigma\| \leq B$ in Assumption 4.2 and the independence of $\bar{\theta}_i(t)$ $(i \in [m])$. Let $\theta^{1,(m)} = \{\theta_1, \ldots, \theta_i^1, \ldots, \theta_m\}$ and $\theta^{2,(m)} = \{\theta_1, \ldots, \theta_i^2, \ldots, \theta_m\}$ be two sets that only differ in the $i$-th element. It holds that

$$
\left| L(\theta^{1,(m)}) - L(\theta^{2,(m)}) \right| \leq B \cdot m^{-1} \cdot \mathbb{E}_{\mathcal{D}}\left[ \left| \sigma(x; \theta_i^1) - \sigma(x; \theta_i^2) \right| \right] \leq B/m,
$$

where the first inequality follows from (D.21) and (D.22) and the second inequality follows from Assumption 4.2. Applying McDiarmid's inequality [70], we have for a fixed $t \in [0, T]$ that

$$
\mathbb{P}\left( \left| L(\bar{\theta}^{(m)}(t)) - \mathbb{E}_{\rho_t}\left[ L(\bar{\theta}^{(m)}(t)) \right] \right| \geq p \right) \leq \exp(-mp^2/B). \tag{D.27}
$$

It holds for any $s, t \in [0, T]$ that

$$
\left| \left| L(\bar{\theta}^{(m)}(t)) - \mathbb{E}_{\rho_t}\left[ L(\bar{\theta}^{(m)}(t)) \right] \right| - \left| L(\bar{\theta}^{(m)}(s)) - \mathbb{E}_{\rho_t}\left[ L(\bar{\theta}^{(m)}(s)) \right] \right| \right|
$$
$$
\leq B \cdot \left\| \bar{\theta}^{(m)}(t) - \bar{\theta}^{(m)}(s) \right\|_{(m)} \leq B \cdot |t - s|,
$$

where the first inequality follows from (D.21), (D.22), and (D.23) and the second inequality follows from (D.38) of Lemma D.8. Applying the union bound to (D.27) for $t \in \iota \cdot \{0, 1, \ldots, \lfloor T/\iota \rfloor\}$, we have that

$$
\mathbb{P}\left( \sup_{t \in [0,T]} \left| L(\bar{\theta}^{(m)}(t)) - \mathbb{E}_{\rho_t}\left[ L(\bar{\theta}^{(m)}(t)) \right] \right| \geq p + B\iota \right) \leq (T/\iota + 1) \cdot \exp(-mp^2/B),
$$

Setting $\iota = m^{-1/2}$ and $p = B \cdot \sqrt{\log(mT\delta)/m}$, we have that

$$
\sup_{t \in [0,T]} \left| L(\bar{\theta}^{(m)}(t)) - \mathbb{E}_{\rho_t}\left[ L(\bar{\theta}^{(m)}(t)) \right] \right| \leq B \cdot \sqrt{\log(mT\delta)/m} \tag{D.28}
$$

with probability at least $1 - \delta$. Plugging (D.26) and (D.28) into (D.25), noting that $t = k\epsilon$, we have that

$$
\left| L(\bar{\theta}^{(m)}(k\epsilon)) - \bar{L}(\rho_{k\epsilon}) \right| \leq B \cdot \sqrt{\log(mT\delta)/m}, \quad \forall k \leq T/\epsilon \ (k \in \mathbb{N}) \tag{D.29}
$$

with probability at least $1 - \delta$.

Plugging (D.24) and (D.29) into (D.20), we have that

$$
\left| L(\theta^{(m)}(k)) - \bar{L}(\rho_{k\epsilon}) \right| \leq B \cdot e^{BT} \cdot \left( \sqrt{\log(m/\delta)/m} + \sqrt{\epsilon \cdot (D + \log(m/\delta))} \right), \quad \forall k \leq T/\epsilon \ (k \in \mathbb{N})
$$

with probability at least $1 - \delta$. Thus, we complete the proof of Lemma D.6. $\qquad\square$

### D.3   Technical Lemmas for §D

In what follows, we present the technical lemmas used in §D. Recall that $\widehat{Q}$, $\widehat{g}$, and $\widehat{G}_k$ are defined in (3.1), (D.1), and (D.2), respectively. Let $B > 0$ be a constant depending on $\alpha$, $\eta$, $\gamma$, $B_r$, and $B_j$ $(j \in \{0, 1, 2\})$, whose value varies from line to line.

**Lemma D.7.** Under Assumptions 4.1 and 4.2, it holds for $\theta^{(m)} = \{\theta_i\}_{i=1}^m$ and $\underline{\theta}^{(m)} = \{\underline{\theta}_i\}_{i=1}^m$ that

$$\sup_{x \in \mathcal{X}} |\widehat{Q}(x; \theta^{(m)})| \leq B, \tag{D.30}$$

$$\sup_{x \in \mathcal{X}} |\widehat{Q}(x; \theta^{(m)}) - \widehat{Q}(x; \underline{\theta}^{(m)})| \leq B \cdot \|\theta^{(m)} - \underline{\theta}^{(m)}\|_{(m)}, \tag{D.31}$$

$$\|\widehat{G}_k(\theta_i; \theta^{(m)})\| \leq B, \tag{D.32}$$

$$\|\widehat{G}_k(\theta_i; \theta^{(m)}) - \widehat{G}_k(\underline{\theta}_i; \underline{\theta}^{(m)})\| \leq B \cdot \|\theta^{(m)} - \underline{\theta}^{(m)}\|_{(m)}, \quad \forall k \in \mathbb{N}, \tag{D.33}$$

$$\|\widehat{g}(\theta_i; \theta^{(m)})\| \leq B, \tag{D.34}$$

$$\|\widehat{g}(\theta_i; \theta^{(m)}) - \widehat{g}(\underline{\theta}_i; \underline{\theta}^{(m)})\| \leq B \cdot \|\theta^{(m)} - \underline{\theta}^{(m)}\|_{(m)}. \tag{D.35}$$

Meanwhile, for any $Q \in \mathcal{F}$, it holds that

$$\sup_{x \in \mathcal{X}} \|Q(x)\| \leq B. \tag{D.36}$$

For any $\rho \in \mathscr{P}_2(\mathbb{R}^D)$, it holds that

$$\|g(\theta; \rho)\| \leq B. \tag{D.37}$$

*Proof.* For (D.30) and (D.31) of Lemma D.7, following from Assumptions 4.1 and 4.2 and the definition of $\widehat{Q}$ in (3.1), we have for any $x \in \mathcal{X}$, $\theta^{(m)}$, and $\underline{\theta}^{(m)}$ that

$$|\widehat{Q}(x; \theta^{(m)})| \leq \alpha \cdot m^{-1} \sum_{i=1}^m |\sigma(x; \theta_i)| \leq B,$$

$$|\widehat{Q}(x; \theta^{(m)}) - \widehat{Q}(x; \underline{\theta}^{(m)})| \leq \alpha \cdot m^{-1} \sum_{i=1}^m |\sigma(x; \theta_i) - \sigma(x; \underline{\theta}_i)| \leq B \cdot \|\theta^{(m)} - \underline{\theta}^{(m)}\|_{(m)}.$$

For (D.32) and (D.33) of Lemma D.7, following from the definition of $\widehat{G}_k$ in (D.2), we have for any $\theta^{(m)}$ and $\underline{\theta}^{(m)}$ that

$$\|\widehat{G}_k(\theta_i; \theta^{(m)})\| = \alpha \cdot |\widehat{Q}(x_k; \theta^{(m)}) - r_k - \gamma \cdot \widehat{Q}(x_k'; \theta^{(m)})| \cdot \|\nabla_\theta \sigma(x_k; \theta_i)\| \leq B,$$

$$\|\widehat{G}_k(\theta_i; \theta^{(m)}) - \widehat{G}_k(\underline{\theta}_i; \underline{\theta}^{(m)})\|$$
$$\leq \alpha \cdot \sup_{\theta^{(m)}} |\widehat{Q}(x_k; \theta^{(m)}) - r_k - \gamma \cdot \widehat{Q}(x_k'; \theta^{(m)})| \cdot \|\nabla_\theta \sigma(x_k; \theta_i) - \nabla_\theta \sigma(x_k; \underline{\theta}_i)\|$$
$$+ \alpha \cdot |\widehat{Q}(x_k; \theta^{(m)}) - \gamma \cdot \widehat{Q}(x_k'; \theta^{(m)}) - \widehat{Q}(x_k; \underline{\theta}^{(m)}) + \gamma \cdot \widehat{Q}(x_k'; \underline{\theta}^{(m)})| \cdot \sup_{\theta_i \in \mathbb{R}^D} \|\nabla_\theta \sigma(x_k; \theta_i)\|$$
$$\leq B \cdot \|\theta^{(m)} - \underline{\theta}^{(m)}\|_{(m)}.$$

The inequalities in (D.34) and (D.35) of Lemma D.7 for $\widehat{g}$ follow from the fact that

$$\widehat{g}(\theta_i; \theta^{(m)}) = \mathbb{E}_{(x_k, r_k, x_k') \sim \widetilde{\mathcal{D}}} [G_k(\theta_i; \theta^{(m)})].$$

The inequalities in (D.36) and (D.37) follow from the definition of $\mathcal{F}$ and $g$ in (4.3) and (3.5), respectively. Thus, we complete the proof of Lemma D.7. □

Recall that $\rho_t$ is the PDE solution in (3.4) and $\widetilde{\theta}^{(m)}(t)$ and $\bar{\theta}^{(m)}(t)$ are the CTTD and IP dynamics defined in (D.5) and (D.6), respectively.

**Lemma D.8.** Under Assumptions 4.1 and 4.2, it holds for any $s, t \in [0, T]$ that

$$\|\bar{\theta}^{(m)}(t) - \bar{\theta}^{(m)}(s)\|_{(m)} \leq B \cdot |t - s|, \tag{D.38}$$

$$\|\widetilde{\theta}^{(m)}(t) - \widetilde{\theta}^{(m)}(s)\|_{(m)} \leq B \cdot |t - s|, \tag{D.39}$$

$$\mathcal{W}_2(\rho_t, \rho_s) \leq B \cdot |t - s|. \tag{D.40}$$

*Proof.* For (D.38) of Lemma D.8, by the definition of $\bar{\theta}_i(t)$ in (D.6) and (D.37) of Lemma D.7, we have for any $s, t \in [0, T]$ and $i \in [m]$ that

$$\left\| \bar{\theta}_i(t) - \bar{\theta}_i(s) \right\| = \eta \cdot \int_s^t \left\| g\big(\bar{\theta}_i(\tau); \rho_\tau\big) \right\| \mathrm{d}\tau \leq B \cdot |t - s|.$$

Similarly, for (D.39) of Lemma D.8, by the definition of $\widetilde{\theta}_i(t)$ in (D.5) and (D.34) of Lemma D.7, we have for any $i \in [m]$ and $s, t \in [0, T]$ that $\|\widetilde{\theta}_i(t) - \widetilde{\theta}_i(s)\| \leq B \cdot |t - s|$.

For (D.40) of Lemma D.8, following from the fact that $\bar{\theta}_i(t) \overset{\text{i.i.d.}}{\sim} \rho_t$ ($i \in [m]$) and the definition of $\mathcal{W}_2$ in (2.4), it holds for any $s, t \in [0, T]$ that

$$\mathcal{W}_2(\rho_t, \rho_s) \leq \mathbb{E}\Big[ \big\| \bar{\theta}_i(t) - \bar{\theta}_i(s) \big\|^2 \Big]^{1/2} \leq B \cdot |t - s|.$$

Thus, we complete the proof of Lemma D.8. $\qquad\square$

**Lemma D.9** (Lemma 30 in [53])**.** Let $\{X_i\}_{i=1}^m$ be i.i.d. random variables with $\|X_i\| \leq \xi$ and $\mathbb{E}[X_i] = 0$. Then, it holds for any $p > 0$ that

$$\mathbb{P}\bigg( \Big\| m^{-1} \cdot \sum_{i=1}^m X_i \Big\| \geq C\xi \cdot (m^{-1/2} + p) \bigg) \leq \exp(-mp^2),$$

where $C > 0$ is an absolute constant.

**Lemma D.10** (Lemma A.3 in [6] and Lemma 31 in [53])**.** Let $X_k \in \mathbb{R}^D$ ($k \in \mathbb{N}$) be a martingale with respect to the filtration $\mathcal{G}_k$ ($k \geq 0$) with $X_0 = 0$. We assume for $\xi > 0$ and any $\lambda \in \mathbb{R}^D$ that

$$\mathbb{E}\Big[ \exp\big(\langle \lambda, X_k - X_{k-1}\rangle\big) \,\Big|\, \mathcal{G}_{k-1} \Big] \leq \exp\big(\xi^2 \cdot \|\lambda\|^2/2\big).$$

Then, it holds that

$$\mathbb{P}\Big( \max_{\substack{k \leq n \\ (k \in \mathbb{N})}} \|X_k\| \geq C\xi \cdot \sqrt{n} \cdot (\sqrt{D} + p) \Big) \leq \exp(-p^2),$$

where $C > 0$ is an absolute constant.

# E   Auxiliary Lemmas

We use the definition of absolutely continuous curves in $\mathscr{P}_2(\mathbb{R}^D)$ in [5].

**Definition E.1** (Absolutely Continuous Curve)**.** Let $\beta : [a, b] \to \mathscr{P}_2(\mathbb{R}^D)$ be a curve. Then, $\beta$ is an absolutely continuous curve if there exists a square-integrable function $f : [a, b] \to \mathbb{R}$ such that

$$\mathcal{W}_2(\beta_s, \beta_t) \leq \int_s^t f(\tau) \, \mathrm{d}\tau$$

for any $a \leq s < t \leq b$.

Then, we have the following first variation formula.

**Lemma E.2** (First Variation Formula, Theorem 8.4.7 in [5])**.** Given $\nu \in \mathscr{P}_2(\mathbb{R}^D)$ and an absolutely continuous curve $\mu : [0, T] \to \mathscr{P}_2(\mathbb{R}^D)$, let $\beta : [0, 1] \to \mathscr{P}_2(\mathbb{R}^D)$ be the geodesic connecting $\mu_t$ and $\nu$. It holds that

$$\frac{\mathrm{d}}{\mathrm{d}t} \frac{\mathcal{W}_2(\mu_t, \nu)^2}{2} = -\langle \mu_t', \beta_0' \rangle_{\mu_t},$$

where $\mu_t' = \partial_t \mu_t$, $\beta_0' = \partial_t \beta_t \,|_{t=0}$, and the inner product is defined in (2.5).

**Lemma E.3** (Talagrand's Inequality, Corollary 2.1 in [59])**.** Let $\nu$ be $N(0, \kappa \cdot I_D)$. It holds for any $\mu \in \mathscr{P}_2(\mathbb{R}^D)$ that

$$\mathcal{W}_2(\mu, \nu)^2 \leq 2D_{\mathrm{KL}}(\mu \,\|\, \nu)/\kappa.$$

**Lemma E.4** (Eulerian Representation of Geodesics, Proposition 5.38 in [68])**.** Let $\beta : [0, 1] \to \mathscr{P}_2(\mathbb{R}^D)$ be a geodesic and $v$ be the corresponding vector field such that $\partial_t \beta_t = -\operatorname{div}(\beta_t \cdot v_t)$. It holds that

$$\partial_t(\beta_t \cdot v_t) = -\operatorname{div}(\beta_t \cdot v_t \otimes v_t).$$

[Supplementary Material 2]

# A Pseudocode of TD Learning

In this section, we present the pseudocode of TD learning in Algorithm 1, which is introduced in §3.

---

**Algorithm 1** Temporal-Difference Learning with Two-Layer Neural Network for Policy Evaluation

---

**Initialization:** $\theta_i(0) \overset{\text{i.i.d.}}{\sim} \rho_0$ ($i \in [m]$), number of iterations $K = \lfloor T/\epsilon \rfloor$, and policy $\pi$ of interest.

**for** $k = 0, \ldots, K-1$ **do**

    Sample the state-action pair $(s, a)$ from the stationary distribution $\mathcal{D}$ of $\pi$, receive the reward $r$, and obtain the subsequent state-action pair $(s', a')$.

    Calculate the Bellman residual $\delta = \widehat{Q}(x; \theta^{(m)}(k)) - r - \gamma \cdot \widehat{Q}(x'; \theta^{(m)}(k))$, where $x = (s, a)$ and $x' = (s', a')$.

    Perform the TD update $\theta_i(k+1) \leftarrow \theta_i(k) - \eta\epsilon \cdot \alpha \cdot \delta \cdot \nabla_\theta \sigma(x; \theta_i(k))$ ($i \in [m]$).

**end for**

**Output:** $\{\theta^{(m)}(k)\}_{k=0}^{K-1}$

---

# B Q-Learning and Policy Improvement

In this section, we extend our analysis of TD to Q-learning and soft Q-learning for policy improvement. In §B.1, we introduce Q-learning and its mean-field limit. In §B.2, we establish the global optimality and convergence of Q-learning. In §B.3, we further extend our analysis to soft Q-learning, which is equivalent to policy gradient.

## B.1 Q-Learning

Q-learning aims to solve the following projected Bellman optimality equation,

$$Q = \Pi_\mathcal{F} \mathcal{T}^* Q. \tag{B.1}$$

Here $\mathcal{T}^*$ is the Bellman optimality operator, which is defined as follows,

$$\mathcal{T}^* Q(s, a) = \mathbb{E}\big[r + \gamma \cdot \max_{\underline{a} \in \mathcal{A}} Q(s', \underline{a}) \,\big|\, r \sim R(\cdot \,|\, s, a), s' \sim P(\cdot \,|\, s, a)\big].$$

When $\Pi_\mathcal{F}$ is the identity mapping, the fixed point solution to (B.1) is the Q-function $Q^{\pi^*}$ of the optimal policy $\pi^*$, which maximizes the expected total reward $J(\pi)$ defined in (2.1) [65]. We consider the parameterization of the Q-function in (3.1) and update the parameter $\theta^{(m)}$ as follows,

$$\theta_i(k+1) \tag{B.2}$$
$$= \theta_i(k) - \eta\epsilon \cdot \alpha \cdot \Big(\widehat{Q}\big(s_k, a_k; \theta^{(m)}(k)\big) - r_k - \gamma \cdot \max_{\underline{a} \in \mathcal{A}} \widehat{Q}\big(s'_k, \underline{a}; \theta^{(m)}(k)\big)\Big) \cdot \nabla_\theta \sigma\big(s_k, a_k; \theta_i(k)\big),$$

where $i \in [m]$, $(s_k, a_k)$ is sampled from the stationary distribution $\mathcal{D}_\mathrm{E} \in \mathscr{P}(\mathcal{S} \times \mathcal{A})$ of an exploration policy $\pi_\mathrm{E}$, $r_k \sim R(\cdot \,|\, s_k, a_k)$ is the reward, and $s'_k \sim P(\cdot \,|\, s_k, a_k)$ is the subsequent state. For notational simplicity, we denote by $\widetilde{\mathcal{D}}_\mathrm{E} \in \mathscr{P}(\mathcal{S} \times \mathcal{A} \times \mathbb{R} \times \mathcal{S})$ the distribution of $(s_k, a_k, r_k, s'_k)$. For an initial distribution $\nu_0 \in \mathscr{P}(\mathbb{R}^D)$, we initialize $\{\theta_i\}_{i=1}^m$ as $\theta_i \overset{\text{i.i.d.}}{\sim} \rho_0$ ($i \in [m]$). See Algorithm 2 for a detailed description.

**Mean-Field Limit.** Corresponding to $\epsilon \to 0^+$ and $m \to \infty$, the mean-field limit of the Q-learning dynamics in (B.2) is characterized by the following PDE with $\nu_0$ as the initial distribution,

$$\partial_t \nu_t = -\eta \cdot \mathrm{div}\big(\nu_t \cdot h(\cdot; \nu_t)\big). \tag{B.3}$$

Here $h(\cdot; \nu_t) : \mathbb{R}^D \to \mathbb{R}^D$ is a vector field, which is defined as follows,

$$h(\theta; \nu) = -\alpha \cdot \mathbb{E}_{(s,a,r,s') \sim \widetilde{\mathcal{D}}_\mathrm{E}}\Big[\big(Q(s, a; \nu) - r - \gamma \cdot \max_{\underline{a} \in \mathcal{A}} Q(s', \underline{a}; \nu)\big) \cdot \nabla_\theta \sigma(s, a; \theta)\Big]. \tag{B.4}$$

In parallel to Proposition 3.1, the empirical distribution $\widehat{\nu}_k^{(m)} = m^{-1} \cdot \sum_{i=1}^m \delta_{\theta_i(k)}$ weakly converges to $\nu_{k\epsilon}$ as $\epsilon \to 0^+$ and $m \to \infty$.

**Algorithm 2** Q-Learning with Two-Layer Neural Network for Policy Improvement

---

**Initialization.** $\theta_i(0) \overset{\text{i.i.d.}}{\sim} \nu_0$ $(i \in [m])$, number of iterations $K = \lfloor T/\epsilon \rfloor$, and exploration policy $\pi_{\mathrm{E}}$.
**for** $k = 0, \ldots, K-1$ **do**
    Sample the state-action pair $(s, a)$ from the stationary distribution $\mathcal{D}_{\mathrm{E}}$ of $\pi_{\mathrm{E}}$, receive the reward $r$, and obtain the subsequent state $s'$.
    Calculate the Bellman residual $\delta = \widehat{Q}(x; \theta^{(m)}(k)) - r - \gamma \cdot \widehat{Q}(x'; \theta^{(m)}(k))$, where $x = (s, a)$ and $x' = (s', \mathrm{argmax}_{\underline{a} \in \mathcal{A}} \widehat{Q}(s', \underline{a}; \theta^{(m)}(k)))$.
    Perform the Q-learning update $\theta_i(k+1) \leftarrow \theta_i(k) - \eta\epsilon \cdot \alpha \cdot \delta \cdot \nabla_\theta \sigma(x; \theta_i(k))$ $(i \in [m])$.
**end for**
**Output:** $\{\theta^{(m)}(k)\}_{k=0}^{K-1}$

---

## B.2 Global Optimality and Convergence of Q-Learning

The $\max$ operator in the Bellman optimality operator $\mathcal{T}^*$ makes the analysis of Q-learning more challenging than that of TD. Correspondingly, we lay out an extra regularity condition on the exploration policy $\pi_{\mathrm{E}}$. Recall that the function class $\mathcal{F}$ is defined in (4.3).

**Assumption B.1.** We assume for an absolute constant $\kappa > 0$ and any $Q^1, Q^2 \in \mathcal{F}$ that

$$\mathbb{E}_{(s,a) \sim \mathcal{D}_{\mathrm{E}}}\left[ \left(Q^1(s,a) - Q^2(s,a)\right)^2 \right] \geq (\gamma + \kappa)^2 \cdot \mathbb{E}_{(s,a) \sim \mathcal{D}_{\mathrm{E}}}\left[ \left(\max_{\underline{a} \in \mathcal{A}} Q^1(s, \underline{a}) - \max_{\underline{a} \in \mathcal{A}} Q^2(s, \underline{a})\right)^2 \right].$$

Although Assumption B.1 is strong, we are not aware of any weaker regularity condition in the literature, even in the linear setting [25, 55, 78] and the NTK regime [21]. Let the initial distribution $\nu_0$ be the standard Gaussian distribution $N(0, I_D)$. In parallel to Theorem 4.3, we establish the following theorem, which characterizes the global optimality and convergence of Q-learning. Recall that we write $\mathcal{X} = \mathcal{S} \times \mathcal{A}$ and $x = (s, a) \in \mathcal{X}$. Also, $\nu_t$ is the PDE solution in (B.3), while $\theta^{(m)}(k)$ is the Q-learning dynamics in (B.2).

**Theorem B.2.** There exists a unique fixed point solution to the projected Bellman optimality equation $Q = \Pi_{\mathcal{F}} \mathcal{T}^* Q$, which takes the form of $Q^\dagger(x) = \int \sigma(x; \theta) \, \mathrm{d}\bar{\nu}(\theta)$. We assume that $D_{\chi^2}(\bar{\nu} \,\|\, \nu_0) < \infty$ and $\bar{\nu}(\theta) > 0$ for any $\theta \in \mathbb{R}^D$. Under Assumptions 4.1, 4.2, and B.1, it holds for $\eta = \alpha^{-2}$ that

$$\inf_{t \in [0,T]} \mathbb{E}_{x \sim \mathcal{D}_{\mathrm{E}}}\left[ \left(Q(x; \nu_t) - Q^\dagger(x)\right)^2 \right] \leq \frac{(\kappa + \gamma) \cdot D_{\chi^2}(\bar{\nu} \,\|\, \nu_0)}{2\kappa \cdot T} + \frac{(\kappa + \gamma) \cdot C_*}{\kappa \cdot \alpha}, \tag{B.5}$$

where $C_* > 0$ is a constant depending on $D_{\chi^2}(\bar{\nu} \,\|\, \nu_0)$, $B_1$, $B_2$, and $B_r$. Moreover, it holds with probability at least $1 - \delta$ that

$$\min_{\substack{k \leq T/\epsilon \\ (k \in \mathbb{N})}} \mathbb{E}_{x \sim \mathcal{D}_{\mathrm{E}}}\left[ \left(\widehat{Q}(x; \theta^{(m)}(k)) - Q^\dagger(x)\right)^2 \right]$$

$$\leq \frac{(\kappa + \gamma) \cdot D_{\chi^2}(\bar{\nu} \,\|\, \nu_0)}{2\kappa \cdot T} + \frac{(\kappa + \gamma) \cdot C_*}{\kappa \cdot \alpha} + \Delta(\epsilon, m, \delta, T), \tag{B.6}$$

where $\Delta(\epsilon, m, \delta, T) > 0$ is an error term such that

$$\lim_{m \to \infty} \lim_{\epsilon \to 0^+} \Delta(\epsilon, m, \delta, T) = 0.$$

*Proof.* See §B.4 for a detailed proof. $\square$

Theorem B.2 proves that the optimality gap $\mathbb{E}_{x \sim \mathcal{D}_{\mathrm{E}}}[(Q(x; \nu_t) - Q^\dagger(x))^2]$ decays to zero at a sublinear rate up to the error of $O(\alpha^{-1})$, where $\alpha > 0$ is the scaling parameter in (3.1). In parallel to Theorem 4.3, varying $\alpha$ leads to a tradeoff between such an error of $O(\alpha^{-1})$ and the deviation of $\nu_t$ from $\nu_0$. Moreover, based on the counterparts of Proposition 3.1 and Lemma D.6, Theorem B.2 gives the global optimality and convergence of the Q-learning dynamics $\theta^{(m)}(k)$ in (B.2), which is in parallel to Corollary 4.4.

## B.3 Soft Q-Learning

In this section, we generalize Theorem B.2 to soft Q-learning. To introduce soft Q-learning, we first define the soft Bellman optimality operator as follows,

$$\mathcal{T}_\beta Q(s,a) = \mathbb{E}\big[r + \gamma \cdot \mathrm{softmax}^\beta_{\underline{a} \in \mathcal{A}} Q(s', \underline{a}) \,\big|\, r \sim R(\cdot \,|\, s,a), s' \sim P(\cdot \,|\, s,a)\big],$$

where the softmax operator is defined as follows,

$$\mathrm{softmax}^\beta_{\underline{a} \in \mathcal{A}} Q(s, \underline{a}) = \beta \cdot \log \mathbb{E}_{\underline{a} \sim \bar{\pi}(\cdot \,|\, s)}\Big[\exp\big(\beta^{-1} \cdot Q(s, \underline{a})\big)\Big].$$

Here $\bar{\pi}(\cdot \,|\, s)$ is the uniform policy. Soft Q-learning aims to find the fixed point solution to the projected soft Bellman optimality equation $Q = \Pi_\mathcal{F} \mathcal{T}_\beta Q$. In parallel to the Q-learning dynamics in (B.2), we consider the following soft Q-learning dynamics,

$$\theta_i(k+1) \tag{B.7}$$
$$= \theta_i(k) - \eta\epsilon \cdot \alpha \cdot \Big(\widehat{Q}\big(s_k, a_k; \theta^{(m)}(k)\big) - r_k - \gamma \cdot \mathrm{softmax}^\beta_{\underline{a} \in \mathcal{A}} \widehat{Q}\big(s'_k, \underline{a}; \theta^{(m)}(k)\big)\Big) \cdot \nabla_\theta \sigma\big(s_k, a_k; \theta_i(k)\big),$$

whose mean-field limit is characterized by the following PDE,

$$\partial_t \nu_t = -\eta \cdot \mathrm{div}\big(\nu_t \cdot h(\cdot; \nu_t)\big). \tag{B.8}$$

In parallel to (B.4), $h(\cdot; \nu_t) : \mathbb{R}^D \to \mathbb{R}^D$ is a vector field, which is defined as follows,

$$h(\theta; \nu) = -\alpha \cdot \mathbb{E}_{(s,a,r,s') \sim \widetilde{\mathcal{D}}_\mathrm{E}}\Big[\big(Q(s,a;\nu) - r - \gamma \cdot \mathrm{softmax}^\beta_{\underline{a} \in \mathcal{A}} Q(s', \underline{a}; \nu)\big) \cdot \nabla_\theta \sigma(s, a; \theta)\Big].$$

In parallel to Assumption B.1, we lay out the following regularity condition.

**Assumption B.3.** We assume for an absolute constant $\kappa > 0$ and any $\nu^1, \nu^2 \in \mathscr{P}(\mathbb{R}^D)$ that

$$\mathbb{E}_{(s,a) \sim \mathcal{D}_\mathrm{E}}\Big[\big(Q(s,a;\nu^1) - Q(s,a;\nu^2)\big)^2\Big]$$
$$\geq (\gamma + \kappa)^2 \cdot \mathbb{E}_{(s,a) \sim \mathcal{D}_\mathrm{E}}\Big[\big(\mathrm{softmax}^\beta_{\underline{a} \in \mathcal{A}} Q(s, \underline{a}; \nu^1) - \mathrm{softmax}^\beta_{\underline{a} \in \mathcal{A}} Q(s, \underline{a}; \nu^2)\big)^2\Big].$$

The following proposition parallels Theorem B.2, which characterizes the global optimality and convergence of soft Q-learning. Recall that $\nu_t$ is the PDE solution in (B.8) and $\theta^{(m)}(k)$ is the soft Q-learning dynamics in (B.7).

**Proposition B.4.** There exists a unique fixed point solution to the projected soft Bellman optimality equation $Q = \Pi_\mathcal{F} \mathcal{T}_\beta Q$, which takes the form of $Q^\ddagger(x) = \int \sigma(x; \theta) \, \mathrm{d}\underline{\nu}(\theta)$. We assume that $D_{\chi^2}(\underline{\nu} \,\|\, \nu_0) < \infty$ and $\underline{\nu}(\theta) > 0$ for any $\theta \in \mathbb{R}^D$. Under Assumptions 4.1, 4.2, and B.3, it holds for $\eta = \alpha^{-2}$ that

$$\inf_{t \in [0,T]} \mathbb{E}_{x \sim \mathcal{D}_\mathrm{E}}\Big[\big(Q(x; \nu_t) - Q^\ddagger(x)\big)^2\Big] \leq \frac{(\kappa + \gamma) \cdot D_{\chi^2}(\underline{\nu} \,\|\, \nu_0)}{2\kappa \cdot T} + \frac{(\kappa + \gamma) \cdot C_*}{\kappa \cdot \alpha},$$

where $C_* > 0$ is a constant depending on $D_{\chi^2}(\underline{\nu} \,\|\, \nu_0)$, $B_1$, $B_2$, and $B_r$. Moreover, it holds with probability at least $1 - \delta$ that

$$\min_{\substack{k \leq T/\epsilon \\ (k \in \mathbb{N})}} \mathbb{E}_{x \sim \mathcal{D}_\mathrm{E}}\Big[\big(\widehat{Q}\big(x; \theta^{(m)}(k)\big) - Q^\ddagger(x)\big)^2\Big] \leq \frac{(\kappa + \gamma) \cdot D_{\chi^2}(\underline{\nu} \,\|\, \nu_0)}{2\kappa \cdot T} + \frac{(\kappa + \gamma) \cdot C_*}{\kappa \cdot \alpha} + \Delta(\epsilon, m, \delta, T),$$

where $\Delta(\epsilon, m, \delta, T) > 0$ is an error term such that

$$\lim_{m \to \infty} \lim_{\epsilon \to 0^+} \Delta(\epsilon, m, \delta, T) = 0.$$

*Proof.* Replacing the max operator by the softmax operator in the proof of Theorem B.2 in §B.4 implies Proposition B.4. $\qquad\square$

Moreover, soft Q-learning is equivalent to a variant of policy gradient [37, 57, 58, 61]. Hence, Proposition B.4 also characterizes the global optimality and convergence of such a variant of policy gradient.

## B.4 Proof of Theorem B.2

For notational simplicity, we denote by $\mathbb{E}_{\mathcal{D}_{\mathrm{E}}}$ the expectation with respect to $x \sim \mathcal{D}_{\mathrm{E}}$ and $\mathbb{E}_{\widetilde{\mathcal{D}}_{\mathrm{E}}}$ the expectation with respect to $(x, r, x') \sim \widetilde{\mathcal{D}}_{\mathrm{E}}$.

*Proof.* In parallel to the proof of Lemma 5.1 in §C.1, to establish the existence and uniqueness of the fixed point solution to the projected Bellman optimality equation $Q = \Pi_{\mathcal{F}} \mathcal{T}^* Q$, it suffices to show that $\Pi_{\mathcal{F}} \mathcal{T}^* : \mathcal{F} \to \mathcal{F}$ is a contraction mapping. In particular, it holds for any $Q^1, Q^2 \in \mathcal{F}$ that

$$
\|\Pi_{\mathcal{F}} \mathcal{T}^* Q^1 - \Pi_{\mathcal{F}} \mathcal{T}^* Q^2\|^2_{\mathcal{L}_2(\mathcal{D}_{\mathrm{E}})} \leq \gamma^2 \cdot \mathbb{E}_{\widetilde{\mathcal{D}}_{\mathrm{E}}} \left[ \left( \max_{a \in \mathcal{A}} Q^1(s', \underline{a}) - \max_{a \in \mathcal{A}} Q^2(s', \underline{a}) \right)^2 \right]
$$

$$
= \gamma^2 \cdot \mathbb{E}_{\mathcal{D}_{\mathrm{E}}} \left[ \left( \max_{a \in \mathcal{A}} Q^1(s, \underline{a}) - \max_{a \in \mathcal{A}} Q^2(s, \underline{a}) \right)^2 \right]
$$

$$
\leq \frac{\gamma^2}{(\gamma + \kappa)^2} \cdot \mathbb{E}_{\mathcal{D}_{\mathrm{E}}} \left[ \left( Q^1(s, a) - Q^2(s, a) \right)^2 \right],
$$

where the equality follows from the fact that $\mathcal{D}_{\mathrm{E}}$ is the stationary distribution and the last inequality follows from Assumption B.1. Thus, $\Pi_{\mathcal{F}} \mathcal{T}^* : \mathcal{F} \to \mathcal{F}$ is a contraction mapping. Following from the Banach fixed point theorem [28], there exists a unique fixed point solution $Q^\dagger \in \mathcal{F}$ to the projected Bellman optimality equation $Q = \Pi_{\mathcal{F}} \mathcal{T}^* Q$. Moreover, in parallel to the proof of Lemma 5.1 in §C.1, there exists $\nu^\dagger \in \mathscr{P}_2(\mathbb{R}^D)$ such that $Q(x; \nu^\dagger) = Q^\dagger(x)$, $h(x; \nu^\dagger) = 0$, and $\mathcal{W}_2(\nu^\dagger, \nu_0) \leq \alpha^{-1} \cdot \bar{D}$, where $\bar{D} = D_{\chi^2}(\bar{\nu} \,\|\, \nu_0)^{1/2}$.

For notational simplicity, we define $Q^{\mathcal{A}}(x) = \max_{\underline{a} \in \mathcal{A}} Q(s, \underline{a})$. In parallel to (C.13) in the proof of Lemma 5.2 in §C.2, we have that

$$
\frac{\mathrm{d}}{\mathrm{d}t} \frac{\mathcal{W}_2(\nu_t, \nu^\dagger)^2}{2} = \eta \cdot \underbrace{\int_0^1 \langle \partial_s h(\cdot; \beta_s), v_s \rangle_{\beta_s} \mathrm{d}s}_{(\mathrm{i})} + \eta \cdot \underbrace{\int_0^1 \int \langle h(\theta; \beta_s), \partial_s(v_s \cdot \beta_s)(\theta) \rangle \mathrm{d}\theta \, \mathrm{d}s}_{(\mathrm{ii})}, \quad \text{(B.9)}
$$

where $\beta : [0, 1] \to \mathscr{P}_2(\mathbb{R}^D)$ is the geodesic connecting $\nu_t$ and $\nu^\dagger$ with $\partial_s \beta_s = -\operatorname{div}(\beta_s \cdot v_s)$.

**Upper bounding term (i) of (B.9).** In parallel to (C.5) and (C.6) in the proof of Lemma C.1, we have that

$$
\langle \partial_s h(\cdot; \beta_s), v_s \rangle_{\beta_s} = -\mathbb{E}_{\widetilde{\mathcal{D}}_{\mathrm{E}}} \left[ \partial_s \big( Q(x; \beta_s) - \gamma \cdot Q^{\mathcal{A}}(x'; \beta_s) \big) \cdot \partial_s Q(x; \beta_s) \right] \quad \text{(B.10)}
$$

$$
\leq -\mathbb{E}_{\mathcal{D}_{\mathrm{E}}} \left[ \big( \partial_s Q(x; \beta_s) \big)^2 \right] + \gamma \cdot \mathbb{E}_{\mathcal{D}_{\mathrm{E}}} \left[ \big( \partial_s Q(x; \beta_s) \big)^2 \right]^{1/2} \cdot \mathbb{E}_{\mathcal{D}_{\mathrm{E}}} \left[ \big( \partial_s Q^{\mathcal{A}}(x; \beta_s) \big)^2 \right]^{1/2}.
$$

For the second term on the right-hand side of (B.10), we have that

$$
\mathbb{E}_{\mathcal{D}_{\mathrm{E}}} \left[ \big( \partial_s Q^{\mathcal{A}}(x; \beta_s) \big)^2 \right] = \lim_{u \to 0} \mathbb{E}_{\mathcal{D}_{\mathrm{E}}} \left[ \left( u^{-1} \cdot \big( Q^{\mathcal{A}}(x; \beta_{s+u}) - Q^{\mathcal{A}}(x; \beta_s) \big) \right)^2 \right]
$$

$$
\leq (\gamma + \kappa)^{-2} \cdot \lim_{u \to 0} u^{-2} \cdot \mathbb{E}_{\mathcal{D}_{\mathrm{E}}} \left[ \big( Q(x; \beta_{s+u}) - Q(x; \beta_s) \big)^2 \right]
$$

$$
= (\gamma + \kappa)^{-2} \cdot \mathbb{E}_{\mathcal{D}_{\mathrm{E}}} \left[ \big( \partial_s Q(x; \beta_s) \big)^2 \right], \quad \text{(B.11)}
$$

where the inequality follows from Assumption B.1 and the fact that $Q(\cdot; \nu) \in \alpha \cdot \mathcal{F}$. Plugging (B.11) into (B.10), we have that

$$
\langle \partial_s h(\cdot; \beta_s), v_s \rangle_{\beta_s} \leq -\frac{\kappa}{\gamma + \kappa} \cdot \mathbb{E}_{\mathcal{D}_{\mathrm{E}}} \left[ \big( \partial_s Q(x; \beta_s) \big)^2 \right],
$$

which further implies that

$$
\int_0^1 \langle \partial_s h(\cdot; \beta_s), v_s \rangle_{\beta_s} \mathrm{d}s \leq -\frac{\kappa}{\gamma + \kappa} \cdot \int_0^1 \mathbb{E}_{\mathcal{D}_{\mathrm{E}}} \left[ \big( \partial_s Q(x; \beta_s) \big)^2 \right] \mathrm{d}s
$$

$$
\leq -\frac{\kappa}{\gamma + \kappa} \cdot \mathbb{E}_{\mathcal{D}_{\mathrm{E}}} \left[ \left( \int_0^1 \partial_s Q(x; \beta_s) \, \mathrm{d}s \right)^2 \right]
$$

$$
= -\frac{\kappa}{\gamma + \kappa} \cdot \mathbb{E}_{\mathcal{D}_{\mathrm{E}}} \left[ \big( Q(x; \nu_t) - Q(x; \nu^\dagger) \big)^2 \right]. \quad \text{(B.12)}
$$

**Upper bounding term (ii) of** (B.9). In parallel to the proof of Lemma C.2 in §C.2, noting that $|Q^{\mathcal{A}}(x;\nu)| \leq \sup_{x\in\mathcal{X}}|Q(x;\nu)|$ for any $\nu \in \mathscr{P}_2(\mathbb{R}^D)$, we have that

$$\big\|\nabla_\theta h(\theta;\nu_t)\big\|_{\mathrm{F}} \leq \alpha \cdot B_2 \cdot \big(2\alpha \cdot B_1 \cdot \mathcal{W}_2(\nu_t,\nu_0) + B_r\big).$$

In parallel to (C.15) and (C.16), we have that

$$\int_0^1 \int \Big|\big\langle h(\theta;\beta_s), \partial_s(v_s \cdot \beta_s)(\theta)\big\rangle\Big|\,\mathrm{d}\theta\,\mathrm{d}s \leq C_* \cdot \alpha^{-1}, \tag{B.13}$$

where $C_* > 0$ is a constant that depends on $\bar{D}$, $B_1$, $B_2$, and $B_r$.

Plugging (B.12) and (B.13) into (B.9), we have that

$$\frac{\mathrm{d}}{\mathrm{d}t}\frac{\mathcal{W}_2(\nu_t,\nu^\dagger)^2}{2} \leq -\frac{\eta\cdot\kappa}{\gamma+\kappa}\cdot\mathbb{E}_{\mathcal{D}_{\mathrm{E}}}\Big[\big(Q(x;\nu_t)-Q(x;\nu^\dagger)\big)^2\Big] + C_* \cdot \eta \cdot \alpha^{-1}.$$

Thus, in parallel to the proof of Theorem 4.3 in §5, we have that

$$\inf_{t\in[0,T]}\mathbb{E}_{\mathcal{D}}\Big[\big(Q(x;\nu_t)-Q^\dagger(x)\big)^2\Big] \leq \frac{(\kappa+\gamma)\cdot D_{\chi^2}(\bar{\nu}\,\|\,\nu_0)}{2\kappa\cdot T} + C_* \cdot \alpha^{-1} \cdot \frac{\kappa+\gamma}{\kappa},$$

which completes the proof of (B.5) in Theorem B.2. Meanwhile, in parallel to the proof of Lemma D.6 in §D.2, we upper bound the error of approximating $\widehat{\nu}_k$ by $\nu_{k\epsilon}$, which further implies (B.6) of Theorem B.2. $\qquad\square$

## C  Proofs of Supporting Lemmas

For notational simplicity, we denote by $\mathbb{E}_{\mathcal{D}}$ the expectation with respect to $x \sim \mathcal{D}$ and $\mathbb{E}_{\widetilde{\mathcal{D}}}$ the expectation with respect to $(x,r,x') \sim \widetilde{\mathcal{D}}$. Also, with a slight abuse of notations, we write $\theta^{(m)} = \{\theta_i\}_{i=1}^m$.

### C.1  Proof of Lemma 5.1

*Proof.* **Existence and uniqueness of** $Q^*$. To establish the existence of the fixed point solution $Q^*$ to the projected Bellman equation $Q = \Pi_{\mathcal{F}}\mathcal{T}^\pi Q$, it suffices to show that $\Pi_{\mathcal{F}}\mathcal{T}^\pi : \mathcal{F} \to \mathcal{F}$ is a contraction mapping. It holds for any $Q^1, Q^2 \in \mathcal{F}$ that

$$\|\Pi_{\mathcal{F}}\mathcal{T}^\pi Q^1 - \Pi_{\mathcal{F}}\mathcal{T}^\pi Q^2\|_{\mathcal{L}_2(\mathcal{D})}^2 \leq \gamma^2 \cdot \mathbb{E}_{\widetilde{\mathcal{D}}}\Big[\big(Q^1(x')-Q^2(x')\big)^2\Big]$$
$$= \gamma^2 \cdot \big\|Q^1 - Q^2\big\|_{\mathcal{L}_2(\mathcal{D})}^2,$$

where the last equality follows from the fact that $\mathcal{D}$ is the stationary distribution. Thus, $\Pi_{\mathcal{F}}\mathcal{T}^\pi : \mathcal{F} \to \mathcal{F}$ is a contraction mapping. Note that $\mathcal{F}$ is complete. Following from the Banach fixed point theorem [28], there exists a unique $Q^* \in \mathcal{F}$ that solves the projected Bellman equation $Q = \Pi_{\mathcal{F}}\mathcal{T}^\pi Q$. Moreover, by the definition of $\mathcal{F}$ in (4.3), there exists $\bar{\rho} \in \mathscr{P}_2(\mathbb{R}^D)$ such that

$$Q^*(x) = \int \sigma(x;\theta)\,\mathrm{d}\bar{\rho}(\theta).$$

**Proof of (i) in Lemma 5.1.** We define

$$\rho^* = \rho_0 + \alpha^{-1} \cdot (\bar{\rho} - \rho_0). \tag{C.1}$$

By the definition of $Q(\cdot;\rho)$ in (3.2) and the fact that $Q(x;\rho_0) = 0$, we have that $Q(x;\rho^*) = Q^*(x)$, which completes the proof of (i) in Lemma 5.1.

**Proof of (ii) in Lemma 5.1.** For (ii) of Lemma 5.1, note that $Q(\cdot;\rho^*) = \Pi_{\mathcal{F}}\mathcal{T}^\pi Q(\cdot;\rho^*)$. Thus, we have that

$$\big\langle Q(\cdot;\rho^*) - \mathcal{T}^\pi Q(\cdot;\rho^*), f(\cdot) - Q(\cdot;\rho^*)\big\rangle_{\mathcal{D}} \geq 0, \quad \forall f \in \mathcal{F},$$

which further implies that

$$\mathbb{E}_{\widetilde{\mathcal{D}}}\Big[\big(Q(x;\rho^*) - r - \gamma \cdot Q(x';\rho^*)\big) \cdot \int \sigma(x;\theta)\,\mathrm{d}(\rho - \bar{\rho})(\theta)\Big] \geq 0, \quad \forall \rho \in \mathscr{P}_2(\mathbb{R}^D). \qquad \text{(C.2)}$$

Let $\rho = (\mathrm{id} + h \cdot v)_{\sharp}\bar{\rho}$ for a sufficiently small scaling parameter $h \in \mathbb{R}_+$ and any Lipschitz-continuous mapping $v : \mathbb{R}^D \to \mathbb{R}^D$. Then, following from (C.2), we have that

$$\int \mathbb{E}_{\widetilde{\mathcal{D}}}\Big[\big(Q(x;\rho^*) - r - \gamma \cdot Q(x';\rho^*)\big) \cdot \big(\sigma\big(x;\theta + h \cdot v(\theta)\big) - \sigma(x;\theta)\big)\Big]\,\mathrm{d}\bar{\rho}(\theta) \geq 0 \qquad \text{(C.3)}$$

for any $v : \mathbb{R}^D \to \mathbb{R}^D$. Dividing the both sides of (C.3) by $h$ and letting $h \to 0^+$, we have for any $v : \mathbb{R}^D \to \mathbb{R}^D$ that

$$0 \leq \int \mathbb{E}_{\widetilde{\mathcal{D}}}\Big[\big(Q(x;\rho^*) - r - \gamma \cdot Q(x';\rho^*)\big) \cdot \big\langle \nabla_\theta \sigma(x;\theta), v(\theta)\big\rangle\Big]\,\mathrm{d}\bar{\rho}(\theta)$$

$$= -\alpha^{-1} \cdot \int \big\langle g(\theta;\rho^*), v(\theta)\big\rangle\,\mathrm{d}\bar{\rho}(\theta),$$

where the equality follows from the definition of $g$ in (3.5). Thus, we have that $g(\theta;\rho^*) = 0$ for $\bar{\rho}$-a.e., which completes the proof of (ii) in Lemma 5.1.

**Proof of (iii) in Lemma 5.1.** Following from the definition of $\rho^*$ in (C.1), we have that

$$D_{\chi^2}(\rho^* \,\|\, \rho_0)$$
$$= \int \Big(\frac{\rho^*(\theta)}{\rho_0(\theta)} - 1\Big)^2\,\mathrm{d}\rho_0(\theta) = \int \Big(\frac{(1 - \alpha^{-1}) \cdot \rho_0(\theta) + \alpha^{-1} \cdot \bar{\rho}(\theta)}{\rho_0(\theta)} - 1\Big)^2\,\mathrm{d}\rho_0(\theta) = \alpha^{-2} \cdot \bar{D}^2,$$

where $\bar{D} = D_{\chi^2}(\bar{\rho} \,\|\, \rho_0)^{1/2}$. By Lemma E.3, we have that

$$\mathcal{W}_2(\rho^*, \rho_0) \leq D_{\mathrm{KL}}(\rho^* \,\|\, \rho_0)^{1/2} \leq D_{\chi^2}(\rho^* \,\|\, \rho_0)^{1/2} \leq \alpha^{-1} \cdot \bar{D},$$

which completes the proof of (iii) in Lemma 5.1. $\qquad \square$

## C.2  Proof of Lemma 5.2

We first introduce the following lemmas. The first lemma establishes the one-point monotonicity of $g(\cdot;\beta_t)$ along a curve $\beta : [0,1] \to \mathscr{P}_2(\mathbb{R}^D)$ on the Wasserstein space.

**Lemma C.1.** Let $\beta : [0,1] \to \mathscr{P}_2(\mathbb{R}^D)$ be a curve such that $\partial_t \beta_t = -\operatorname{div}(\beta_t \cdot v_t)$ for a vector field $v$. We have that

$$\big\langle \partial_t g(\cdot;\beta_t), v_t\big\rangle_{\beta_t} \leq -(1 - \gamma) \cdot \mathbb{E}_{\mathcal{D}}\Big[\big(\partial_t Q(x;\beta_t)\big)^2\Big].$$

Furthermore, we have that

$$\int_0^1 \big\langle \partial_s g(\cdot;\beta_s), v_s\big\rangle_{\beta_s}\,\mathrm{d}s \leq -(1 - \gamma) \cdot \mathbb{E}_{\mathcal{D}}\Big[\big(Q(x;\beta_0) - Q(x;\beta_1)\big)^2\Big]. \qquad \text{(C.4)}$$

*Proof.* Following from the definition of $g$ in (3.5), we have that

$$\partial_t g(\theta;\beta_t) = -\alpha \cdot \mathbb{E}_{\widetilde{\mathcal{D}}}\Big[\partial_t\big(Q(x;\beta_t) - \gamma \cdot Q(x';\beta_t)\big) \cdot \nabla_\theta \sigma(x;\theta)\Big].$$

Thus, following from integration by parts and the continuity equation $\partial_t \beta_t = -\operatorname{div}(\beta_t \cdot v_t)$, we have that

$$\big\langle \partial_t g(\cdot;\beta_t), v_t\big\rangle_{\beta_t} = -\int \Big\langle \alpha \cdot \mathbb{E}_{\widetilde{\mathcal{D}}}\Big[\partial_t\big(Q(x;\beta_t) - \gamma \cdot Q(x';\beta_t)\big) \cdot \nabla_\theta \sigma(x;\theta)\Big], v_t(\theta) \cdot \beta_t(\theta)\Big\rangle\,\mathrm{d}\theta$$

$$= -\int \alpha \cdot \mathbb{E}_{\widetilde{\mathcal{D}}}\Big[\partial_t\big(Q(x;\beta_t) - \gamma \cdot Q(x';\beta_t)\big) \cdot \sigma(x;\theta)\Big] \cdot \partial_t \beta_t(\theta)\,\mathrm{d}\theta$$

$$= -\mathbb{E}_{\widetilde{\mathcal{D}}}\Big[\partial_t\big(Q(x;\beta_t) - \gamma \cdot Q(x';\beta_t)\big) \cdot \partial_t Q(x;\beta_t)\Big], \qquad \text{(C.5)}$$

where the last equality follows from the definition of $Q$ in (3.2). Applying the Cauchy-Schwartz inequality to (C.5), we have that

$$
\begin{aligned}
\langle \partial_t g(\cdot; \beta_t), v_t \rangle_{\beta_t} &= -\mathbb{E}_{\widetilde{\mathcal{D}}}\Big[\big(\partial_t Q(x; \beta_t)\big)^2\Big] + \gamma \cdot \mathbb{E}_{\widetilde{\mathcal{D}}}\big[\partial_t Q(x'; \beta_t) \cdot \partial_t Q(x; \beta_t)\big] \\
&\le -\mathbb{E}_{\widetilde{\mathcal{D}}}\Big[\big(\partial_t Q(x; \beta_t)\big)^2\Big] + \gamma \cdot \mathbb{E}_{\widetilde{\mathcal{D}}}\Big[\big(\partial_t Q(x; \beta_t)\big)^2\Big]^{1/2} \cdot \mathbb{E}_{\widetilde{\mathcal{D}}}\Big[\big(\partial_t Q(x'; \beta_t)\big)^2\Big]^{1/2} \\
&= -(1-\gamma) \cdot \mathbb{E}_{\mathcal{D}}\Big[\big(\partial_t Q(x; \beta_t)\big)^2\Big],
\end{aligned}
\tag{C.6}
$$

where the last equality follows from the fact that the marginal distributions of $\widetilde{\mathcal{D}}$ with respect to $x$ and $x'$ are $\mathcal{D}$, since $\mathcal{D}$ is the stationary distribution. Furthermore, we have that

$$
\begin{aligned}
\int_0^1 \langle \partial_s g(\cdot; \beta_s), v_s \rangle_{\beta_s} \, \mathrm{d}s &\le -(1-\gamma) \cdot \int_0^1 \mathbb{E}_{\mathcal{D}}\Big[\big(\partial_s Q(x; \beta_s)\big)^2\Big] \, \mathrm{d}s \\
&\le -(1-\gamma) \cdot \mathbb{E}_{\mathcal{D}}\bigg[\bigg(\int_0^1 \partial_s Q(x; \beta_s) \, \mathrm{d}s\bigg)^2\bigg] \\
&= -(1-\gamma) \cdot \mathbb{E}_{\mathcal{D}}\Big[\big(Q(x; \beta_1) - Q(x; \beta_0)\big)^2\Big],
\end{aligned}
$$

which completes the proof of Lemma C.1. $\qquad\square$

The following lemma upper bounds the norms of $Q$ and $\nabla_\theta g$.

**Lemma C.2.** Under Assumptions 4.1 and 4.2, it holds for any $\rho \in \mathscr{P}_2(\mathbb{R}^D)$ that

$$
\sup_{x \in \mathcal{X}} \big|Q(x; \rho)\big| \le \alpha \cdot \min\big\{B_1 \cdot \mathcal{W}_2(\rho, \rho_0), \, B_0\big\},
\tag{C.7}
$$

$$
\sup_{\theta \in \mathbb{R}^D} \big\|\nabla_\theta g(\theta; \rho)\big\|_{\mathrm{F}} \le \alpha \cdot B_2 \cdot \min\big\{2\alpha \cdot B_1 \cdot \mathcal{W}_2(\rho, \rho_0) + B_r, \, 2\alpha \cdot B_0 + B_r\big\}.
\tag{C.8}
$$

*Proof.* We introduce the Wasserstein-1 distance, which is defined as

$$
\mathcal{W}_1(\mu^1, \mu^2) = \inf\big\{\mathbb{E}\big[\|X - Y\|\big] \,\big|\, \mathrm{law}(X) = \mu^1, \mathrm{law}(Y) = \mu^2\big\}
$$

for any $\mu^1, \mu^2 \in \mathscr{P}(\mathbb{R}^D)$ with finite first moments. Thus, we have that $\mathcal{W}_1(\mu^1, \mu^2) \le \mathcal{W}_2(\mu^1, \mu^2)$. The Wasserstein-1 distance has the following dual representation [5],

$$
\mathcal{W}_1(\mu^1, \mu^2) = \sup\bigg\{\int f(x) \, \mathrm{d}(\mu^1 - \mu^2)(x) \,\bigg|\, \text{continuous } f : \mathbb{R}^D \to \mathbb{R}, \mathrm{Lip}(f) \le 1\bigg\}.
\tag{C.9}
$$

Following from Assumptions 4.1 and 4.2, we have that $\|\nabla_\theta \sigma(x; \theta)\| \le B_1$ for any $x \in \mathcal{X}$ and $\theta \in \mathbb{R}^D$, which implies that $\mathrm{Lip}(\sigma(x; \cdot)/B_1) \le 1$ for any $x \in \mathcal{X}$. Note that $Q(x; \rho_0) = 0$ for any $x \in \mathcal{X}$. Thus, by (C.9) we have for any $\rho \in \mathscr{P}_2(\mathbb{R}^D)$ and $x \in \mathcal{X}$ that

$$
\big|Q(x; \rho)\big| = \alpha \cdot \bigg|\int \sigma(x; \theta) \cdot \mathrm{d}(\rho - \rho_0)(\theta)\bigg| \le \alpha \cdot B_1 \cdot \mathcal{W}_1(\rho, \rho_0) \le \alpha \cdot B_1 \cdot \mathcal{W}_2(\rho, \rho_0).
\tag{C.10}
$$

Meanwhile, following from Assumptions 4.1 and 4.2, we have for any $x \in \mathcal{X}$ and $\rho \in \mathscr{P}_2(\mathbb{R}^D)$ that

$$
\big|Q(x; \rho)\big| = \alpha \cdot \bigg|\int \sigma(x; \theta) \, \mathrm{d}\rho(\theta)\bigg| \le \alpha \cdot B_0.
\tag{C.11}
$$

Combining (C.10) and (C.11), we have for any $\rho \in \mathscr{P}_2(\mathbb{R}^D)$ that

$$
\sup_{x \in \mathcal{X}} \big|Q(x; \rho)\big| \le \alpha \cdot \min\big\{B_1 \cdot \mathcal{W}_2(\rho, \rho_0), \, B_0\big\},
\tag{C.12}
$$

which completes the proof of (C.7) in Lemma C.2. Following from the definition of $g$ in (3.5), we have for any $x \in \mathcal{X}$ and $\rho \in \mathscr{P}_2(\mathbb{R}^D)$ that

$$
\begin{aligned}
\big\|\nabla_\theta g(\theta; \rho)\big\|_{\mathrm{F}} &\le \alpha \cdot \mathbb{E}_{\widetilde{\mathcal{D}}}\Big[\big|Q(x; \rho) - r - \gamma \cdot Q(x'; \rho)\big| \cdot \big\|\nabla_{\theta\theta}^2 \sigma(x; \theta)\big\|_{\mathrm{F}}\Big] \\
&\le \alpha \cdot \min\big\{2\alpha \cdot B_1 \cdot \mathcal{W}_2(\rho, \rho_0) + B_r, \, 2\alpha \cdot B_0 + B_r\big\} \cdot B_2.
\end{aligned}
$$

Here the last inequality follows from (C.12) and the fact that $\|\nabla_{\theta\theta}^2 \sigma(x; \theta)\|_{\mathrm{F}} \le B_2$ for any $x \in \mathcal{X}$ and $\rho \in \mathscr{P}_2(\mathbb{R}^D)$, which follows from Assumptions 4.1 and 4.2. Thus, we complete the proof of Lemma C.2. $\qquad\square$

We are now ready to present the proof of Lemma 5.2.

*Proof.* Recall that $\rho_t$ is the PDE solution in (3.4), that is,

$$\partial_t \rho_t = -\eta \cdot \operatorname{div}\big(\rho_t \cdot g(\cdot; \rho_t)\big),$$

where

$$g(\theta; \rho) = -\alpha \cdot \mathbb{E}_{\widetilde{\mathcal{D}}}\Big[\big(Q(x; \rho) - r - \gamma \cdot Q(x'; \rho)\big) \cdot \nabla_\theta \sigma(x; \theta)\Big].$$

We fix a $t \in [0, T]$. We denote by $\beta : [0, 1] \to \mathscr{P}_2(\mathbb{R}^D)$ the geodesic connecting $\rho_t$ and $\rho^*$. Specifically, $\beta$ satisfies that $\beta'_s = -\operatorname{div}(\beta_s \cdot v_s)$ for a vector field $v$. Following from Lemma E.2, we have that

$$
\begin{aligned}
\frac{\mathrm{d}}{\mathrm{d}t} \frac{\mathcal{W}_2(\rho_t, \rho^*)^2}{2} &= -\eta \cdot \big\langle g(\cdot; \rho_t), v_0 \big\rangle_{\rho_t} \\
&= \eta \cdot \int_0^1 \partial_s \big\langle g(\cdot; \beta_s), v_s \big\rangle_{\beta_s} \, \mathrm{d}s - \eta \cdot \big\langle g(\cdot; \rho^*), v_1 \big\rangle_{\rho^*} \\
&= \eta \cdot \underbrace{\int_0^1 \big\langle \partial_s g(\cdot; \beta_s), v_s \big\rangle_{\beta_s} \, \mathrm{d}s}_{\text{(i)}} + \eta \cdot \underbrace{\int_0^1 \int \big\langle g(\theta; \beta_s), \partial_s (v_s \cdot \beta_s)(\theta) \big\rangle \, \mathrm{d}\theta \, \mathrm{d}s}_{\text{(ii)}},
\end{aligned}
$$
(C.13)

where the last equality follows from (ii) of Lemma 5.1.

For term (i) of (C.13), following from (C.4) of Lemma C.1, we have that

$$
\begin{aligned}
\int_0^1 \big\langle \partial_s g(\cdot; \beta_s), v_s \big\rangle_{\beta_s} \, \mathrm{d}s &\le -(1 - \gamma) \cdot \mathbb{E}_{\mathcal{D}}\Big[\big(Q(x; \beta_0) - Q(x; \beta_1)\big)^2\Big] \\
&= -(1 - \gamma) \cdot \mathbb{E}_{\mathcal{D}}\Big[\big(Q(x; \rho_t) - Q^*(x)\big)^2\Big].
\end{aligned}
$$
(C.14)

For term (ii) of (C.14), we have that

$$
\begin{aligned}
\int \Big|\big\langle g(\theta; \beta_s), \partial_s (v_s \cdot \beta_s)(\theta) \big\rangle\Big| \, \mathrm{d}\theta &= \int \Big|\big\langle \nabla_\theta g(\theta; \beta_s), \beta_s(\theta) \cdot v_s(\theta) \otimes v_s(\theta) \big\rangle\Big| \, \mathrm{d}\theta \\
&\le \sup_{\theta \in \mathbb{R}^D} \big\|\nabla_\theta g(\theta; \beta_s)\big\|_{\mathrm{F}} \cdot \|v_s\|_{\beta_s}^2,
\end{aligned}
$$

where the equality follows from integration by parts and Lemma E.4. Since $\beta$ is the geodesic connecting $\rho_t$ and $\rho^*$, (2.7) implies that $\|v_s\|_{\beta_s}^2 = \mathcal{W}_2(\beta_0, \beta_1)^2 = \mathcal{W}_2(\rho_t, \rho^*)^2$ for any $s \in [0, 1]$. Applying (C.8) of Lemma C.2, we have that

$$
\begin{aligned}
\int \Big|\big\langle g(\theta; \beta_s), \partial_s (v_s \cdot \beta_s)(\theta) \big\rangle\Big| \, \mathrm{d}\theta &\le \alpha \cdot B_2 \cdot \big(2\alpha \cdot B_1 \cdot \mathcal{W}_2(\rho_t, \rho_0) + B_r\big) \cdot \mathcal{W}_2(\rho_t, \rho^*)^2 \\
&\le 4\alpha \cdot B_2 \cdot \big(6\alpha \cdot B_1 \cdot \mathcal{W}_2(\rho_0, \rho^*) + B_r\big) \cdot \mathcal{W}_2(\rho_0, \rho^*)^2,
\end{aligned}
$$
(C.15)

where the last inequality follows from the condition of Lemma 5.2 that $\mathcal{W}_2(\rho_t, \rho^*) \le 2\mathcal{W}_2(\rho_0, \rho^*)$ and the fact that $\mathcal{W}_2(\rho_t, \rho_0) \le \mathcal{W}_2(\rho_t, \rho^*) + \mathcal{W}_2(\rho_0, \rho^*)$. Then, applying (iii) of Lemma 5.1 to (C.15), we have that

$$
\begin{aligned}
\int_0^1 \int \Big|\big\langle g(\theta; \beta_s), \partial_s (v_s \cdot \beta_s)(\theta) \big\rangle\Big| \, \mathrm{d}\theta \, \mathrm{d}s &\le 4\alpha^{-1} \cdot B_2 \cdot \bar{D}^2 \cdot (6B_1 \cdot \bar{D} + B_r) \\
&= C_* \cdot \alpha^{-1},
\end{aligned}
$$
(C.16)

where $C_* > 0$ is a constant depending on $\bar{D}$, $B_1$, $B_2$, and $B_r$.

Finally, plugging (C.14) and (C.16) into (C.13), we have that

$$\frac{\mathrm{d}}{\mathrm{d}t} \frac{\mathcal{W}_2(\rho_t, \rho^*)^2}{2} \le -(1 - \gamma) \cdot \eta \cdot \mathbb{E}_{\mathcal{D}}\Big[\big(Q(x; \rho_t) - Q^*(x)\big)^2\Big] + C_* \cdot \alpha^{-1} \cdot \eta,$$

which completes the proof of Lemma 5.2. $\qquad\square$

# D Mean-Field Limit of Neural Networks

In this section, we prove Proposition 3.1, whose formal version is presented as follows. Recall that $\rho_t$ is the PDE solution in (3.4) and $\widehat{\rho}_k = m^{-1} \cdot \sum_{i=1}^m \theta_i(k)$ is the empirical distribution of $\theta^{(m)}(k) = \{\theta_i(k)\}_{i=1}^m$. Note that we omit the dependence of $\widehat{\rho}_k$ on $m$ and $\epsilon$ for notational simplicity.

**Proposition D.1** (Formal Version of Proposition 3.1). Let $f : \mathbb{R}^D \to \mathbb{R}$ be any continuous function such that $\|f\|_\infty \leq 1$ and $\mathrm{Lip}(f) \leq 1$. Under Assumptions 4.1 and 4.2, it holds that

$$\sup_{\substack{k \leq T/\epsilon \\ (k \in \mathbb{N})}} \left| \int f(\theta)\, \mathrm{d}\rho_{k\epsilon}(\theta) - \int f(\theta)\, \mathrm{d}\widehat{\rho}_k(\theta) \right|$$

$$\leq B \cdot e^{BT} \cdot \left( \sqrt{\log(m/\delta)/m} + \sqrt{\epsilon \cdot \big( D + \log(m/\delta) \big)} \right)$$

with probability at least $1 - \delta$. Here $B$ is a constant that depends on $\alpha, \eta, \gamma, B_r$, and $B_j$ ($j \in \{0, 1, 2\}$).

The proof of Proposition D.1 is based on [6, 53, 54], which utilizes the propagation of chaos [66]. Recall that $g(\cdot; \rho)$ is a vector field defined as follows,

$$g(\theta; \rho) = -\alpha \cdot \mathbb{E}_{\widetilde{\mathcal{D}}}\Big[ \big( Q(x; \rho) - r - \gamma \cdot Q(x'; \rho) \big) \cdot \nabla_\theta \sigma(x; \theta) \Big].$$

Correspondingly, we define the finite-width and stochastic counterparts of $g(\theta; \rho)$ as follows,

$$\widehat{g}(\theta; \theta^{(m)}) = -\alpha \cdot \mathbb{E}_{\widetilde{\mathcal{D}}}\Big[ \big( \widehat{Q}(x; \theta^{(m)}) - r - \gamma \cdot \widehat{Q}(x'; \theta^{(m)}) \big) \cdot \nabla_\theta \sigma(x; \theta) \Big], \tag{D.1}$$

$$\widehat{G}_k(\theta; \theta^{(m)}) = -\alpha \cdot \big( \widehat{Q}(x_k; \theta^{(m)}) - r_k - \gamma \cdot \widehat{Q}(x_k'; \theta^{(m)}) \big) \cdot \nabla_\theta \sigma(x_k; \theta), \tag{D.2}$$

where $(x_k, r_k, x_k') \sim \widetilde{\mathcal{D}}$. Following from [6, 53], we consider the following four dynamics.

- **Temporal-difference (TD).** We consider the following TD dynamics $\theta^{(m)}(k)$, where $k \in \mathbb{N}$, with $\theta_i(0) \overset{\text{i.i.d.}}{\sim} \rho_0$ ($i \in [m]$) as its initialization,

$$\theta_i(k+1) = \theta_i(k) - \eta\epsilon \cdot \alpha \cdot \Big( \widehat{Q}\big(x_k; \theta^{(m)}(k)\big) - r_k - \gamma \cdot \widehat{Q}\big(x_k'; \theta^{(m)}(k)\big) \Big) \cdot \nabla_\theta \sigma\big(x_k; \theta_i(k)\big)$$

$$= \theta_i(k) + \eta\epsilon \cdot \widehat{G}_k\big(\theta_i(k); \theta^{(m)}(k)\big), \tag{D.3}$$

where $(x_k, r_k, x_k') \sim \widetilde{\mathcal{D}}$. Note that this definition is equivalent to (2.3).

- **Expected temporal-difference (ETD).** We consider the following expected TD dynamics $\breve{\theta}^{(m)}(k)$, where $k \in \mathbb{N}$, with $\breve{\theta}_i(0) = \theta_i(0)$ ($i \in [m]$) as its initialization,

$$\breve{\theta}_i(k+1) = \breve{\theta}_i(k) - \eta\epsilon \cdot \alpha \cdot \mathbb{E}_{\widetilde{\mathcal{D}}}\Big[ \big( \widehat{Q}\big(x; \breve{\theta}^{(m)}(k)\big) - r - \gamma \cdot \widehat{Q}\big(x'; \breve{\theta}^{(m)}(k)\big) \big) \cdot \nabla_\theta \sigma\big(x; \breve{\theta}_i(k)\big) \Big]$$

$$= \breve{\theta}_i(k) + \eta\epsilon \cdot \widehat{g}\big(\breve{\theta}_i(k); \breve{\theta}^{(m)}(k)\big). \tag{D.4}$$

- **Continuous-time temporal-difference (CTTD).** We consider the following continuous-time TD dynamics $\widetilde{\theta}^{(m)}(t)$, where $t \in \mathbb{R}_+$, with $\widetilde{\theta}_i(0) = \theta_i(0)$ ($i \in [m]$) as its initialization,

$$\frac{\mathrm{d}}{\mathrm{d}t} \widetilde{\theta}_i(t) = -\eta \cdot \alpha \cdot \mathbb{E}_{\widetilde{\mathcal{D}}}\Big[ \big( \widehat{Q}\big(x; \widetilde{\theta}^{(m)}(t)\big) - r - \gamma \cdot \widehat{Q}\big(x'; \widetilde{\theta}^{(m)}(t)\big) \big) \cdot \nabla_\theta \sigma\big(x; \widetilde{\theta}_i(t)\big) \Big]$$

$$= \eta \cdot \widehat{g}\big(\widetilde{\theta}_i(t); \widetilde{\theta}^{(m)}(t)\big). \tag{D.5}$$

- **Ideal particle (IP).** We consider the following ideal particle dynamics $\bar{\theta}^{(m)}(t)$, where $t \in \mathbb{R}_+$, with $\bar{\theta}_i(0) = \theta_i(0)$ ($i \in [m]$) as its initialization,

$$\frac{\mathrm{d}}{\mathrm{d}t} \bar{\theta}_i(t) = -\eta \cdot \alpha \cdot \mathbb{E}_{\widetilde{\mathcal{D}}}\Big[ \big( Q(x; \rho_t) - r - \gamma \cdot Q(x'; \rho_t) \big) \cdot \nabla_\theta \sigma\big(x; \bar{\theta}_i(t)\big) \Big]$$

$$= \eta \cdot g\big(\bar{\theta}_i(t); \rho_t\big), \tag{D.6}$$

where $\rho_t$ is the PDE solution in (3.4).

We aim to prove that $\widehat{\rho}_k = m^{-1} \cdot \sum_{i=1}^m \delta_{\theta_i(k)}$ weakly converges to $\rho_{k\epsilon}$. For any continuous function $f : \mathbb{R}^D \to \mathbb{R}$ such that $\|f\|_\infty \leq 1$ and $\mathrm{Lip}(f) \leq 1$, we use the IP, CTTD, and ETD dynamics as the interpolating dynamics,

$$\overbrace{\left| \int f(\theta) \, \mathrm{d}\rho_{k\epsilon}(\theta) - \int f(\theta) \, \mathrm{d}\widehat{\rho}_k(\theta) \right|}^{\text{PDE} - \text{TD}}$$

$$\leq \left| \int f(\theta) \, \mathrm{d}\rho_{k\epsilon}(\theta) - m^{-1} \cdot \sum_{i=1}^m f\big(\bar{\theta}_i(k\epsilon)\big) \right| + \left| m^{-1} \cdot \sum_{i=1}^m f\big(\bar{\theta}_i(k\epsilon)\big) - m^{-1} \cdot \sum_{i=1}^m f\big(\widetilde{\theta}_i(k\epsilon)\big) \right|$$

$$+ \left| m^{-1} \cdot \sum_{i=1}^m f\big(\widetilde{\theta}_i(k\epsilon)\big) - m^{-1} \cdot \sum_{i=1}^m f\big(\check{\theta}_i(k)\big) \right| + \left| m^{-1} \cdot \sum_{i=1}^m f\big(\check{\theta}_i(k)\big) - m^{-1} \cdot \sum_{i=1}^m f\big(\theta_i(k)\big) \right|$$

$$\leq \underbrace{\left| \int f(\theta) \, \mathrm{d}\rho_{k\epsilon}(\theta) - m^{-1} \cdot \sum_{i=1}^m f\big(\bar{\theta}_i(k\epsilon)\big) \right|}_{\text{PDE} - \text{IP}} + \underbrace{\big\| \bar{\theta}^{(m)}(k\epsilon) - \widetilde{\theta}^{(m)}(k\epsilon) \big\|_{(m)}}_{\text{IP} - \text{CTTD}}$$

$$+ \underbrace{\big\| \widetilde{\theta}^{(m)}(k\epsilon) - \check{\theta}^{(m)}(k) \big\|_{(m)}}_{\text{CTTD} - \text{ETD}} + \underbrace{\big\| \check{\theta}^{(m)}(k) - \theta^{(m)}(k) \big\|_{(m)}}_{\text{ETD} - \text{TD}}, \tag{D.7}$$

where the last inequality follows from the the fact that $\mathrm{Lip}(f) \leq 1$. Here the norm $\|\cdot\|_{(m)}$ of $\theta^{(m)} = \{\theta_i\}_{i=1}^m$ is defined as follows,

$$\|\theta^{(m)}\|_{(m)} = \sup_{i \in [m]} \|\theta_i\|. \tag{D.8}$$

In what follows, we define $B > 0$ as a constant that depends on $\alpha$, $\eta$, $\gamma$, $B_r$, and $B_j$ ($j \in \{0, 1, 2\}$), whose value varies from line to line. We establish the following lemmas to upper bound the terms on the right-hand side of (D.8).

**Lemma D.2** (Upper Bound of PDE – IP). Let $f$ be any continuous function such that $\|f\|_\infty \leq 1$ and $\mathrm{Lip}(f) \leq 1$. Under Assumptions 4.1 and 4.2, it holds for any $f$ that

$$\sup_{t \in [0,T]} \left| \int f(\theta) \, \mathrm{d}\rho_t(\theta) - m^{-1} \cdot \sum_{i=1}^m f\big(\bar{\theta}_i(t)\big) \right| \leq B \cdot \sqrt{\log(mT/\delta)/m}$$

with probability at least $1 - \delta$.

*Proof.* See §D.1.1 for a detailed proof. □

**Lemma D.3** (Upper Bound of IP – CTTD). Under Assumptions 4.1 and 4.2, it holds that

$$\sup_{t \in [0,T]} \big\| \bar{\theta}^{(m)}(t) - \widetilde{\theta}^{(m)}(t) \big\|_{(m)} \leq B \cdot e^{BT} \cdot \sqrt{\log(m/\delta)/m}$$

with probability at least $1 - \delta$.

*Proof.* See §D.1.2 for a detailed proof. □

**Lemma D.4** (Upper Bound of CTTD – ETD). Under Assumptions 4.1 and 4.2, it holds that

$$\sup_{\substack{k \leq T/\epsilon \\ (k \in \mathbb{N})}} \big\| \widetilde{\theta}^{(m)}(k\epsilon) - \check{\theta}^{(m)}(k) \big\|_{(m)} \leq B \cdot e^{BT} \cdot \epsilon.$$

*Proof.* See §D.1.3 for a detailed proof. □

**Lemma D.5** (Upper Bound of ETD – TD). Under Assumptions 4.1 and 4.2, it holds that

$$\sup_{\substack{k \leq T/\epsilon \\ (k \in \mathbb{N})}} \big\| \check{\theta}^{(m)}(k) - \theta^{(m)}(k) \big\|_{(m)} \leq B \cdot e^{BT} \cdot \sqrt{\epsilon \cdot \big(D + \log(m/\delta)\big)}$$

with probability at least $1 - \delta$

*Proof.* See §D.1.4 for a detailed proof. □

We are now ready to present the proof of Proposition D.1.

*Proof.* Plugging Lemmas D.2-D.5 into (D.7), we have that

$$\sup_{\substack{k \leq T/\epsilon \\ (k \in \mathbb{N})}} \left| \int f(\theta) \, \mathrm{d}\rho_{k\epsilon}(\theta) - \int f(\theta) \, \mathrm{d}\widehat{\rho}_k(\theta) \right|$$

$$\leq B \cdot e^{BT} \cdot \left( \sqrt{\log(m/\delta)/m} + \sqrt{\epsilon \cdot \left( D + \log(m/\delta) \right)} \right)$$

with probability at least $1 - \delta$. Thus, we complete the proof of Proposition D.1. □

## D.1 Proofs of Lemmas D.2-D.5

In this section, we present the proofs of Lemmas D.2-D.5, which are based on [6, 53, 54]. We include the required technical lemmas in §D.3. Recall that $B > 0$ is a constant that depends on $\alpha$, $\eta$, $\gamma$, $B_r$, and $B_j$ ($j \in \{0, 1, 2\}$), whose value varies from line to line.

### D.1.1 Proof of Lemma D.2

*Proof.* For the IP dynamics in (D.6), it holds that $\bar{\theta}_i(t) \sim \rho_t$ ($i \in [m]$) (Proposition 8.1.8 in [5]). Furthermore, since the randomness of $\bar{\theta}_i(t)$ comes from $\theta_i(0)$ while $\theta_i(0)$ ($i \in [m]$) are independent, we have that $\bar{\theta}_i(t) \overset{\text{i.i.d.}}{\sim} \rho_t$ ($i \in [m]$). Thus, we have that

$$\mathbb{E}_{\rho_t} \left[ m^{-1} \cdot \sum_{i=1}^m f(\bar{\theta}_i(t)) \right] = \int f(\theta) \, \mathrm{d}\rho_t(\theta).$$

Let $\theta^{1,(m)} = \{\theta_1, \ldots, \theta_i^1, \ldots, \theta_m\}$ and $\theta^{2,(m)} = \{\theta_1, \ldots, \theta_i^2, \ldots, \theta_m\}$ be two sets that only differ in the $i$-th element. Then, by the condition of Lemma D.2 that $\|f\|_\infty \leq 1$, we have that

$$\left| m^{-1} \cdot \sum_{j=1}^m f(\theta_j^1) - m^{-1} \cdot \sum_{j=1}^m f(\theta_j^2) \right| = m^{-1} \cdot \left| f(\theta_i^1) - f(\theta_i^2) \right| \leq 2/m.$$

Applying McDiarmid's inequality [70], we have for a fixed $t \in [0, T]$ that

$$\mathbb{P} \left( \left| m^{-1} \cdot \sum_{i=1}^m f(\bar{\theta}_i(t)) - \int f(\theta) \, \mathrm{d}\rho_t(\theta) \right| \geq p \right) \leq \exp(-mp^2/4). \tag{D.9}$$

Moreover, we have for any $s, t \in [0, T]$ that

$$\left| \left| m^{-1} \cdot \sum_{i=1}^m f(\bar{\theta}_i(t)) - \int f(\theta) \, \mathrm{d}\rho_t(\theta) \right| - \left| m^{-1} \cdot \sum_{i=1}^m f(\bar{\theta}_i(s)) - \int f(\theta) \, \mathrm{d}\rho_s(\theta) \right| \right|$$

$$\leq \left| m^{-1} \cdot \sum_{i=1}^m f(\bar{\theta}_i(t)) - m^{-1} \cdot \sum_{i=1}^m f(\bar{\theta}_i(s)) \right| + \left| \int f(\theta) \, \mathrm{d}\rho_t(\theta) - \int f(\theta) \, \mathrm{d}\rho_s(\theta) \right|$$

$$\leq \left\| \bar{\theta}^{(m)}(t) - \bar{\theta}^{(m)}(s) \right\|_{(m)} + \mathcal{W}_1(\rho_t, \rho_s)$$

$$\leq \left\| \bar{\theta}^{(m)}(t) - \bar{\theta}^{(m)}(s) \right\|_{(m)} + \mathcal{W}_2(\rho_t, \rho_s),$$

where the second inequality follows from the fact that $\mathrm{Lip}(f) \leq 1$ and (C.9). Applying (D.38) and (D.40) of Lemma D.8, we have for any $s, t \in [0, T]$ that

$$\left| \left| m^{-1} \cdot \sum_{i=1}^m f(\bar{\theta}_i(t)) - \int f(\theta) \, \mathrm{d}\rho_t(\theta) \right| - \left| m^{-1} \cdot \sum_{i=1}^m f(\bar{\theta}_i(s)) - \int f(\theta) \, \mathrm{d}\rho_s(\theta) \right| \right| \leq B \cdot |t - s|.$$

Applying the union bound to (D.9) for $t \in \iota \cdot \{0, 1, \ldots, \lfloor T/\iota \rfloor\}$, we have that

$$\mathbb{P}\left(\sup_{t \in [0,T]} \left| m^{-1} \cdot \sum_{i=1}^{m} f(\bar{\theta}_i(t)) - \int f(\theta) \, \mathrm{d}\rho_t(\theta) \right| \geq p + B \cdot \iota \right) \leq (T/\iota + 1) \cdot \exp(-mp^2/4).$$

Setting $\iota = m^{-1/2}$ and $p = B \cdot \sqrt{\log(mT/\delta)/m}$, we have that

$$\sup_{t \in [0,T]} \left| m^{-1} \cdot \sum_{i=1}^{m} f(\bar{\theta}_i(t)) - \int f(\theta) \, \mathrm{d}\rho_t(\theta) \right| \leq B \cdot \sqrt{\log(mT/\delta)/m}$$

with probability at least $1 - \delta$. Thus, we complete the proof of Lemma D.2. $\qquad\square$

### D.1.2   Proof of Lemma D.3

*Proof.* Recall that $g$ and $\widehat{g}$ are defined in (3.5) and (D.1), respectively, that is,

$$g(\theta; \rho) = -\alpha \cdot \mathbb{E}_{\widetilde{\mathcal{D}}}\Big[\big(Q(x; \rho) - r - \gamma \cdot Q(x'; \rho)\big) \cdot \nabla_\theta \sigma(x; \theta)\Big],$$

$$\widehat{g}(\theta; \theta^{(m)}) = -\alpha \cdot \mathbb{E}_{\widetilde{\mathcal{D}}}\Big[\big(\widehat{Q}(x; \theta^{(m)}) - r - \gamma \cdot \widehat{Q}(x'; \theta^{(m)})\big) \cdot \nabla_\theta \sigma(x; \theta)\Big].$$

Following from the definition of $\widetilde{\theta}_i(t)$ and $\bar{\theta}_i(t)$ in (D.5) and (D.6), respectively, we have for any $i \in [m]$ and $t \in [0, T]$ that

$$\big\|\bar{\theta}_i(t) - \widetilde{\theta}_i(t)\big\|$$
$$\leq \int_0^t \left\| \frac{\mathrm{d}\widetilde{\theta}_i(s)}{\mathrm{d}s} - \frac{\mathrm{d}\bar{\theta}_i(s)}{\mathrm{d}s} \right\| \mathrm{d}s$$
$$= \eta \cdot \int_0^t \left\| \widehat{g}\big(\widetilde{\theta}_i(s); \widetilde{\theta}^{(m)}(s)\big) - g\big(\bar{\theta}_i(s); \rho_s\big) \right\| \mathrm{d}s$$
$$\leq \eta \cdot \int_0^t \left\| \widehat{g}\big(\widetilde{\theta}_i(s); \widetilde{\theta}^{(m)}(s)\big) - \widehat{g}\big(\bar{\theta}_i(s); \bar{\theta}^{(m)}(s)\big) \right\| \mathrm{d}s + \eta \cdot \int_0^t \left\| \widehat{g}\big(\bar{\theta}_i(s); \bar{\theta}^{(m)}(s)\big) - g\big(\bar{\theta}_i(s); \rho_s\big) \right\| \mathrm{d}s$$
$$\leq B \cdot \int_0^t \big\|\widetilde{\theta}^{(m)}(s) - \bar{\theta}^{(m)}(s)\big\|_{(m)} \, \mathrm{d}s + \eta \cdot \int_0^t \left\| \widehat{g}\big(\bar{\theta}_i(s); \bar{\theta}^{(m)}(s)\big) - g\big(\bar{\theta}_i(s); \rho_s\big) \right\| \mathrm{d}s, \quad \text{(D.10)}$$

where the last inequality follows from (D.35) of Lemma D.7. We now upper bound the second term on the right-hand side of (D.10). Following from the definition of $\widehat{Q}$, $Q$, and $\widehat{g}$ in (3.1), (3.2), and (D.1), respectively, we have for any $s \in [0, T]$ and $i \in [m]$ that

$$\left\| \widehat{g}\big(\bar{\theta}_i(s); \bar{\theta}^{(m)}(s)\big) - g\big(\bar{\theta}_i(s); \rho_s\big) \right\| = \alpha^2 \cdot \left\| m^{-1} \cdot \sum_{j=1}^{m} Z_i^j(s) \right\|, \quad \text{(D.11)}$$

where

$$Z_i^j(s) = \mathbb{E}_{\widetilde{\mathcal{D}}}\left[\left(\sigma\big(x; \bar{\theta}_j(s)\big) - \int \sigma(x; \theta) \, \mathrm{d}\rho_s(\theta) - \gamma \cdot \sigma\big(x'; \bar{\theta}_j(s)\big) + \gamma \cdot \int \sigma(x'; \theta) \, \mathrm{d}\rho_s(\theta)\right) \cdot \nabla_\theta \sigma\big(x; \bar{\theta}_i(s)\big)\right].$$

Following from Assumptions 4.1 and 4.2, we have that $\|Z_i^j(s)\| \leq B$. When $i \neq j$, following from the fact that $\bar{\theta}_i(s) \overset{\text{i.i.d.}}{\sim} \rho_s$ $(i \in [m])$, it holds that $\mathbb{E}[Z_i^j(s) \mid \bar{\theta}_i(s)] = 0$. Following from Lemma D.9, we have for fixed $s \in [0, T]$ and $i \in [m]$ that

$$\mathbb{P}\left(\left\| m^{-1} \cdot \sum_{j \neq i} Z_i^j(s) \right\| \geq B \cdot (m^{-1/2} + p)\right) = \mathbb{E}\left[\mathbb{P}\left(\left\| m^{-1} \cdot \sum_{j \neq i} Z_i^j(s) \right\| \geq B \cdot (m^{-1/2} + p) \, \Big| \, \bar{\theta}_i(s)\right)\right]$$
$$\leq \exp(-mp^2). \quad \text{(D.12)}$$

By (C.9), we have that

$$\sup_{x \in \mathcal{X}} \left| \int \sigma(x; \theta) \, \mathrm{d}\rho_s(\theta) - \int \sigma(x; \theta) \, \mathrm{d}\rho_t(\theta) \right| \leq B \cdot \mathcal{W}_1(\rho_s, \rho_t) \leq B \cdot \mathcal{W}_2(\rho_s, \rho_t) \leq B \cdot |s - t|,$$

where the last inequality follows from (D.40) of Lemma D.8. Thus, following from Assumptions 4.1 and 4.2, Lemma D.8, and the fact that $\mathrm{Lip}(fg) \leq \|f\|_\infty \cdot \mathrm{Lip}(g) + \|g\|_\infty \cdot \mathrm{Lip}(f)$ for any functions $f$ and $g$, we have for any $s, t \in [0, T]$ that

$$\left| \left\| m^{-1} \cdot \sum_{j \neq i} Z_i^j(s) \right\| - \left\| m^{-1} \cdot \sum_{j \neq i} Z_i^j(t) \right\| \right| \leq B \cdot |t - s|.$$

Applying the union bound to (D.12) for $i \in [m]$ and $t \in \iota \cdot \{0, 1, \ldots, \lfloor T/\iota \rfloor\}$, we have that

$$\mathbb{P}\left( \sup_{\substack{i \in [m], \\ s \in [0,T]}} \left\| m^{-1} \cdot \sum_{j \neq i} Z_i^j(s) \right\| \geq B \cdot (m^{-1/2} + p) + B\iota \right) \leq m \cdot (T/\iota + 1) \cdot \exp(-mp^2).$$

Setting $\iota = m^{-1/2}$ and $p = B \cdot \sqrt{\log(mT/\delta)/m}$, we have that

$$\sup_{\substack{i \in [m], \\ s \in [0,T]}} \left\| m^{-1} \cdot \sum_{j \neq i} Z_i^j(s) \right\| \leq B \cdot \sqrt{\log(mT/\delta)/m} \tag{D.13}$$

with probability at least $1 - \delta$. When $i = j$, it holds that $\|m^{-1} \cdot Z_i^i(s)\| \leq B/m$ in (D.11), which follows from Assumptions 4.1 and 4.2. Thus, plugging (D.13) into (D.11), we have that

$$\sup_{\substack{i \in [m], \\ s \in [0,T]}} \left\| \widehat{g}\big(\bar{\theta}_i(s); \bar{\theta}^{(m)}(s)\big) - g\big(\bar{\theta}_i(s); \rho_s\big) \right\| \leq \sup_{\substack{i \in [m], \\ s \in [0,T]}} \alpha^2 \cdot \left( \left\| m^{-1} \cdot Z_i^i(s) \right\| + \left\| m^{-1} \cdot \sum_{j \neq i} Z_i^j(s) \right\| \right)$$

$$\leq B \cdot \sqrt{\log(mT/\delta)/m} \tag{D.14}$$

with probability at least $1 - \delta$.

Conditioning on the event in (D.14), we obtain from (D.10) that

$$\left\| \widetilde{\theta}^{(m)}(t) - \bar{\theta}^{(m)}(t) \right\|_{(m)} \leq B \cdot \int_0^t \left\| \widetilde{\theta}^{(m)}(s) - \bar{\theta}^{(m)}(s) \right\|_{(m)} \mathrm{d}s + BT \cdot \sqrt{\log(mT/\delta)/m}$$

for any $t \in [0, T]$. Following from Gronwall's Lemma [41], we have that

$$\left\| \widetilde{\theta}^{(m)}(t) - \bar{\theta}^{(m)}(t) \right\|_{(m)} \leq B \cdot e^{Bt} \cdot BT \cdot \sqrt{\log(mT/\delta)/m}$$

$$\leq B \cdot e^{BT} \cdot \sqrt{\log(m/\delta)/m}, \qquad \forall t \in [0, T]$$

with probability at least $1 - \delta$. Here the last inequality holds since we allow the value of $B$ to vary from line to line. Thus, we complete the proof of Lemma D.3 ∎

### D.1.3   Proof of Lemma D.4

*Proof.* By the definition of $\widehat{g}$, $\breve{\theta}_i(t)$, and $\widetilde{\theta}_i(t)$ in (D.1), (D.4), and (D.5), respectively, it holds that

$$\left\| \widetilde{\theta}_i(k\epsilon) - \breve{\theta}_i(k) \right\| \leq \eta \cdot \int_0^{k\epsilon} \left\| \widehat{g}\big(\widetilde{\theta}_i(s); \widetilde{\theta}^{(m)}(s)\big) - \widehat{g}\big(\breve{\theta}_i(\lfloor s/\epsilon \rfloor); \breve{\theta}^{(m)}(\lfloor s/\epsilon \rfloor)\big) \right\| \mathrm{d}s$$

$$\leq \eta \cdot \int_0^{k\epsilon} \left\| \widehat{g}\big(\widetilde{\theta}_i(s); \widetilde{\theta}^{(m)}(s)\big) - \widehat{g}\big(\widetilde{\theta}_i(\lfloor s/\epsilon \rfloor \cdot \epsilon); \widetilde{\theta}^{(m)}(\lfloor s/\epsilon \rfloor \cdot \epsilon)\big) \right\| \mathrm{d}s$$

$$+ \eta \cdot \sum_{\ell=0}^{k-1} \left\| \widehat{g}\big(\widetilde{\theta}_i(\ell\epsilon); \widetilde{\theta}^{(m)}(\ell\epsilon)\big) - \widehat{g}\big(\breve{\theta}_i(\ell); \breve{\theta}^{(m)}(\ell)\big) \right\|$$

$$\leq B \cdot k \cdot \epsilon^2 + B \cdot \sum_{\ell=0}^{k-1} \left\| \widetilde{\theta}^{(m)}(\ell\epsilon) - \breve{\theta}^{(m)}(\ell) \right\|_{(m)},$$

where the last inequality follows from (D.35) of Lemma D.7 and (D.39) of Lemma D.8. Following from the definition of $\|\cdot\|_{(m)}$ in (D.8), it holds for any $k \leq T/\epsilon$ ($k \in \mathbb{N}$) that

$$\left\| \widetilde{\theta}^{(m)}(k\epsilon) - \breve{\theta}^{(m)}(k) \right\|_{(m)} \leq B \cdot T \cdot \epsilon + B \cdot \sum_{\ell=0}^{k-1} \left\| \widetilde{\theta}^{(m)}(\ell\epsilon) - \breve{\theta}^{(m)}(\ell) \right\|_{(m)}.$$

Following from the discrete Gronwall's lemma [41], we have that

$$\sup_{\substack{k \leq T/\epsilon \\ (k \in \mathbb{N})}} \big\| \widetilde{\theta}^{(m)}(k\epsilon) - \breve{\theta}^{(m)}(k) \big\|_{(m)} \leq B^2 \cdot T \cdot \epsilon \cdot e^{BT} \leq B \cdot e^{BT} \cdot \epsilon,$$

where the last inequality holds since we allow the value of $B$ to vary from line to line. Thus, we complete the proof of Lemma D.4. $\qquad\square$

### D.1.4 Proof of Lemma D.5

*Proof.* Let $\mathcal{G}_k = \sigma(\theta^{(m)}(0), z_0, \ldots, z_k)$ be the $\sigma$-algebra generated by $\theta^{(m)}(0)$ and $z_\ell = (x_\ell, r_\ell, x'_\ell)$ $(\ell \leq k)$. Recall that $\widehat{g}$ and $\widehat{G}_k$ are defined in (D.1) and (D.2), respectively. We have for any $i \in [m]$ and $k \in \mathbb{N}_+$ that

$$\mathbb{E}\Big[\widehat{G}_k\big(\theta_i(k); \theta^{(m)}(k)\big) \,\Big|\, \mathcal{G}_{k-1}\Big] = \widehat{g}\big(\theta_i(k); \theta^{(m)}(k)\big).$$

Recall that $\theta^{(m)}(k)$ and $\breve{\theta}^{(m)}(k)$ are the TD and ETD dynamics defined in (D.3) and (D.4), respectively. Thus, we have for any $i \in [m]$ and $k \in \mathbb{N}_+$ that

$$
\begin{aligned}
\big\| \breve{\theta}_i(k) - \theta_i(k) \big\| &= \eta\epsilon \cdot \bigg\| \sum_{\ell=0}^{k-1} \widehat{G}_\ell\big(\theta_i(\ell); \theta^{(m)}(\ell)\big) - \sum_{\ell=0}^{k-1} \widehat{g}\big(\breve{\theta}_i(\ell); \breve{\theta}^{(m)}(\ell)\big) \bigg\| \\
&\leq \eta\epsilon \cdot \bigg\| \sum_{\ell=0}^{k-1} X_i(\ell) \bigg\| + \eta\epsilon \cdot \sum_{\ell=0}^{k-1} \big\| \widehat{g}\big(\breve{\theta}_i(\ell); \breve{\theta}^{(m)}(\ell)\big) - \widehat{g}\big(\theta_i(\ell); \theta^{(m)}(\ell)\big) \big\| \\
&\leq \eta\epsilon \cdot \big\| A_i(k) \big\| + B\epsilon \cdot \sum_{\ell=0}^{k-1} \big\| \breve{\theta}^{(m)}(\ell) - \theta^{(m)}(\ell) \big\|_{(m)}, \quad\quad \text{(D.15)}
\end{aligned}
$$

where the last inequality follows from (D.35) of Lemma D.7, and $X_i(\ell)$ and $A_i(k)$ are defined as

$$
\begin{aligned}
X_i(0) &= 0, \\
X_i(\ell) &= \widehat{G}_\ell\big(\theta_i(\ell); \theta^{(m)}(\ell)\big) - \mathbb{E}\Big[\widehat{G}_\ell\big(\theta_i(\ell); \theta^{(m)}(\ell)\big) \,\Big|\, \mathcal{G}_{\ell-1}\Big] \quad \forall \ell \geq 1, \\
A_i(k) &= \sum_{\ell=0}^{k-1} X_i(\ell).
\end{aligned}
$$

Following from (D.32) of Lemma D.7, we have that $\|X_i(\ell)\| \leq B$. Thus, the stochastic process $\{A_i(k)\}_{k \in \mathbb{N}_+}$ is a martingale with $\|A_i(k) - A_i(k-1)\| \leq B$. Applying Lemma D.10, we have that

$$\mathbb{P}\Big( \max_{\substack{k \leq T/\epsilon \\ (k \in \mathbb{N}_+)}} \big\| A_i(k) \big\| \geq B \cdot \sqrt{T/\epsilon} \cdot (\sqrt{D} + p) \Big) \leq \exp(-p^2). \quad\quad \text{(D.16)}$$

Applying the union bound to (D.16) for $i \in [m]$, we have that

$$\mathbb{P}\Big( \max_{\substack{i \in [m], \\ k \leq T/\epsilon \, (k \in \mathbb{N}_+)}} \big\| A_i(k) \big\| \geq B \cdot \sqrt{T/\epsilon} \cdot (\sqrt{D} + p) \Big) \leq m \cdot \exp(-p^2).$$

By setting $p = \sqrt{\log(m/\delta)}$, we have that

$$\big\| A_i(k) \big\| \leq B \cdot \sqrt{T/\epsilon} \cdot \big(\sqrt{D} + \sqrt{\log(m/\delta)}\big), \quad \forall i \in [m], k \leq T/\epsilon \ (k \in \mathbb{N}_+) \quad\quad \text{(D.17)}$$

with probability at least $1 - \delta$. By (D.15) and (D.17), we have that

$$
\begin{aligned}
&\big\| \breve{\theta}^{(m)}(k) - \theta^{(m)}(k) \big\|_{(m)} \\
&\leq B \cdot \sqrt{T\epsilon} \cdot \big(\sqrt{D} + \sqrt{\log(m/\delta)}\big) + B\epsilon \cdot \sum_{\ell=0}^{k-1} \big\| \breve{\theta}^{(m)}(\ell) - \theta^{(m)}(\ell) \big\|_{(m)}, \quad \forall k \leq T/\epsilon \ (k \in \mathbb{N})
\end{aligned}
$$

with probability at least $1 - \delta$. Applying the discrete Gronwall's Lemma [41], we have that

$$
\begin{aligned}
\big\| \breve{\theta}^{(m)}(k) - \theta^{(m)}(k) \big\|_{(m)} &\leq B \cdot e^{BT} \cdot B \cdot \sqrt{T\epsilon} \cdot \big(\sqrt{D} + \sqrt{\log(m/\delta)}\big) \\
&\leq B \cdot e^{BT} \cdot \sqrt{\epsilon \cdot \big(D + \log(m/\delta)\big)}, \quad \forall k \leq T/\epsilon \ (k \in \mathbb{N})
\end{aligned}
$$

with probability at least $1 - \delta$. Here the last inequality holds since we allow the value of $B$ to vary from line to line. Thus, we complete the proof of Lemma D.5. $\qquad\square$

## D.2 Proof of Corollary 4.4

The proof of Corollary 4.4 follows from Theorem 4.3 and the following lemma, which characterizes the error of approximating the TD dynamics $\theta^{(m)}(k)$ in (3.3) using the PDE solution $\rho_t$ in (3.4).

**Lemma D.6.** Let $B$ be a constant that depends on $\alpha$, $\eta$, $\gamma$, $B_0$, $B_1$, and $B_2$. Under Assumptions 4.1 and 4.2, it holds for any $k \leq T/\epsilon$ ($k \in \mathbb{N}$) that

$$\mathbb{E}_{x \sim \mathcal{D}}\left[\left(\widehat{Q}\big(x; \theta^{(m)}(k)\big) - Q^*(x)\right)^2\right]$$
$$\leq \mathbb{E}_{x \sim \mathcal{D}}\left[\left(Q(x; \rho_{k\epsilon}) - Q^*(x)\right)^2\right] + B \cdot e^{BT} \cdot \left(\sqrt{m^{-1} \cdot \log(m/\delta)} + \sqrt{\epsilon \cdot \big(D + \log(m/\delta)\big)}\right)$$

with probability at least $1 - \delta$.

*Proof.* Recall that $\widehat{Q}$ and $Q(\cdot; \rho)$ are defined in (3.1) and (3.2), respectively. For notational simplicity, we denote the optimality gaps for $\theta^{(m)} = \{\theta_i\}_{i=1}^m$ and $\rho \in \mathscr{P}_2(\mathbb{R}^D)$ by

$$L(\theta^{(m)}) = \mathbb{E}_{\mathcal{D}}\left[\left(\widehat{Q}(x; \theta^{(m)}) - Q^*(x)\right)^2\right], \tag{D.18}$$

$$\bar{L}(\rho) = \mathbb{E}_{\mathcal{D}}\left[\left(Q(x; \rho) - Q^*(x)\right)^2\right]. \tag{D.19}$$

Recall that $\theta^{(m)}(k)$, $\bar{\theta}^{(m)}(k\epsilon)$, and $\rho_t$ are the TD dynamics, the IP dynamics, and the PDE solution defined in (D.3), (D.6), and (3.4), respectively. It holds for any $k \in \mathbb{N}$ that

$$\left|L\big(\theta^{(m)}(k)\big) - \bar{L}(\rho_{k\epsilon})\right| \leq \underbrace{\left|L\big(\theta^{(m)}(k)\big) - L\big(\bar{\theta}^{(m)}(k\epsilon)\big)\right|}_{\text{(i)}} + \underbrace{\left|L\big(\bar{\theta}^{(m)}(k\epsilon)\big) - \bar{L}(\rho_{k\epsilon})\right|}_{\text{(ii)}}. \tag{D.20}$$

In what follows, we upper bound the two terms on the right-hand side of (D.20).

**Upper bounding term (i) of** (D.20). Following from the definition of $L$ in (D.18), it holds for any $k \in \mathbb{N}$ that

$$\left|L\big(\theta^{(m)}(k)\big) - L\big(\bar{\theta}^{(m)}(k\epsilon)\big)\right|$$
$$= \left|\mathbb{E}_{\mathcal{D}}\left[\left(\widehat{Q}\big(x; \theta^{(m)}(k)\big) + \widehat{Q}\big(x; \bar{\theta}_i(k\epsilon)\big) - 2Q^*(x)\right) \cdot \left(\widehat{Q}\big(x; \theta^{(m)}(k)\big) - \widehat{Q}\big(x; \bar{\theta}_i(k\epsilon)\big)\right)\right]\right|. \tag{D.21}$$

Following from (D.30), (D.31), and (D.36) of Lemma D.7, we have for any $k \in \mathbb{N}$ that

$$\sup_{x \in \mathcal{X}}\left|\widehat{Q}\big(x; \theta^{(m)}(k)\big) + \widehat{Q}\big(x; \bar{\theta}_i(k\epsilon)\big) - 2Q^*(x)\right| \leq B, \tag{D.22}$$

$$\sup_{x \in \mathcal{X}}\left|\widehat{Q}\big(x; \theta^{(m)}(k)\big) - \widehat{Q}\big(x; \bar{\theta}_i(k\epsilon)\big)\right| \leq B \cdot \left\|\theta^{(m)}(k) - \bar{\theta}^{(m)}(k\epsilon)\right\|_{(m)}. \tag{D.23}$$

Thus, we have that

$$\left|L\big(\theta^{(m)}(k)\big) - L\big(\bar{\theta}^{(m)}(k\epsilon)\big)\right|$$
$$\leq B \cdot \left\|\theta^{(m)}(k) - \bar{\theta}^{(m)}(k\epsilon)\right\|_{(m)}$$
$$\leq B \cdot e^{BT} \cdot \left(\sqrt{\log(m/\delta)/m} + \sqrt{\epsilon \cdot \big(D + \log(m/\delta)\big)}\right), \quad \forall k \leq T/\epsilon \ (k \in \mathbb{N}) \tag{D.24}$$

with probability at least $1 - \delta$. Here the last inequality follows from Lemmas D.3-D.5.

**Upper bounding term (ii) of** (D.20). Let $t = k\epsilon$. It holds for any $t \in [0, T]$ that

$$\left|L\big(\bar{\theta}^{(m)}(t)\big) - \bar{L}(\rho_t)\right| \leq \left|L\big(\bar{\theta}^{(m)}(t)\big) - \mathbb{E}_{\rho_t}\left[L\big(\bar{\theta}^{(m)}(t)\big)\right]\right| + \left|\mathbb{E}_{\rho_t}\left[L\big(\bar{\theta}^{(m)}(t)\big)\right] - \bar{L}(\rho_t)\right|, \tag{D.25}$$

where the expectation is with respect to $\bar{\theta}_i(t) \overset{\text{i.i.d.}}{\sim} \rho_t$ ($i \in [m]$). For the second term on the right-hand side of (D.25), following from the fact that $\mathbb{E}_{\rho_t}[\widehat{Q}(x;\bar{\theta}^{(m)}(t))] = Q(x;\rho_t)$ for any $x \in \mathcal{X}$, we have that

$$\left| \mathbb{E}_{\rho_t}\left[ L(\bar{\theta}^{(m)}(t)) \right] - \bar{L}(\rho_t) \right| = \left| \int \mathbb{E}_{\rho_t}\left[ \widehat{Q}(x;\bar{\theta}^{(m)}(t))^2 - Q(x;\rho_t)^2 \right] \mathrm{d}\mathcal{D}(x) \right|$$

$$= \left| \int \mathrm{Var}_{\rho_t}\left[ \widehat{Q}(x;\bar{\theta}^{(m)}(t)) \right] \mathrm{d}\mathcal{D}(x) \right|$$

$$\leq B/m, \tag{D.26}$$

where the inequality follows from the fact that $\|\sigma\| \leq B$ in Assumption 4.2 and the independence of $\bar{\theta}_i(t)$ ($i \in [m]$). Let $\theta^{1,(m)} = \{\theta_1, \ldots, \theta_i^1, \ldots, \theta_m\}$ and $\theta^{2,(m)} = \{\theta_1, \ldots, \theta_i^2, \ldots, \theta_m\}$ be two sets that only differ in the $i$-th element. It holds that

$$\left| L(\theta^{1,(m)}) - L(\theta^{2,(m)}) \right| \leq B \cdot m^{-1} \cdot \mathbb{E}_{\mathcal{D}}\left[ \left| \sigma(x;\theta_i^1) - \sigma(x;\theta_i^2) \right| \right] \leq B/m,$$

where the first inequality follows from (D.21) and (D.22) and the second inequality follows from Assumption 4.2. Applying McDiarmid's inequality [70], we have for a fixed $t \in [0,T]$ that

$$\mathbb{P}\left( \left| L(\bar{\theta}^{(m)}(t)) - \mathbb{E}_{\rho_t}\left[ L(\bar{\theta}^{(m)}(t)) \right] \right| \geq p \right) \leq \exp(-mp^2/B). \tag{D.27}$$

It holds for any $s, t \in [0,T]$ that

$$\left| \left| L(\bar{\theta}^{(m)}(t)) - \mathbb{E}_{\rho_t}\left[ L(\bar{\theta}^{(m)}(t)) \right] \right| - \left| L(\bar{\theta}^{(m)}(s)) - \mathbb{E}_{\rho_t}\left[ L(\bar{\theta}^{(m)}(s)) \right] \right| \right|$$

$$\leq B \cdot \left\| \bar{\theta}^{(m)}(t) - \bar{\theta}^{(m)}(s) \right\|_{(m)} \leq B \cdot |t - s|,$$

where the first inequality follows from (D.21), (D.22), and (D.23) and the second inequality follows from (D.38) of Lemma D.8. Applying the union bound to (D.27) for $t \in \iota \cdot \{0, 1, \ldots, \lfloor T/\iota \rfloor\}$, we have that

$$\mathbb{P}\left( \sup_{t \in [0,T]} \left| L(\bar{\theta}^{(m)}(t)) - \mathbb{E}_{\rho_t}\left[ L(\bar{\theta}^{(m)}(t)) \right] \right| \geq p + B\iota \right) \leq (T/\iota + 1) \cdot \exp(-mp^2/B),$$

Setting $\iota = m^{-1/2}$ and $p = B \cdot \sqrt{\log(mT\delta)/m}$, we have that

$$\sup_{t \in [0,T]} \left| L(\bar{\theta}^{(m)}(t)) - \mathbb{E}_{\rho_t}\left[ L(\bar{\theta}^{(m)}(t)) \right] \right| \leq B \cdot \sqrt{\log(mT\delta)/m} \tag{D.28}$$

with probability at least $1 - \delta$. Plugging (D.26) and (D.28) into (D.25), noting that $t = k\epsilon$, we have that

$$\left| L(\bar{\theta}^{(m)}(k\epsilon)) - \bar{L}(\rho_{k\epsilon}) \right| \leq B \cdot \sqrt{\log(mT\delta)/m}, \quad \forall k \leq T/\epsilon \ (k \in \mathbb{N}) \tag{D.29}$$

with probability at least $1 - \delta$.

Plugging (D.24) and (D.29) into (D.20), we have that

$$\left| L(\theta^{(m)}(k)) - \bar{L}(\rho_{k\epsilon}) \right| \leq B \cdot e^{BT} \cdot \left( \sqrt{\log(m/\delta)/m} + \sqrt{\epsilon \cdot (D + \log(m/\delta))} \right), \quad \forall k \leq T/\epsilon \ (k \in \mathbb{N})$$

with probability at least $1 - \delta$. Thus, we complete the proof of Lemma D.6. $\qquad \square$

### D.3 Technical Lemmas for §D

In what follows, we present the technical lemmas used in §D. Recall that $\widehat{Q}$, $\widehat{g}$, and $\widehat{G}_k$ are defined in (3.1), (D.1), and (D.2), respectively. Let $B > 0$ be a constant depending on $\alpha, \eta, \gamma, B_r$, and $B_j$ ($j \in \{0, 1, 2\}$), whose value varies from line to line.

**Lemma D.7.** Under Assumptions 4.1 and 4.2, it holds for $\theta^{(m)} = \{\theta_i\}_{i=1}^m$ and $\underline{\theta}^{(m)} = \{\underline{\theta}_i\}_{i=1}^m$ that

$$\sup_{x \in \mathcal{X}} |\widehat{Q}(x; \theta^{(m)})| \leq B, \tag{D.30}$$

$$\sup_{x \in \mathcal{X}} |\widehat{Q}(x; \theta^{(m)}) - \widehat{Q}(x; \underline{\theta}^{(m)})| \leq B \cdot \|\theta^{(m)} - \underline{\theta}^{(m)}\|_{(m)}, \tag{D.31}$$

$$\|\widehat{G}_k(\theta_i; \theta^{(m)})\| \leq B, \tag{D.32}$$

$$\|\widehat{G}_k(\theta_i; \theta^{(m)}) - \widehat{G}_k(\underline{\theta}_i; \underline{\theta}^{(m)})\| \leq B \cdot \|\theta^{(m)} - \underline{\theta}^{(m)}\|_{(m)}, \quad \forall k \in \mathbb{N}, \tag{D.33}$$

$$\|\widehat{g}(\theta_i; \theta^{(m)})\| \leq B, \tag{D.34}$$

$$\|\widehat{g}(\theta_i; \theta^{(m)}) - \widehat{g}(\underline{\theta}_i; \underline{\theta}^{(m)})\| \leq B \cdot \|\theta^{(m)} - \underline{\theta}^{(m)}\|_{(m)}. \tag{D.35}$$

Meanwhile, for any $Q \in \mathcal{F}$, it holds that

$$\sup_{x \in \mathcal{X}} \|Q(x)\| \leq B. \tag{D.36}$$

For any $\rho \in \mathscr{P}_2(\mathbb{R}^D)$, it holds that

$$\|g(\theta; \rho)\| \leq B. \tag{D.37}$$

*Proof.* For (D.30) and (D.31) of Lemma D.7, following from Assumptions 4.1 and 4.2 and the definition of $\widehat{Q}$ in (3.1), we have for any $x \in \mathcal{X}$, $\theta^{(m)}$, and $\underline{\theta}^{(m)}$ that

$$|\widehat{Q}(x; \theta^{(m)})| \leq \alpha \cdot m^{-1} \sum_{i=1}^m |\sigma(x; \theta_i)| \leq B,$$

$$|\widehat{Q}(x; \theta^{(m)}) - \widehat{Q}(x; \underline{\theta}^{(m)})| \leq \alpha \cdot m^{-1} \sum_{i=1}^m |\sigma(x; \theta_i) - \sigma(x; \underline{\theta}_i)| \leq B \cdot \|\theta^{(m)} - \underline{\theta}^{(m)}\|_{(m)}.$$

For (D.32) and (D.33) of Lemma D.7, following from the definition of $\widehat{G}_k$ in (D.2), we have for any $\theta^{(m)}$ and $\underline{\theta}^{(m)}$ that

$$\|\widehat{G}_k(\theta_i; \theta^{(m)})\| = \alpha \cdot |\widehat{Q}(x_k; \theta^{(m)}) - r_k - \gamma \cdot \widehat{Q}(x_k'; \theta^{(m)})| \cdot \|\nabla_\theta \sigma(x_k; \theta_i)\| \leq B,$$

$$\|\widehat{G}_k(\theta_i; \theta^{(m)}) - \widehat{G}_k(\underline{\theta}_i; \underline{\theta}^{(m)})\|$$

$$\leq \alpha \cdot \sup_{\theta^{(m)}} |\widehat{Q}(x_k; \theta^{(m)}) - r_k - \gamma \cdot \widehat{Q}(x_k'; \theta^{(m)})| \cdot \|\nabla_\theta \sigma(x_k; \theta_i) - \nabla_\theta \sigma(x_k; \underline{\theta}_i)\|$$

$$+ \alpha \cdot |\widehat{Q}(x_k; \theta^{(m)}) - \gamma \cdot \widehat{Q}(x_k'; \theta^{(m)}) - \widehat{Q}(x_k; \underline{\theta}^{(m)}) + \gamma \cdot \widehat{Q}(x_k'; \underline{\theta}^{(m)})| \cdot \sup_{\theta_i \in \mathbb{R}^D} \|\nabla_\theta \sigma(x_k; \theta_i)\|$$

$$\leq B \cdot \|\theta^{(m)} - \underline{\theta}^{(m)}\|_{(m)}.$$

The inequalities in (D.34) and (D.35) of Lemma D.7 for $\widehat{g}$ follow from the fact that

$$\widehat{g}(\theta_i; \theta^{(m)}) = \mathbb{E}_{(x_k, r_k, x_k') \sim \widetilde{\mathcal{D}}} [G_k(\theta_i; \theta^{(m)})].$$

The inequalities in (D.36) and (D.37) follow from the definition of $\mathcal{F}$ and $g$ in (4.3) and (3.5), respectively. Thus, we complete the proof of Lemma D.7. $\qquad \square$

Recall that $\rho_t$ is the PDE solution in (3.4) and $\widetilde{\theta}^{(m)}(t)$ and $\bar{\theta}^{(m)}(t)$ are the CTTD and IP dynamics defined in (D.5) and (D.6), respectively.

**Lemma D.8.** Under Assumptions 4.1 and 4.2, it holds for any $s, t \in [0, T]$ that

$$\|\bar{\theta}^{(m)}(t) - \bar{\theta}^{(m)}(s)\|_{(m)} \leq B \cdot |t - s|, \tag{D.38}$$

$$\|\widetilde{\theta}^{(m)}(t) - \widetilde{\theta}^{(m)}(s)\|_{(m)} \leq B \cdot |t - s|, \tag{D.39}$$

$$\mathcal{W}_2(\rho_t, \rho_s) \leq B \cdot |t - s|. \tag{D.40}$$

*Proof.* For (D.38) of Lemma D.8, by the definition of $\bar{\theta}_i(t)$ in (D.6) and (D.37) of Lemma D.7, we have for any $s, t \in [0, T]$ and $i \in [m]$ that

$$\left\| \bar{\theta}_i(t) - \bar{\theta}_i(s) \right\| = \eta \cdot \int_s^t \left\| g\big(\bar{\theta}_i(\tau); \rho_\tau\big) \right\| \mathrm{d}\tau \leq B \cdot |t - s|.$$

Similarly, for (D.39) of Lemma D.8, by the definition of $\widetilde{\theta}_i(t)$ in (D.5) and (D.34) of Lemma D.7, we have for any $i \in [m]$ and $s, t \in [0, T]$ that $\|\widetilde{\theta}_i(t) - \widetilde{\theta}_i(s)\| \leq B \cdot |t - s|$.

For (D.40) of Lemma D.8, following from the fact that $\bar{\theta}_i(t) \overset{\text{i.i.d.}}{\sim} \rho_t$ $(i \in [m])$ and the definition of $\mathcal{W}_2$ in (2.4), it holds for any $s, t \in [0, T]$ that

$$\mathcal{W}_2(\rho_t, \rho_s) \leq \mathbb{E}\Big[\big\|\bar{\theta}_i(t) - \bar{\theta}_i(s)\big\|^2\Big]^{1/2} \leq B \cdot |t - s|.$$

Thus, we complete the proof of Lemma D.8. $\qquad\square$

**Lemma D.9** (Lemma 30 in [53])**.** Let $\{X_i\}_{i=1}^m$ be i.i.d. random variables with $\|X_i\| \leq \xi$ and $\mathbb{E}[X_i] = 0$. Then, it holds for any $p > 0$ that

$$\mathbb{P}\bigg(\Big\|m^{-1} \cdot \sum_{i=1}^m X_i\Big\| \geq C\xi \cdot (m^{-1/2} + p)\bigg) \leq \exp(-mp^2),$$

where $C > 0$ is an absolute constant.

**Lemma D.10** (Lemma A.3 in [6] and Lemma 31 in [53])**.** Let $X_k \in \mathbb{R}^D$ $(k \in \mathbb{N})$ be a martingale with respect to the filtration $\mathcal{G}_k$ $(k \geq 0)$ with $X_0 = 0$. We assume for $\xi > 0$ and any $\lambda \in \mathbb{R}^D$ that

$$\mathbb{E}\Big[\exp\big(\langle \lambda, X_k - X_{k-1}\rangle\big)\,\Big|\,\mathcal{G}_{k-1}\Big] \leq \exp\big(\xi^2 \cdot \|\lambda\|^2/2\big).$$

Then, it holds that

$$\mathbb{P}\Big(\max_{\substack{k \leq n \\ (k \in \mathbb{N})}} \|X_k\| \geq C\xi \cdot \sqrt{n} \cdot (\sqrt{D} + p)\Big) \leq \exp(-p^2),$$

where $C > 0$ is an absolute constant.

# E   Auxiliary Lemmas

We use the definition of absolutely continuous curves in $\mathscr{P}_2(\mathbb{R}^D)$ in [5].

**Definition E.1** (Absolutely Continuous Curve)**.** Let $\beta : [a, b] \to \mathscr{P}_2(\mathbb{R}^D)$ be a curve. Then, $\beta$ is an absolutely continuous curve if there exists a square-integrable function $f : [a, b] \to \mathbb{R}$ such that

$$\mathcal{W}_2(\beta_s, \beta_t) \leq \int_s^t f(\tau)\,\mathrm{d}\tau$$

for any $a \leq s < t \leq b$.

Then, we have the following first variation formula.

**Lemma E.2** (First Variation Formula, Theorem 8.4.7 in [5])**.** Given $\nu \in \mathscr{P}_2(\mathbb{R}^D)$ and an absolutely continuous curve $\mu : [0, T] \to \mathscr{P}_2(\mathbb{R}^D)$, let $\beta : [0, 1] \to \mathscr{P}_2(\mathbb{R}^D)$ be the geodesic connecting $\mu_t$ and $\nu$. It holds that

$$\frac{\mathrm{d}}{\mathrm{d}t} \frac{\mathcal{W}_2(\mu_t, \nu)^2}{2} = -\langle \mu_t', \beta_0'\rangle_{\mu_t},$$

where $\mu_t' = \partial_t \mu_t$, $\beta_0' = \partial_t \beta_t\,|_{t=0}$, and the inner product is defined in (2.5).

**Lemma E.3** (Talagrand's Inequality, Corollary 2.1 in [59])**.** Let $\nu$ be $N(0, \kappa \cdot I_D)$. It holds for any $\mu \in \mathscr{P}_2(\mathbb{R}^D)$ that

$$\mathcal{W}_2(\mu, \nu)^2 \leq 2D_{\mathrm{KL}}(\mu \,\|\, \nu)/\kappa.$$

**Lemma E.4** (Eulerian Representation of Geodesics, Proposition 5.38 in [68])**.** Let $\beta : [0, 1] \to \mathscr{P}_2(\mathbb{R}^D)$ be a geodesic and $v$ be the corresponding vector field such that $\partial_t \beta_t = -\operatorname{div}(\beta_t \cdot v_t)$. It holds that

$$\partial_t(\beta_t \cdot v_t) = -\operatorname{div}(\beta_t \cdot v_t \otimes v_t).$$