[Reviews · NeurIPS 2020]

Review 1

Summary and Contributions: This paper studies Temporal Difference (TD) learning with the value function represented by overparameterized neural networks. From the mean-field perspective, it shows that TD(0) for both policy evaluation and Q learning converges to the optimal solutions at a sublinear (1/T) rate. Moreover, the mean-field theory shows that, as the TD learning algorithm proceeds, the feature representation learned by the neural networks also evolves, which shows that neural network based TD learning algorithms actually are able to learn good features automatically.

Strengths: 1. The theoretical results seem solid. 2. TD(0) with general nonlinear function approximation is known to have divergent counterexamples. This paper address such a fundamental problem for a specific nonlinear function -- overparameterized neural networks, and establishes a sublinear convergence guarantee. This is similar to the prior work [21] which studies the convergence of neural TD learning via linearization and neural tangent kernel. [21] also establishes a sublinear convergence rate. 3. Using the mean-field analysis, this work also depicts the evolution of the kernel function induced by the neural network value function. This result shows that, under the mean-field regime, neural TD learning also learns a good feature representation. This phenomenon is not captured by the neural tangent kernel based analysis (e.g. [21]) as the kernel is always fixed to the NTK in that line of work. Representation learning for RL seems a less explored direction in theory while it seems to matter a lot in practice. Thus this work seems made a contribution to this important aspect.

Weaknesses: 1. This paper lacks motivation for why we should care about the mean field regime of TD learning or why representation learning is important. Given that there is previous work [21] that has already established the finite-time convergence for neural TD learning, it would be nice to first inform the readers of the limitations of the previous NTK analysis. 2. In the title, this paper seems to claim that “Representation learning” is the main contribution. However, it seems that the related discussion is only in the paragraph after Theorem 4.3. It would be nice to further elaborate on this discussion so as to strengthen the argument. 3. Technically, one limitation of this work is that the convergence analysis is only in a continuous-time fashion. In contrast, [21] has a finite-time convergence guarantee. I understand that this is due to using different technical tools and cannot be directly compared. This paper uses mean-field theory while [21] uses NTK theory which approximates the neural network value functions by linear functions and the convergence analysis is based on convex optimization tools.

Correctness: I didn't check the proof line by line but the theory seems reasonable and the proof seems correct.

Clarity: The paper is overall very well-written. However, as I mentioned above, the presentation can be further improved by adding: (1) the motivation for the mean-field setting and the representation learning problem for neural TD learning; (2) discussions regarding the features learned from the algorithm; (3) comparison with NTK-based analysis for neural TD learning in terms of analysis.

Relation to Prior Work: Yes. This paper discusses and distinguishes from previous work properly. A missing related work: A Finite-Time Analysis of Q-Learning with Neural Network Function Approximation by Xu and Gu, 2019.

Reproducibility: Yes

Additional Feedback: [1] It would be nice to compare with [21] in terms of proof techniques. This would perhaps make the readers more clear about why the mean-field regime is able to learn representations while NTK is not. [2] How do the sublinear rates in this work and [21] compare? Can they directly compare with each other? Since one is continuous-time and the other one is discrete-time, we might need to be more careful when comparing these rates. [3] Is it possible to establish a finite-time convergence rate by discretizing the trajectory of ODE? [4] I understand that this work is a theoretical work, but it would be nice to see how far the mean-field regime is from the practical neural TD learning methods.


Review 2

Summary and Contributions: This paper studies the problem of global convergence of Temporal Difference learning with a specific (but rich) class of non-linear function approximation. The study leans on the mean-field limit theory and focuses on the behavior of the feature representation learned by a two-layer neural network. Authors show the fixed point of the projected Bellman error, an objective function proposed in previous work for producing gradient-based TD learning, and shows that the solution found by TD decays to zero at a sub-linear rate. -- Post rebuttal: I was really hoping to see the numerical results on famous counterexamples. Though this was not provided, I am still happy to see this paper getting in. I really hope that such results will be included in camera ready, and I suggest a format akin to that of Figure 1 from Maei et al., 2010.

Strengths: The work proposes an interesting and novel application of mean-field limit theory for understand TD with non-linear function approximation. This is definitely an interesting research avenue. I have issues with some of the claims made by authors, as well as some of the underlying assumptions, which undermines the soundness of the claims to a limited extent.

Weaknesses: - The paper promises results on, not only the prediction case with TD, but also with Q-learning and control. Yet, the two assumptions B1 and B3 in the appendix are super strong. In fact, it took a while for me to come up with examples where these assumptions hold. What's more, the validity of the assumptions seem to be a property of the problem, and even within a problem these assumptions may not hold on a per state basis. I think this paper needs to scale back its claims when it comes to Q-learning. I think the TD case is in and of itself interesting, and I find it disappointing that the paper has rushed to extend results to the control case by making these self-serving assumptions. I am happy to increase my score if authors scale back their claims pertaining to control. - It is worthwhile to mention that the TD update 3.3 is rarely being used with deep neural networks, because practitioners often find it necessary to use a second network, referred to as target network, to stabilize the learning process.

Correctness: The claims are correct based on the given assumptions, but I have already indicated my concern about some of the assumptions in the control case.

Clarity: Notationally the paper can improve. As a concrete example, the quantity Q^{*} has a specific solution-agnostic (as opposed to solution-dependent) meaning in RL, and has been used here (for example in 4.4) to indicate something different than the classic meaning of Q^{*}. But overall, this is a well-written paper.

Relation to Prior Work: Yes.

Reproducibility: Yes

Additional Feedback: I don't see where we need to use Wasserstein 2 as opposed to its more generic form. Maybe to upper bound it easier, or maybe in Lemma 5.2? Can you clarify? It is worthwhile to mention that Assumption 4.1 may not hold in some settings regardless of how large the normalization factor is. The use of MSPBE as opposed to MSBE is being justified due to failure of function approximator to represent TQ, yet you are studying neural networks that are actually universal function approximators, and therefore capable of representing arbitrarily complicated (but I guess Lipschitz) functions. Can you clarify? How do you square the example from Figure 1 of Tsitsiklis and Van Roy with your result? Via over parameterization? And if yes, can you numerically confirm that such a neural net actually converges in the example provided by Tsitsiklis and Van Roy? And/or in Baird's counter example.


Review 3

Summary and Contributions: The paper presents a novel theoretical result regarding the convergence of TD and Q-learning when the action-value function is approximated using a two-layer neural network. Concretely, the authors show that the action-value function converges at a sublinear rate to within a constant factor of optimal, where the constant error O(1/alpha) depends on the scaling factor alpha applied to the output of the neural network. The result follows from a mean-field analysis that converts the TD dynamics in finite-dimensional parameter space to an infinite-dimensional Wasserstein space.

Strengths: To the best of my knowledge, the proofs are correct, and I believe that the result is an important contribution to the fields of reinforcement learning and deep learning, since there are relatively few positive theoretical results about deep learning.

Weaknesses: In my view (and given my background), the main weakness is that some of the proofs are difficult to follow.

Correctness: As mentioned above, I find some proofs hard to follow, but I have not been able to find any flaws, so I believe them to be correct.

Clarity: For the most part the paper is well-written, but sometimes it omits notation and definitions (see my comments below).

Relation to Prior Work: I believe that the authors do an adequate job of discussing related work.

Reproducibility: No

Additional Feedback: In the definition of the continuity equation, what does "div" stand for? And how it is defined? The definition of Q-hat in (3.1) implies that the activation function sigma is only applied in the first layer of the network. (At first I though the output was an *unweighted* linear combination of the intermediate nodes, but each \theta_i can apparently include a parameter for the second layer as well.) How much harder would the problem be to analyze if the second layer also applied an activation function? I guess dimensions D and d should be closely related, e.g. D = d + 2 since each \theta_i usually defines d + 1 parameters for d inputs, plus one parameter for the second layer. I assume the notation \delta_{\theta_i} on line 150 is the Dirac delta? As a result of not knowing how "div" is defined, the transformation from (3.3) to (3.4) is difficult to parse. This transformation is critical to understanding how the proof of Theorem 4.3 works, so I would try to make it more explicit (and this is also the reason I put "No" for reproducibility). In Theorem 4.3, the definition of \rho-bar is rather implicit (it is the distribution corresponding to the fixed point Q^* of MSPBE). Consequently it took me a while to parse what \rho-bar refers to in (4.4). Author feedback: I think the authors did a good job of responding to the concerns of reviewers, and I appreciate the efforts to make the paper and proofs more readable (by including a flowchart and clarifying several definitions).

[Author Response · NeurIPS 2020]

We appreciate the valuable comments from the reviewers. We will revise accordingly.

**Reviewer #2.** (Motivation of mean-field regime.) As discussed in Lines 27-31, 47-50, and 75-76, the empirical success
of deep RL is empowered by its ability to learn data-dependent feature representation. However, the NTK-based
analysis of TD [21] requires an implicit local linearization with respect to the initial feature representation, which
is not data-dependent, and thus, fails to explain how the feature representation evolves. In contrast, the mean-field
regime allows the feature representation evolving. Specifically, for the induced kernel $\mathbb{K}(\cdot, \cdot; \rho_t)$ defined in (3.7), our
mean-field regime allows $\rho_t$ to go beyond $\rho_0$, while the NTK regime requires $\rho_t = \rho_0$ (Lines 214-227). (Comparison to
the NTK-based analysis.) The NTK-based analysis requires a proper scaling of the neural network to allow the implicit
local linearization (Lines 43-45), while our analysis does not require linearization. Moreover, the analysis in [21] is
based on the one-point monotonicity in the Euclidean space, while we generalize such a notion to the Wasserstein space
(Lines 63-66). (Representation learning.) We discuss the representation learning in Lines 27-31, 47-50, 75-76, 176-184,
and 214-227. We study the evolution of the induced kernel $\mathbb{K}(\cdot, \cdot; \rho_t)$ defined in (3.7), which is fully characterized by
$\rho_t$. We show the global convergence of $\rho_t$ in Theorem 4.3, which implies that the induced kernel also converges to
the globally optimal one. (Discretization.) We study the discretization of the trajectory of PDE in Proposition 3.1 and
Appendix D, based on which we establish a discrete-time convergence rate in Corollary 4.4 by aggregating the the
discretization error. (Missing reference.) We will cite the paper in our revision. Thank you for pointing out.

**Reviewer #3.** (Assumptions for Q-learning.) As discussed in Lines 477-478, similar assumptions are employed in the
analysis of Q-learning in simpler settings (linear or NTK). On the other hand, we do understand that Assumptions B.1
and B.3 are strong by themselves. Thus, we put Q-learning in the appendix as an extension of our main results for TD.
In the revision, we will not claim the convergence of Q-learning as our contribution and emphasize the restrictiveness of
such assumptions. (Target network.) When a target network is employed, TD becomes a bilevel optimization problem,
in which case the convergence can be proved by similar technical tools. (Wasserstein 2 distance.) Similar to the $\ell_2$
distance in $\mathbb{R}^d$, the Wasserstein 2 distance induces an "inner product" (more precisely, a weak Riemannian metric) on
the space of probability measures and is well studied in the literature of optimal transport, which forms the basis of
our analysis. It may be possible to generalize our results to the Wasserstein $p$ distance by exploiting the duality of
the $p$ and $q$ norms, where $1/p + 1/q = 1$. (Assumption 4.1.) As discussed in Lines 189-191, similar assumptions are
commonly used in the mean-field analysis of neural networks and can be ensured through normalizing the state-action
space. Moreover, our analysis can be straightforwardly generalized to the setting where $\|x\| \leq C$ for an absolute
constant $C$. Such a setting covers a majority of RL problems, but yes, we do agree that Assumption 4.1 doesn't always
hold, especially when the state or action space is unbounded. (MSPBE and universal function approximation.) As
discussed in Lines 198-199, our function class defined in (4.3) captures a rich class of functions because of the universal
approximation theorem (UAT). It is worth noting that UAT requires additional conditions on the target function, e.g.,
an upper bounded first moment of the Fourier coefficients [11]. As UAT doesn't ensure the approximation of *any*
target function, we use MSPBE rather than MSBE. (The example in Tsitsiklis and Van Roy.) Yes, we square the
counterexamples of Tsitsiklis and Van Roy (1997) and Baird (1995) via overparameterization. The divergence in their
examples comes from the nonconvexity and the bias of the semigradient. In contrast, we show in Lemma C.1 that,
coupled with an infinitely wide neural network, TD becomes a weakly convex problem (in the sense of one-point
monotonicity) with respect to the distribution of the parameter in the Wasserstein space. We will add the numerical
example in the revision.

**Reviewer #4.** (A flowchart of the proof.) The proof is technical and requires certain preliminary knowledge on optimal
transport, such as the Wasserstein gradient flow. We will include the following flowchart of the proof in the revision.

| Lemma 5.1: Existence of the optimal solution | → | Theorem 4.3: Global optimality and convergence |

| Lemma 5.2: Differential form of descent lemma |

| Lemma C.1: One-point monotonicity in the Wasserstein space | | Lemma C.2: Technical bounds |

(The definition of div.) The operator div is the divergence operator from vector calculus. We will specify the meaning of
div in our revision. Thank you for pointing out. (Activation function in the second layer.) As discussed in Lines 194-203,
we apply the activation function only to the first layer of our neural network, which is commonly used in the mean-field
analysis of neural networks. With an activation function applied to the second layer, our analysis still carries over but
becomes more involved to present. (Relation of $d$ and $D$.) Yes, the dimensions $D$ and $d$ are closely related, which are
used to cover the common cases in the study of neural networks. (Relation of (3.3) and (3.4).) As discussed in Lines
163-175, (3.4) can be viewed as the continuous-time and infinite-width limit of (3.3), while (3.3) can be viewed as the
discretization of (3.4). In Proposition 3.1, we show that (3.3) approximates (3.4) in the limit, whose detailed proof is
included in Appendix D. (Dirac delta.) Yes, the notation $\delta_{\theta_i}$ in Line 150 is the Dirac delta. (Definition of $\bar{\rho}$.) We will
define $\bar{\rho}$ explicitly with a standalone line in the revision. Thank you for pointing out.

[Meta-Review · NeurIPS 2020]

The paper presents some new results regarding the convergence of TD and Q-learning when the action-value function is represented by overparameterized neural networks. The theoretical contribution made by this paper is seen as solid. The weakness described by the reviewers are not major and can be addressed in a minor revision and I therefore recommend accepting this paper.